# CAV2-expressing nerves induce metabolic switch toward mitochondrial oxidative phosphorylation to promote cancer stemness

Ze Zhang [1,2,3,4,7] ✉, Chunli Wang[1,2,3,4,7], Zhonghao Sun[1,2,3,4,7], Xiaotian Shi[1,2,3,4], Yanjie Shuai[1,2,3,4], Yafei Wang[1,2,3,4], Yun Wang[1,2,3,4], Hua Guo [2,3,4,5] ✉, Ruoyan Liu [2,3,4,6] ✉ & Jingtao Luo [1,2,3,4] ✉

Cancer cells and the nervous system engage in a dynamic interplay, significantly influencing initiation and progression in head and neck squamous cell carcinoma (HNSCC). Our findings highlight that cancer cells drive an increase in caveolin-2 (Cav2) expression within trigeminal ganglia and associated neural fibers in the tumor milieu, fostering a reciprocal attractant relationship between tumor cells and nerves. Notably, the knockout of Cav2, either globally or specifically in sensory neurons or glial cells, markedly attenuates the growth of orthotopically implanted tongue tumors. Moreover, Cav2-expressing nerves are implicated in shifting cancer cell metabolism towards mitochondrial oxidative phosphorylation, a process involved in maintenance of cancer stem cells (CSCs). Our results also demonstrate that Cav2-expressing nerves confer stemness properties to cancer cells. Disruption of Cav2 expression, both globally and in specific neural cell types, impedes tumorigenesis and progression in a 4-NQO-induced HNSCC mouse model. This interplay observed between cancer cells, neurons, and glial cells suggests a potential mechanism through which tumor-associated nerves might influence cancer stemness via metabolic reprogramming. This highlights a possible direction for anticancer therapy that warrants further investigation.

There's a burgeoning consensus that nerves play a profound role in the growth and spread of solid tumors, marking a significant intersection between the fields of oncology and neuroscience[1–3]. As preneoplastic lesions transition to invasive forms of cancer, a notable increase in nerve density can be observed, especially in cases of head and neck squamous cell carcinoma (HNSCC), prostate cancer, and pancreatic cancers. These nerves aren't mere bystanders; they actively orchestrate tumor initiation, modulate its progression, and potentially drive metastasis[1,4]. In a reciprocative manner, tumors have been found to adapt and exploit the structures and functions of the nervous system

[1]Department of Head and Neck Oncology, Tianjin Medical University Cancer Institute & Hospital, Tianjin, China. [2]National Clinical Research Center for Cancer, Key Laboratory of Cancer Prevention and Therapy, Tianjin, China. [3]Tianjin Key Laboratory of Basic and Translational Medicine on Head & Neck Cancer, Tianjin Medical University Cancer Institute and Hospital, Tianjin, China. [4]Tianjin's Clinical Research Center for Cancer, Tianjin Medical University Cancer Institute and Hospital, Tianjin, China. [5]Department of Tumor Cell Biology, Tianjin Medical University Cancer Institute and Hospital, Tianjin, China. [6]Department of Gynaecological Oncology, Tianjin Medical University Cancer Institute & Hospital, Tianjin, China. [7]These authors contributed equally: Ze Zhang, Chunli Wang, Zhonghao Sun. ✉e-mail: zhangze@tmu.edu.cn; guohua@tjmuch.com; yanruoliu@tmu.edu.cn; jluo@tmu.edu.cn

for their benefit[2]. Targeting this bidirectional interaction promises a therapeutic frontier, yet the dearth of comprehensive molecular insights into nerve-tumor interactions currently impedes the advent of innovative interventions. The urgency now lies in decoding this cellular dialogue, paving the path for potential therapeutic breakthroughs.

An in-depth analysis of The Cancer Genome Atlas (TCGA) datasets underscores the prominence of caveolin-2 (CAV2) as a top prognostic indicator for HNSCC[5], a cancer known for its widespread innervation by both sensory and adrenergic neural networks[4,6,7]. While TCGA datasets derive from bulk tumor sequencing, the specific cellular source of CAV2 in the tumor microenvironment has not been well delineated. Furthermore, single-cell RNA-seq (scRNA-seq) studies on HNSCC have inadvertently ignored neurons and glial cells[8,9]. Here we have discerned that CAV2, a main component of the caveolae in plasma membranes[10], finds expression within the nerves in HNSCC tumor microenvironment, establishing a crucial nexus between tumors and nerves.

Growing evidence suggests that nerves contribute to tumor initiation[11]. This claim is corroborated by research findings that underscore the regulatory influence of nerves and neuronal activity in the onset of gliomas in the brain[12,13]. Similarly, in solid tumors beyond the nervous system, the Peripheral Nervous System (PNS) also plays a role in cancer initiation. For instance, the sympathetic nerve has been identified as integral to prostate cancer initiation[14]. It has been further illustrated that nerve ablation diminishes pancreatic intraepithelial neoplasia (PanIN) precursor lesions and restricts progression to the pancreatic adenocarcinoma stage[15,16]. To date, the influence of nerves on the initiation of HNSCC and the relevant mechanisms remains poorly understood. Cancer cells predominantly use aerobic glycolysis for energy production[17] yet recent studies suggest a heightened reliance on oxidative phosphorylation by tumor-initiating cells or cancer stem cells (CSCs)[18–20]. These CSCs, characterized by self-renewal and pluripotency, are vital for tumor initiation, drug resistance, and recurrence[21,22].

In this work, we demonstrate that nerves expressing Cav2, compared to those lacking Cav2, drive HNSCC cells to adopt a mitochondrial oxidative phosphorylation phenotype to maintain tumor stemness. We also investigate whether these Cav2-expressing nerves promote the initiation and progression of HNSCC by using multiple models, including a 4-Nitroquinoline (4-NQO)-induced mouse model that recapitulates the initiation, development, and metastasis of human HNSCC[23]. We demonstrate that the genetic ablation of Cav2 in both neuronal and glial cells significantly impedes tumorigenesis and decreases tumor incidence. These results unveil a metabolic axis between nerves and cancer stemness, crucial in promoting both tumorigenesis and tumor progression.

## Results

### Localization of CAV2 in Neural Structures within the HNSCC Microenvironment and Its Prognostic Implications

In delineating the role of CAV2 within HNSCC, we assessed the mRNA expression differential between HNSCC and adjacent normal tissues utilizing the TCGA dataset. This revealed notable overexpression of CAV2 in HNSCC (Supplementary Fig. 1A). Examining its clinical relevance, we found no significant ties between CAV2 expression and clinical tumor (T) or nodal **N** stages (Fig. 1A, B). Likewise, its expression across varied histological grades remained consistent (Supplementary Fig. 1B). Notably, it was increased in patients with perineural invasion (PNI) − an established adverse pathological feature in HNSCC (Fig. 1C), signifying its potential involvement in tumor-nerve crosstalk. Additionally, CAV2 expression lacked correlation with lymphovascular invasion (Fig. 1D) and extracapsular extension (ECE) (Supplementary Fig. 1C), suggesting its selective influence on neural rather than global tumor invasiveness. Furthermore, patients negative for human

papillomavirus (HPV), who bear a heightened predisposition to PNI[24], exhibited elevated CAV2 expression (Supplementary Fig. 1D). This trend was mirrored in specific HNSCC anatomical zones, such as the tongue, where PNI incidence is notably high (Supplementary Fig. 1E).

Prompted by these findings, we delved into CAV2 expression dynamics within the HNSCC microenvironment, with a particular emphasis on the nuanced interplay between tumors and nerve structures. Histological evaluations of HNSCC samples revealed a pronounced enrichment of CAV2 within neural structures, relative to other components in the tumor microenvironment (Fig. 1E). Notably, apart from its primary neural localization, vascular sites also evidenced CAV2 expression (Supplementary Fig. 1F). The absence of an association between CAV2 expression and lymphovascular invasion, limited our pursuit to explore endothelial CAV2 further. Evaluating the HNSCC innervation landscape, we identified dense TRPV1+ (sensory) and some tyrosine hydroxylase (TH; adrenergic) positive fiber populations, consistent with earlier studies[6,7]. Multiplex immunohistochemistry (IHC) data illustrated the pervasive expression of CAV2 across TRPV1+ sensory fibers (Fig. 1F) and a peripheral expression pattern in TH+ fibers, hinting at neural sheath structures (Fig. 1G).

To elucidate the prognostic implications of CAV2, we initiated our analysis by correlating CAV2 mRNA levels, obtained via bulk sequencing, with patient survival in the TCGA. Our findings highlighted a marked association between elevated CAV2 expression and poorer prognosis (Fig. 1H). To delve deeper, we categorized patients from our cohort based on the staining scores of neural CAV2. Remarkably, those with higher neural CAV2 scores demonstrated diminished survival outcomes (Fig. 1I, J). This suggests that the prognostic influence of CAV2, as identified through bulk sequencing, might predominantly stem from the consequential effects of neural CAV2 on patient prognosis.

### HNSCC Induces Upregulation of Neural CAV2 Expression

Investigating the influence of cancer cells on neural Cav2 expression, trigeminal ganglia (TGs) were co-cultured with SCC15 and MOC2 cells in vitro for a 5-day period. We observed a pronounced and immediate elevation in Cav2 protein expression within the first day of co-culturing with both HNSCC cell types. Although this expression waned on starting day 3, the levels on day 5 remained higher compared to TGs not exposed to cancer cells (Fig. 2A). To reinforce these observations, we subjected the TGs to enzymatic dissociation, co-cultured the resultant cell populations with tumor cells, and subsequently assessed in immunofluorescent evaluation. This corroborated the heightened Cav2 expression, particularly during the early co-culture phase (Fig. 2B). Remarkably, Cav2 localization was detected in neuronal growth cones, highlighting its potential significance in axonal pathfinding (Fig. 2C). Despite the widespread expression of Cav2 in non-neuronal cells of the TG (primarily satellite cells and Schwann cells, along with a small number of fibroblasts), its expression levels were not significantly altered by interaction with tumor cells (Fig. 2D). Single-nucleus RNA sequencing of TGs also indicated that Cav2 is expressed in neurons, satellite cells, Schwann cells, and fibroblasts (Supplementary Fig. 2)[25]. With caveolins recognized as major proteins of caveolae, which are diminutive plasma membrane invaginations, we examined these structures in differentiated PC12 cells−a neuronal-like model− and dorsal root ganglia (DRG). Transmission electron microscopy revealed caveolae-like structures, the prevalence of which increased upon cancer cell co-culture (Fig. 2E, F). It is noteworthy that the expression levels of CAV1 and CAV3 primarily determine the formation of caveolae, while the expression level of CAV2 has minimal impact on caveolae formation[26]. Therefore, the increase in caveolae could be due to the upregulation of neural CAV1 expression following co-culture (noting that CAV3 is predominantly expressed in muscle tissues).

We next probed whether the TG, which houses the neuronal somas for nerves that innervate the tongue, manifests Cav2

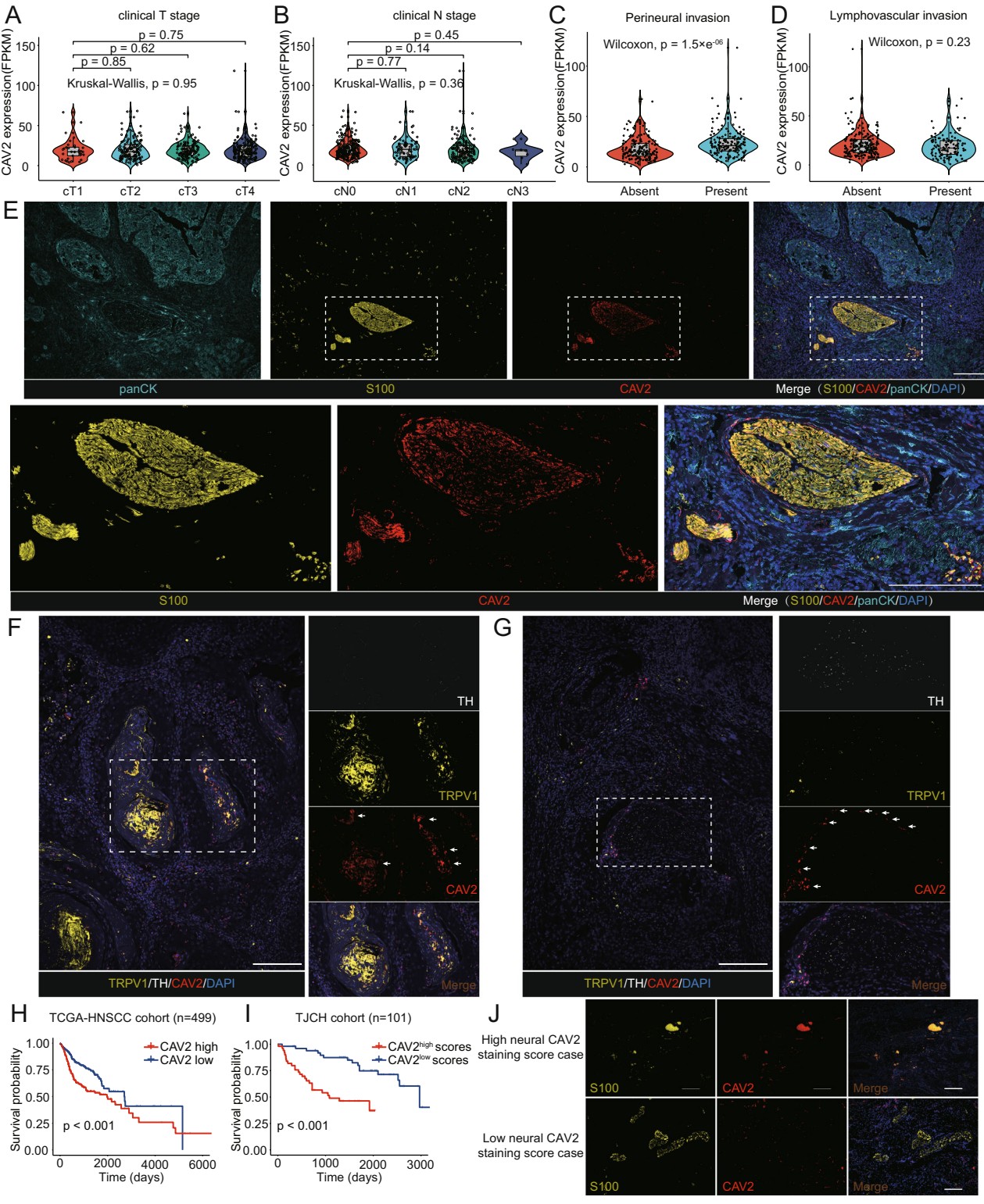

expression, and if such expression is perturbed by oral cancer in vivo. Through the use of retrograde labeling, we marked tongue-innervating neurons and detected Cav2 expression on the surfaces of a subset of TG neuron. Enhanced expression was observed in TG neurons from mice orthotopically introduced with MOC2 cells (Fig. 2G). We also discerned Cav2 expression within the nucleus of certain neuronal groups, a pattern that remained largely unchanged upon introduction of orthotopic MOC2 xenografts (Fig. 2H). It is noteworthy that we performed retrograde labeling prior to implanting xenografts. This procedure labels resident nerves, rather than nerves that sprout and

newly infiltrate the tumor as it grows. In order to effectively assess the expression of Cav2 in tumor-associated neoneurogenesis, we conducted an additional set of experiments. Initially, we implanted slower-growing MOC1 cells into mice. After 7 days, we injected DiI to perform retrograde labeling. We also observed a significant increase in Cav2 expression in the cytoplasm and on the plasma membranes of neurons that were innervating the areas where MOC1 cells were originally implanted (Fig. 2I). To elucidate shifts in neural Cav2 expression in oral nerve fibers, we employed a 4-NQO mouse model of HNSCC that mirrors human disease progression. During tumorigenesis, we

**Fig. 1 | Localization of CAV2 in nerves within the HNSCC microenvironment and its association with prognosis.** Using sequencing data from the TCGA cohort's bulk tissue, correlations between CAV2 expression and clinicopathologic features, including **A** clinical T stage, **B** clinical N stage, **C** PNI, and **D** lymphovascular invasion, are presented. Violin plots show data distribution with the overlaid box plot indicating the median, first (Q1) and third (Q3) quartiles, and the whiskers extending to the 1.5x interquartile range (IQR). Two-sided statistical test. T1: $n = 33$; T2: $n = 143$; T3: $n = 130$; T4: $n = 179$; N0: $n = 239$; N1: $n = 80$; N2: $n = 152$; N3: $n = 7$; PNI + : $n = 165$; PNI-: $n = 186$; lymphovascular invasion + : $n = 120$; lymphovascular invasion-: $n = 219$. **E** Multiplex IHC, using antibodies against CAV2, epithelial marker PanCK, and nerve marker S100, reveals the localization of CAV2. Results were consistent across tissue sections from 15 patients. Subsequent panels offer detailed zoom-ins. Scale: 200 μm. **F** Images highlight CAV2 expression throughout TRPV1[+] sensory neural fibres in human HNSCC tissue, verified in 12 independent patient samples. Scale: 200 μm. **G** CAV2 is shown to be expressed on the periphery of TH[+] adrenergic neural fibres in human HNSCC tissue; data independently replicated in 12 patient specimens. Scale: 200 μm. **H** Kaplan−Meier analysis demonstrates the correlation between overall survival and CAV2 expression, based on TCGA cohort sequencing. Two-sided log-rank test. $n = 499$ patients. **I, J** Kaplan−Meier plots illustrate overall survival of Tianjin Medical University Cancer Institute and Hospital cohort (TJCH) in relation to neural CAV2 expression levels, determined by immunohistochemical staining scores, accompanied by representative images of nerves with high CAV2 staining scores and low scores. Scoring integrated neural staining intensity and extent. Intensity: 0 (negative) to 3 (strongly positive). The final score is calculated by multiplying the staining intensity score with the corresponding proportion of the stained neural area, and then summing these values to obtain a range from 0 to 300. Scale: 200 μm. Two-sided log-rank test. $n = 101$ patients.

observed a stepwise increase in Cav2[+] nerve densities, advancing from healthy mucosa to pronounced malignancy (Fig. 2J–L). A rise in the Cav2[+] nerve fraction relative to sensory nerve counts was also evident through tumor progression (Fig. 2M). To comprehensively evaluate all tumor-infiltrating nerves, we employed an additional staining panel that included the pan-neuronal marker β3-tubulin. This approach yielded similar results, showing that Cav2[+] nerves increased progressively with the advancement of the lesions (Supplementary Fig. 3A). Summarizing, our findings suggest oral carcinomas can influence neural Cav2 levels in both nerve fibers and somas.

### Depletion of neural Cav2 impedes tumor cell infiltration toward nerves and tumor-associated neurogenesis

Next, we evaluated the potential influence of neural Cav2 on nerve-directed tumor cell infiltration. Using a transwell migration assay, a marked elevation in the migration of both human SCC15 and murine MOC2 cells was observed when co-cultured with mouse TGs or DRGs, compared to controls. However, introducing Cav2[+/-] or Cav2[-/-] DRGs or TGs markedly countered this trend. Comparable attraction was observed with homozygous and heterozygous ganglia, suggesting Cav2 haploinsufficiency in nerves (Fig. 3A). To validate our observations, we employed a Matrigel model wherein cancer cells were cultured adjacent to a Matrigel drop containing TG, connected via a Matrigel bridge. This setup facilitates bidirectional interactions between tumor and nerve cells. We found that the tumor invasion towards the nerve index (α/β) markedly decreased upon substituting wild-type TGs with Cav2 knockout variants. Concurrently, there was a reduction in neurite length compared to the wild-type baseline (Fig. 3B). To specifically investigate the effects on neuronal cells, exclusive of glial cell interaction, ganglionic tissue was enzymatically digested to yield isolated neurons for co-culture with tumor cells. Comparative analysis revealed that axonal length and neural filament quantity in Cav2[+/+] neurons co-cultured with tumor cells notably exceeded those in Cav2[-/-] counterparts. However, when cultured independently of tumor cells, no significant differences were observed between Cav2[+/+] and Cav2[-/-] neuronal phenotypes (Fig. 3C and Supplementary Fig. 3B).

With regard to the underlying mechanism, we found insights from the study by Ambre Spencer to be informative[27]. Based on this study, we observed that under NGF stimulation, PC12 cells overexpressing CAV2 exhibited a significant increase in the proportion of cells undergoing neurite outgrowth compared to controls, whereas CAV1 overexpression resulted in a significant decrease in neurite outgrowth (Fig. 3D). Ambre Spencer's research demonstrated that CAV1 inhibits NGF receptor (p75NTR) endocytosis and subsequent downstream signaling by sequestering these receptors within lipid rafts. In contrast, we found that CAV2 slightly enhanced NGF receptor endocytosis and downstream pathway activation (Fig. 3E, F). Given that NGF, as a critical neurotrophic factor, is markedly upregulated in HNSCC tumor tissue[28],

this may represent a potential mechanism through which neural CAV2 promotes tumor-nerve interaction.

### Host Cav2 loss or neural Cav2 deficiency attenuate tumor progression

To elucidate the impact of host-expressed Cav2 on neoplastic progression, murine-derived HNSCC cell lines MOC1 and MOC2 were orthotopically transplanted into wild-type and Cav2-deficient (Cav2[-/-]) C57BL/6 mice (Fig. 4A). A significant suppression of tumor growth in Cav2-deficient mice was recorded for both HNSCC lines. This was highlighted by a dramatic decrease in MOC2 and MOC1 tumor volume in Cav2[-/-] mice by nearly 17-fold and 23-fold, respectively, compared to their wild-type counterparts (Fig. 4B, C). The results indicate that CAV2[-/-] mice exhibited a significantly lower incidence of PNI compared to CAV2[+/+] mice (Fig. 4D). Additionally, quantitative analysis of β3-tubulin staining was performed to determine the impact of Cav2 loss on nerve density within the tumor microenvironment. An augmentation of β3-tubulin staining was observed throughout the tongues of wild-type mice, whereas, within neoplastic regions, the difference was not statistically significant, potentially as a result of the tumor's rapid expansion in wild mice compressing the neurogenic niche (Fig. 4E). Subsequent immunohistochemical analyses using Ki-67 and cleaved caspase-3, as well as Hmgb1 markers to probe tumor proliferation, apoptosis, and necrosis, yielded no notable disparities between the two mouse cohorts (Supplementary Fig. 4A−C). Considering previous reports that Cav2 host deficiency may hinder tumor-promoted angiogenesis[29], we assessed microvascular density via CD31 immunostaining, which revealed no substantial differences (Supplementary Fig. 4D). Given that mTORC1 is commonly activated during metabolic changes in tumors, we evaluated the phosphorylation status of ribosomal protein S6 at sites S235/236, a downstream marker of mTORC1 pathway activation, as a general indicator of metabolic alterations. Strikingly, phosphorylation of S6 was significantly higher in wild-type mice compared to Cav2-deficient mice, and this elevation was particularly prominent at the invasive tumor margin (Fig. 4F). These findings implicate host Cav2 in steering metabolic processes associated with HNSCC progression.

Building upon our prior demonstration that Cav2 predominantly localizes within the nerves of the HNSCC microenvironment, we sought to elucidate the effects of neural-specific Cav2 ablation on tumor progression. The absence of appropriate Cre recombinase lineages precludes a comprehensive neural-specific Cav2 deletion across all neural tissues. Given that sensory nerves predominate over TH[+] and VAChT[+] fibers in the HNSCC milieu and that Cav2 expression is chiefly associated with these sensory nerves, we generated Adv[Cre]Cav2[f/f] mice by breeding Cav2[f/f] with Adv-Cre mice (Fig. 4G). This model exploits the ubiquitous presence of Advillin in sensory neurons within the DRG and TG[30], thereby achieving Cav2 deletion in these cells. Upon orthotopic implantation of MOC2 cells, Adv[Cre]Cav2[f/f]

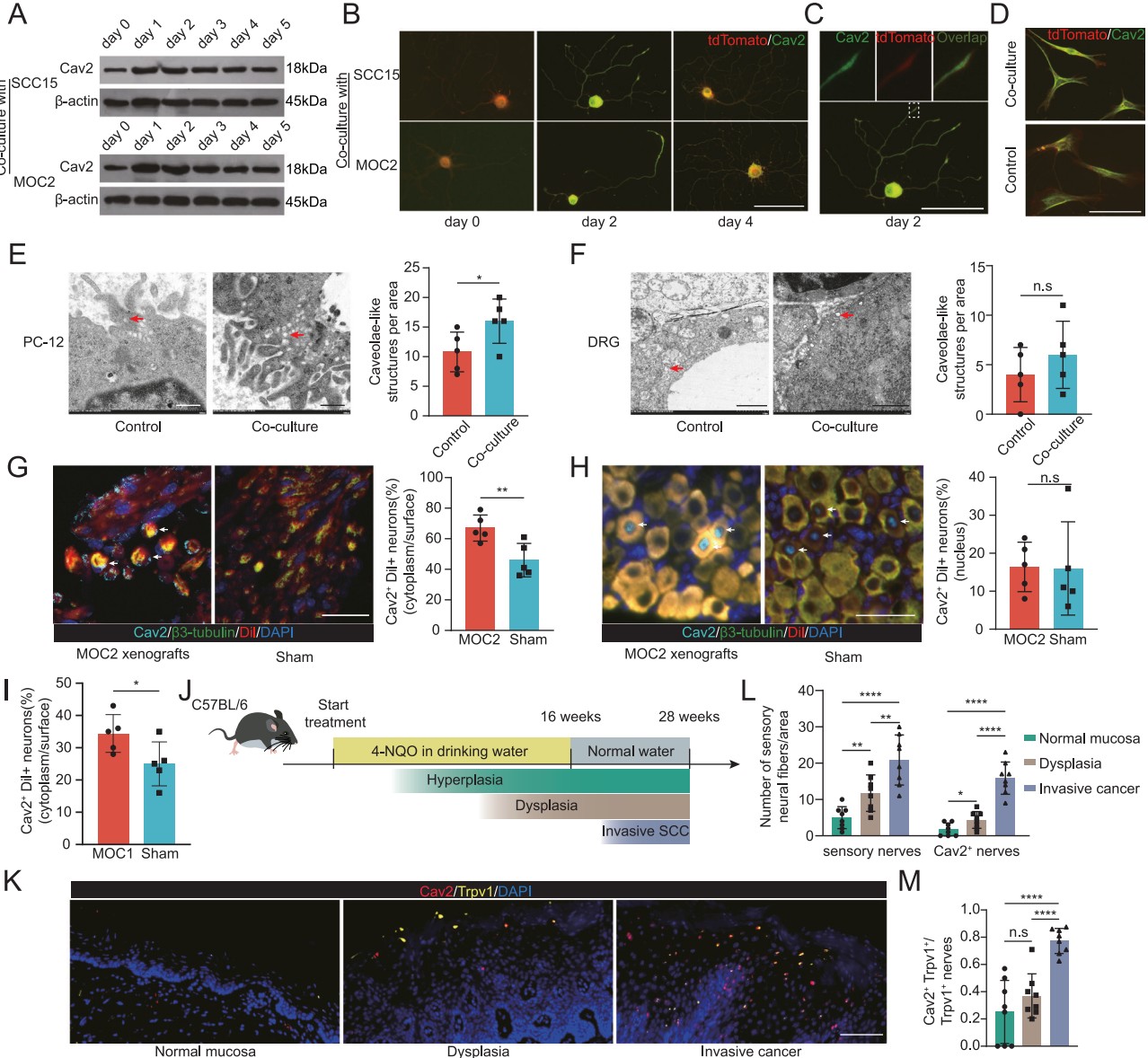

**Fig. 2 | Alterations in neural Cav2 interaction with cancer cells and its up-regulation during HNSCC progression. A** Immunoblot analysis of Cav2 in TGs that were either cultured alone (day 0) or co-cultured with SCC15 or MOC2 cells for 5 days. This experiment was independently repeated three times. **B** Immunofluorescence studies reveal an up-regulated Cav2 expression in TG neurons from B6.129(Cg)-Gt(ROSA)26Sor$^{tm4(ACTB-tdTomato,-EGFP)Luo}$/J mice co-cultured with SCC15 or MOC2 cells. Results were consistent across images from 10 cells. Scale: 100 μm. **C** Subcellular location of Cav2 in the growth cone of TG neurons co-cultured with SCC15. Results were consistent across images from 10 cells. Scale: 100 μm. **D** Cav2 expression in non-neuronal cells within the TG does not undergo significant changes after co-culturing with SCC15. Results were consistent across images from 10 cells. Scale: 100 μm. **E** Transmission electron microscopy of PC12 cells solo- vs. co-cultured with SCC15, highlighting caveolae-like structures (50–100 nm). Scale: 500 nm. $n = 5$ biologically independent co-cultures/group. *$p = 0.0492$, Student's t-test (two-sided). Data are mean ± s.d. **F** Representative images and quantitative analysis of caveolae-like structures in DRG cells. Scale: 1μm. $n = 5$ biologically independent co-cultures/group. n.s: $p = 0.3349$, Student's t-test (two-sided). Data are mean ± s.d. **G** Immunofluorescence and analysis of Cav2 on

DiI-labeled TG neurons from MOC2 xenografts versus sham. Sham controls received injections of tumor-cell–free buffer (30 μL of DMEM/Matrigel at a 1:1 ratio). These images depict Cav2 expression primarily on the surface of certain TG neurons Scale: 40μm. $n = 5$ mice per group. **$p = 0.0096$, Student's t-test (two-sided). Data are mean ± s.d. **H** Cav2 nuclear expression in TG neurons from MOC2 xenografts versus sham. Scale: 100μm. $n = 5$ mice per group. n.s: $p = 0.9502$, Student's t-test (two-sided). Data are mean ± s.d. **I** Quantitative analysis illustrating the prevalence of Cav2 surface expression on DiI$^+$ neurons in both MOC1 and sham conditions. $n = 5$ mice per group. *$p = 0.475$, Student's t-test (two-sided). Data are mean ± s.d. **J** An illustrative diagram depicting the 4-NQO-induced murine HNSCC model. **K, L** Representative images and quantitative analysis of densities of both sensory nerves (Trpv1$^+$) and Cav2$^+$ nerves during various stages of 4-NQO-induced murine tumorigenesis. Scale: 50μm. $n = 8$ mice per group. *$p = 0.0231$, **$p = 0.0057$(line 1 vs line 2), **$p = 0.0090$ (line 2 vs line 3), ****$p < 0.0001$, Student's t-test (two-sided). Data are mean ± s.d. **M** Proportion of Cav2$^+$ nerves among total sensory nerves. $n = 8$ mice per group. n.s: $p = 0.2575$; ****$p < 0.0001$, Student's t-test(two-sided). Data are mean ± s.d.

mice displayed a significant reduction in tumor volume compared to their Cav2$^{f/f}$ counterparts (Fig. 4H-I), confirming the contributory role of neuronal Cav2 in tumor development. However, the discrepancy in growth rates of tongue orthotopic implant tumors between mice with

neuron-specific Cav2 knockout and control mice was significantly less than that between mice with systemic Cav2 knockout and their controls. We found that although there was no significant difference in the incidence of PNI (Fig. 4J) or nerve count (Supplementary Fig. 4E)

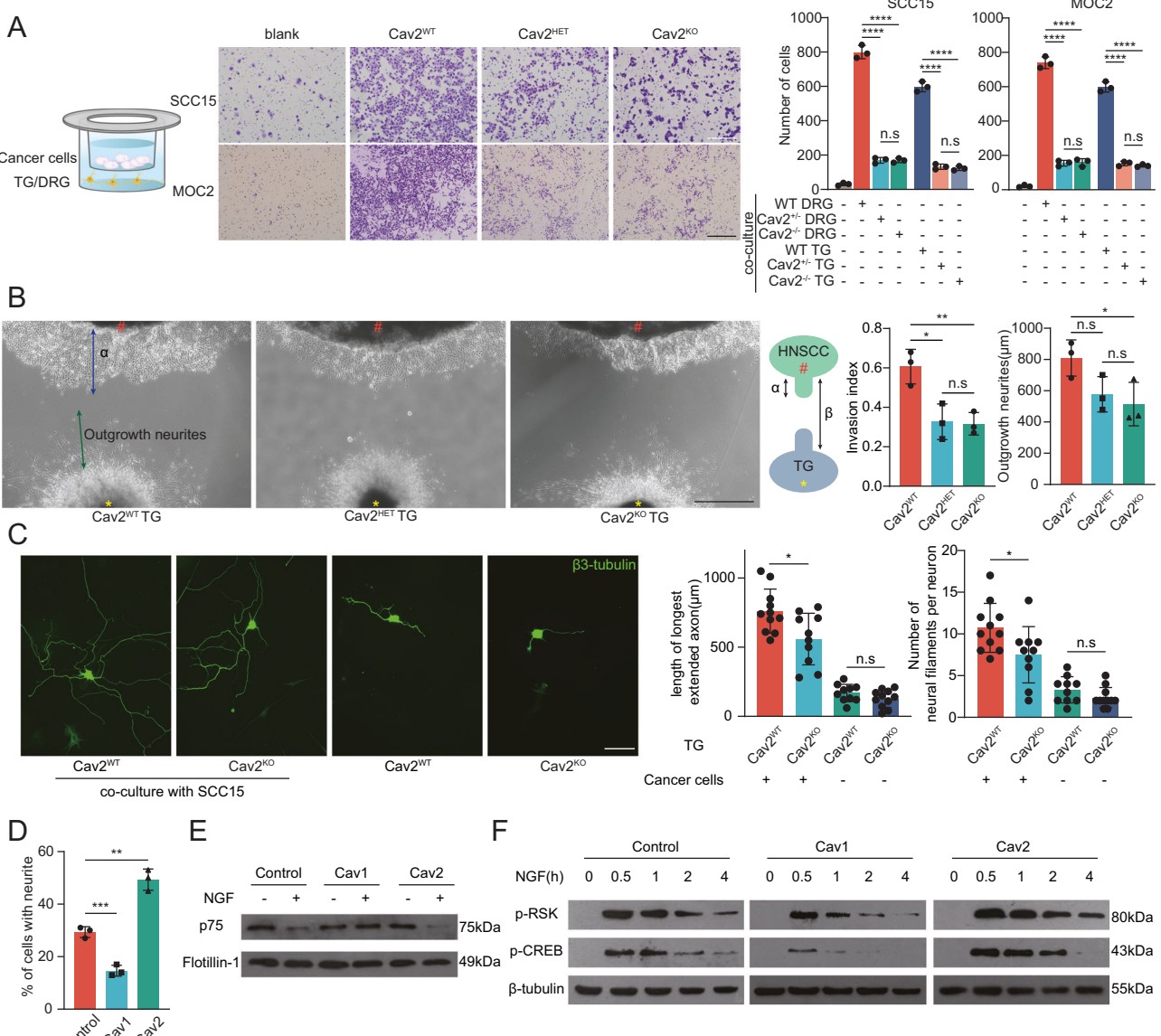

**Fig. 3 | Impact of Neural Cav2 Expression on Tumor Cell Infiltration Towards Nerves and Neurogenesis. A** Left: Diagrammatic representation of the transwell chamber assay used to assess the migration of human HNSCC SCC15 cells or murine MOC2 cells towards murine TGs or DRGs. Middle: Representative images display SCC15 or MOC2 cells being attracted by Cav2+/+ TGs, Cav2+/- TGs, or Cav2-/- TGs. Right: Statistical summary detailing the average number of HNSCC cells in each visual field. Scale: 200μm. $n = 3$ biologically independent co-cultures/group. ****$p < 0.0001$, Student's t-test (two-sided). Data are mean ± s.d. **B** Cancer cells were cultured in a Matrigel environment adjacent to TGs. Images present neural infiltration by SCC15 cells in association with TG from a heterozygous Cav2-deficient mouse (Cav2+/-), a homozygous deficient mouse (Cav2-/-), and their wild-type counterparts. Analysis encompassed nerve invasion index (α/β) and neurite length measurements. Scale: 750μm. $n = 3$ biologically independent co-cultures/group. Invasion index: *$p = 0.0181$, **$p = 0.0085$, Outgrowth neurites: *$p = 0.0476$. Student's t-test (two-sided). Data are mean ± s.d. **C** Fluorescence images of TG neurons post-co-culture with SCC15 cells or controls. The subsequent quantification

evaluates both neuritogenesis and the length of the most extended axons, observed 72 h post co-culture. Scale: 100μm. First and fourth groups: $n = 11$ biologically independent co-cultures; second and third groups: $n = 10$. Length of longest axon: *$p = 0.0150$, Number of neural filaments: *$p = 0.0301$. Student's t-test (two-sided). Data are mean ± s.d. **D** The percentage of PC12 cells, transfected with Cav1 or Cav2 plasmids, exhibiting neurite outgrowth of at least 10μm after 48 h of stimulation with 50 ng/mL NGF. $n = 3$ biologically independent cell cultures/group. **$p = 0.0016$, ***$p = 0.0010$, Student's t-test (two-sided). Data are mean ± s.d. **E** Western blot analysis of p75NTR levels in the lipid raft fraction before and after the addition of NGF (20 ng/mL for 1 h). Experiments were independently repeated three times. **F** Western blot analysis of the phosphorylation levels of RSK and CREB in control and Cav1 or Cav2-transfected PC12 cells under 20 ng/mL NGF stimulation at various time points. Experiments were independently repeated three times. TG trigeminal ganglion, DRG dorsal root ganglion, HNSCC head and neck squamous cell carcinoma, WT wild-type, HET heterozygous, KO knockout.

between the two groups, the β3-tubulin staining area within the tumor regions of Adv$^{Cre}$Cav2$^{f/f}$ mice was relatively smaller (Fig. 4K). Furthermore, the pS6 levels in tumors from Adv$^{Cre}$Cav2$^{f/f}$ mice were significantly lower (Fig. 4L).

Previous immunofluorescence results indicate the expression of Cav2 in glial cells of the TGs and DRGs. Consequently, we continue to

explore the impact of glial cell-derived Cav2 on tumor growth. Aldh1l1-CreERT2 mice are widely applied for conditionally modifying gene expression in astrocytes of the central nervous system[31]. Additionally, it has been found that in the peripheral nervous system's ganglia, most glial cells also express Aldh1l1[32]. We performed multicolor immunohistochemical staining on human HNSCC tissues and

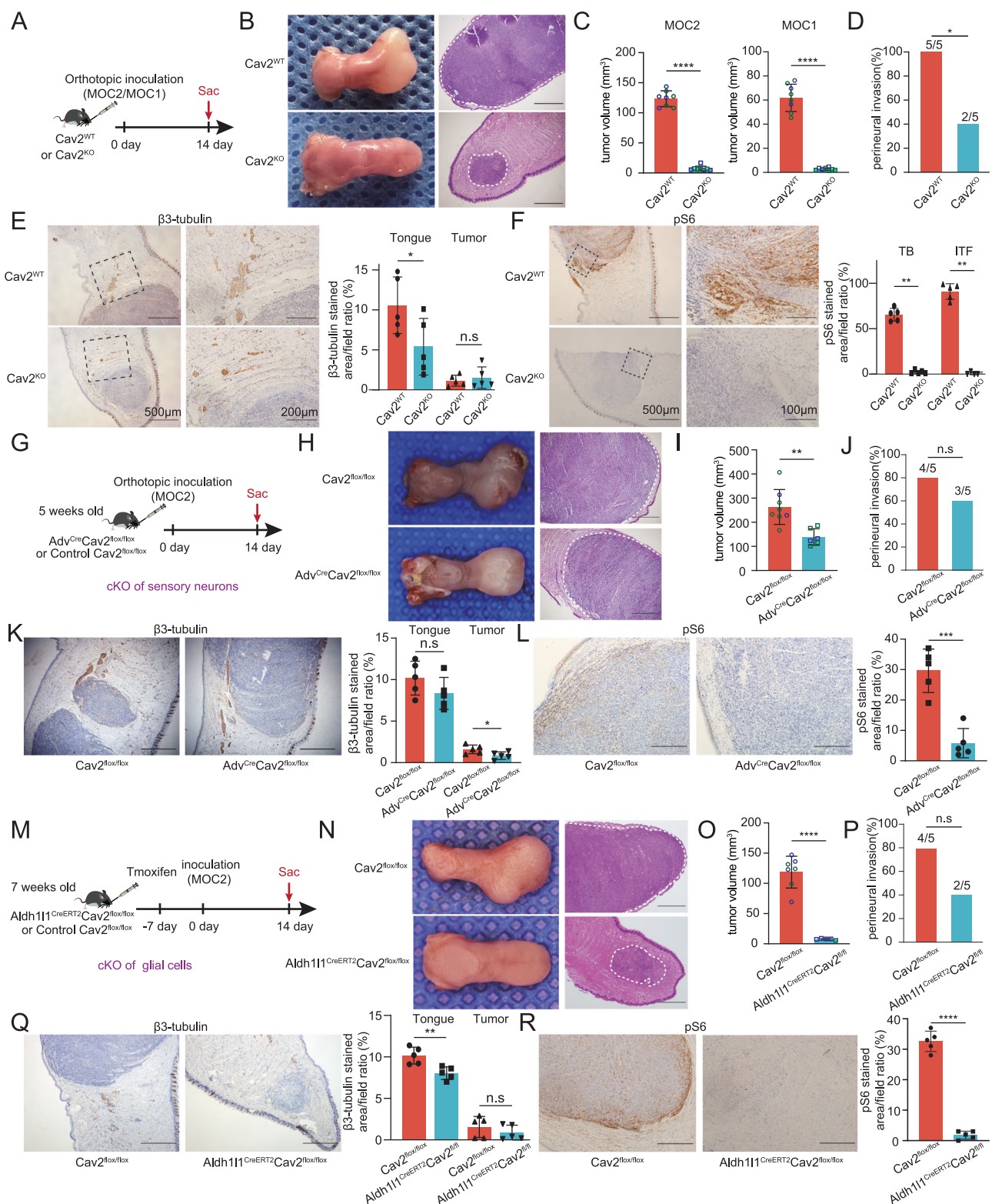

mouse tongue orthotopic tumor model tissues, and found colocalization of Aldh1l1 and S100 (a marker primarily expressed in Schwann cells), indicating that Aldh1l1 is expressed in glial cells of the tongue (Supplementary Fig. 5A, B). We crossed Aldh1l1-CreERT2 mice with Cav2$^{flox/flox}$ mice to generate Aldh1l1$^{CreERT2}$Cav2$^{f/f}$ mice (Fig. 4M), and confirmed by immunohistochemistry that Cav2 expression in nerves was significantly reduced in the tongue tissues of Aldh1l1$^{CreERT2}$Cav2$^{f/f}$ mice (Supplementary Fig. 5C). Surprisingly, we discovered that the growth of tongue orthotopic transplant tumors in these mice was

significantly inhibited, with the degree of inhibition nearly equivalent to the difference observed between systemic Cav2 knockout and control mice in terms of tumor growth in the tongue orthotopic transplant (Fig. 4N, O), indicating the more significant role of glial Cav2 in promoting tumor progression. The IHC results indicate that the neural distribution in the tongues of Aldh1l1$^{CreERT2}$Cav2$^{f/f}$ mice is sparser compared to the control group. However, the incidence of PNI between the two groups did not reach statistical significance (Fig. 4P, Q). The staining intensity of pS6 is

**Fig. 4 | Loss of Cav2 in Host or Neural Cav2 Deletion Slows Tumor Progression.**
**A** Schematic of orthotopic MOC2/MOC1 inoculation in Cav2$^{-/-}$ versus wild-type mice. **B** Representative tumor images (dotted outlines) on post-inoculation day 14. Scale: 500 μm. **C** Tumor volumes at day 14 post-inoculation. $n = 8$ mice per group. ****$p < 0.0001$, Student's t-test (two-sided). Data are mean ± s.d. **D** PNI incidence in both groups. $\chi^2$-test (two-sided), $n = 5$ mice per group. *$p = 0.0384$. **E** IHC for β3-tubulin in tongues of MOC2-inoculated mice, with quantification in the entire tongue and tumor. $n = 5$ per group. *$p = 0.0495$, Student's t-test (two-sided). Data are mean ± s.d. **F** IHC for pS6 in tumor bulk (TB) and invasive tumor front (ITF). $n = 5$ per group. TB: **$p = 0.0079$; ITF: **$p = 0.0079$, Two-sided Mann-Whitney U test. Data are mean ± s.d. **G** Schematic of orthotopic MOC2 inoculation inAdv$^{Cre}$ Cav2$^{flox/flox}$ and control Cav2$^{flox/flox}$ mice. **H** Representative tumor images at day 14 post-inoculation. Scale: 500 μm. **I** Tumor volumes at day 14 post-inoculation. $n = 6$ mice (Adv$^{Cre}$Cav2$^{flox/flox}$) and $n = 8$ (Cav2$^{flox/flox}$). **$p = 0.0022$, Student's t-test (two-sided). Data are mean ± s.d. **J** PNI incidence in both groups. $\chi^2$-test (two-sided). $n = 5$

mice per group. n.s: $p = 0.1515$. **K** IHC for β3-tubulin with quantification for entire tongue and tumor. Scale: 500μm. $n = 5$ per group. *$p = 0.0394$, Student's t-test (two-sided). Data are mean ± s.d. **L** IHC for pS6 in both groups. Scale: 200μm. $n = 5$ per group. ***$p = 0.0003$, Student's t-test (two-sided). Data are mean ± s.d. **M** Schematic of orthotopic MOC2 inoculation inAldh1l1$^{CreERT2}$Cav2$^{flox/flox}$ and control Cav2$^{flox/flox}$ mice (tamoxifen-treated). **N** Representative tumor images (dotted outlines) at day 14 post-inoculation. Scale: 500 μm. **O** Tumor volumes at day 14 post-inoculation. $n = 5$ mice (Aldh1l1$^{CreERT2}$Cav2$^{flox/flox}$) and $n = 7$ (Cav2$^{flox/flox}$). ****$p < 0.0001$, Student's t-test (two-sided). Data are mean ± s.d. **P** PNI incidence in both groups. $\chi^2$-test (two-sided). $n = 5$ mice per group. n.s: $p = 0.0578$. **Q** IHC for β3-tubulin with quantification in whole tongue and tumor. Scale: 500 μm. $n = 5$ per group. **$p = 0.0058$, Student's t-test (two-sided). Data are mean ± s.d. (R) IHC for pS6 in both genotypes. Scale: 200μm. $n = 5$ per group. ****$p < 0.0001$, Student's t-test (two-sided). Data are mean ± s.d. The blue circles represent male mice, and the green circles represent female mice. WT wild-type, KO knockout, cKO conditional knockout.

significantly weaker in Aldh1l1$^{CreERT2}$Cav2$^{f/f}$ mice compared to the control group (Fig. 4R).

## Cav2$^+$ Nerve-Driven Shift to Enhanced Mitochondrial Respiratory Phenotype in Cancer Cells

To discern the mechanistic interplay between neural CAV2 and HNSCC, we conducted RNA-sequencing on SCC15 cells co-cultured with either Cav2$^{+/+}$ or Cav2$^{-/-}$ TGs (Fig. 5A). The data showed that Cav2$^+$ nerves direct the upregulation of specific gene sets, notably including 'oxidative phosphorylation' and 'MYC targets' (Fig. 5B, C). Prompted by indications that Cav2$^+$ nerve interactions may precipitate an enhanced mitochondrial respiratory phenotype within cancer cells, we conducted an evaluation of the mitochondrial respiratory functions in HNSCC cells co-cultured with either Cav2-knockout or wild-type TGs. The results revealed a marked increase in both basal and maximal respiratory capacities in SCC15 cells co-cultured with wild-type TGs, compared to their Cav2 knockout counterparts (Fig. 5D). We also turned our attention to reactive oxygen species (ROS), known to increase following the activation of the mitochondrial electron transport chain. Notably, mitochondrial superoxide and hydrogen peroxide levels were elevated in HNSCC cells in the presence of Cav2$^{+/+}$ TGs compared to their Cav2$^{-/-}$ TG counterparts (Fig. 5E, F and Supplementary Fig. 6A, B). Concordantly, an upregulation of mitochondrial membrane potentials (ΔΨm) was observed in SCC15 cells co-cultured with wild-type TGs, underscoring a consistent enhancement of mitochondrial function (Fig. 5G and Supplementary Fig. 6C). Seeking insights into the mechanisms by which neural Cav2 influences mitochondrial oxidative phosphorylation (OXPHOS) in cancer cells, we explored the effect of Cav2$^{+/+}$ TGs on tumor cell mitochondrial biogenesis. Elevated mitochondrial counts in cells co-cultured with Cav2$^{+/+}$ TGs, as visualized by transmission electron microscopy, were found to be counteracted with the introduction of Cav2$^{-/-}$ TGs (Fig. 5H). Alongside, a marked enrichment in mitochondrial DNA content was observed in SCC15 and MOC2 cells co-cultured with Cav2$^{+/+}$ TGs compared to their Cav2$^{-/-}$ counterparts (Fig. 5I and Supplementary Fig. 6D). Given that Cav2 is expressed in neurons, glial cells, and vascular endothelial cells, we sought to explore the impact of Cav2 expression in specific cell types on the OXPHOS phenotype of tumor cells. First, after enzymatic digestion of Cav2$^{+/+}$ and Cav2$^{-/-}$ TGs, we used cytosine arabinoside to inhibit glial cell growth, thereby enriching for neurons, which were subsequently co-cultured with tumor cells. Results indicated that mitochondrial superoxide level in tumor cells co-cultured with Cav2$^{-/-}$ TG neurons was relatively lower compared to the control group ($p = 0.057$; Supplementary Fig. 6E). To assess the effect of glial Cav2 on tumor cell OXPHOS, we crossed mTmG mice with Adv-Cre mice to obtain mTmG$^{+/-}$; Adv-Cre mice, and then crossed these mice with Cav2$^{+/-}$ mice to generate mTmG$^{+/-}$; Adv-Cre; Cav2$^{+/-}$ mice (as both mTmG and Cav2 are located on chromosome 6, we could

only obtain heterozygous Cav2 mice). FACS sorting was used to isolate Adv-negative cells (predominantly glial cells), which were then co-cultured with tumor cells. It was found that mitochondrial superoxide level in tumor cells co-cultured with Cav2$^{+/-}$ non-neuronal cells was also reduced compared to the control group (Supplementary Fig. 6F). Additionally, to evaluate the role of endothelial Cav2, we co-cultured tumor cells with HUVECs (human umbilical vein endothelial cells) infected with either shCAV2 lentivirus or control lentivirus, and measured the mitochondrial superoxide level in tumor cells. The results showed no significant difference in mitochondrial superoxide level between tumor cells co-cultured with shCAV2 HUVECs and those with control HUVECs (data not shown), suggesting that the observed effects are specific to neural Cav2. Furthermore, we compared lactate secretion from tumor cells co-cultured with Cav2$^{+/+}$ and Cav2$^{-/-}$ TG. The results showed that tumor cells co-cultured with Cav2$^{-/-}$ TG exhibited relatively higher lactate secretion (Supplementary Fig. 6G).

To further evaluate the in vivo effects of Cav2 on the OXPHOS phenotype of HNSCC, we performed multicolor immunohistochemical staining to assess the expression of OXPHOS-related markers (SDHB, UQCRC, ATP5A1, NDUFB8) in tumors from the mouse models presented in Fig. 4. The overall expression of these markers was higher in orthotopic oral tumors from wild-type mice compared to Cav2$^{-/-}$ mice (Supplementary Fig. 7A). Similarly, expression levels in control mice were higher than in Aldh1l1$^{CreERT2}$Cav2$^{f/f}$ and Adv$^{Cre}$Cav2$^{f/f}$ mice (Supplementary Fig. 7B, C).

In a pivotal recent investigation by Zhang et al.[4], it was delineated that CGRP$^+$ nerves support HNSCC cell growth in glucose-deprived conditions, and the blockade of neurogenic CGRP amplifies the potency of anti-glycolysis treatments. The authors found that CGRP-expressing nerve-driven tumor growth is contingent on a 2-DG-induced low-glucose simulation. However, our observations indicate that Cav2$^{+/+}$ TG co-cultures foster an enhanced proliferation of cancer cells compared to Cav2$^{-/-}$ TG co-cultures, regardless of 2-DG conditions (Supplementary Fig. 8A, B). Furthermore, our analysis of the TCGA database reveals a significant negative association between CAV2 expression and patient prognosis, particularly pronounced in those with low HIF1A levels (Fig. 5J). This stands in contrast to Zhang's findings, where elevated CALCA (encodes CGRP) levels correlated with poorer survival, but solely in the cohort with high HIF1A expression. Additionally, our multicolor immunohistochemical analysis demonstrated distinct expression patterns of CGRP and CAV2 in nerves within the HNSCC microenvironment, revealing that some neurons express CGRP but lack CAV2, while others show the opposite expression pattern (Supplementary Fig. 8C), consistent with findings from single-nucleus RNA sequencing of TG (Supplementary Fig. 8D). This alludes to the intriguing possibility that Cav2-expressing nerves and CGRP-expressing nerves affect metabolic alterations within cancer cells via different mechanisms. Collectively, our results indicate that Cav2$^+$

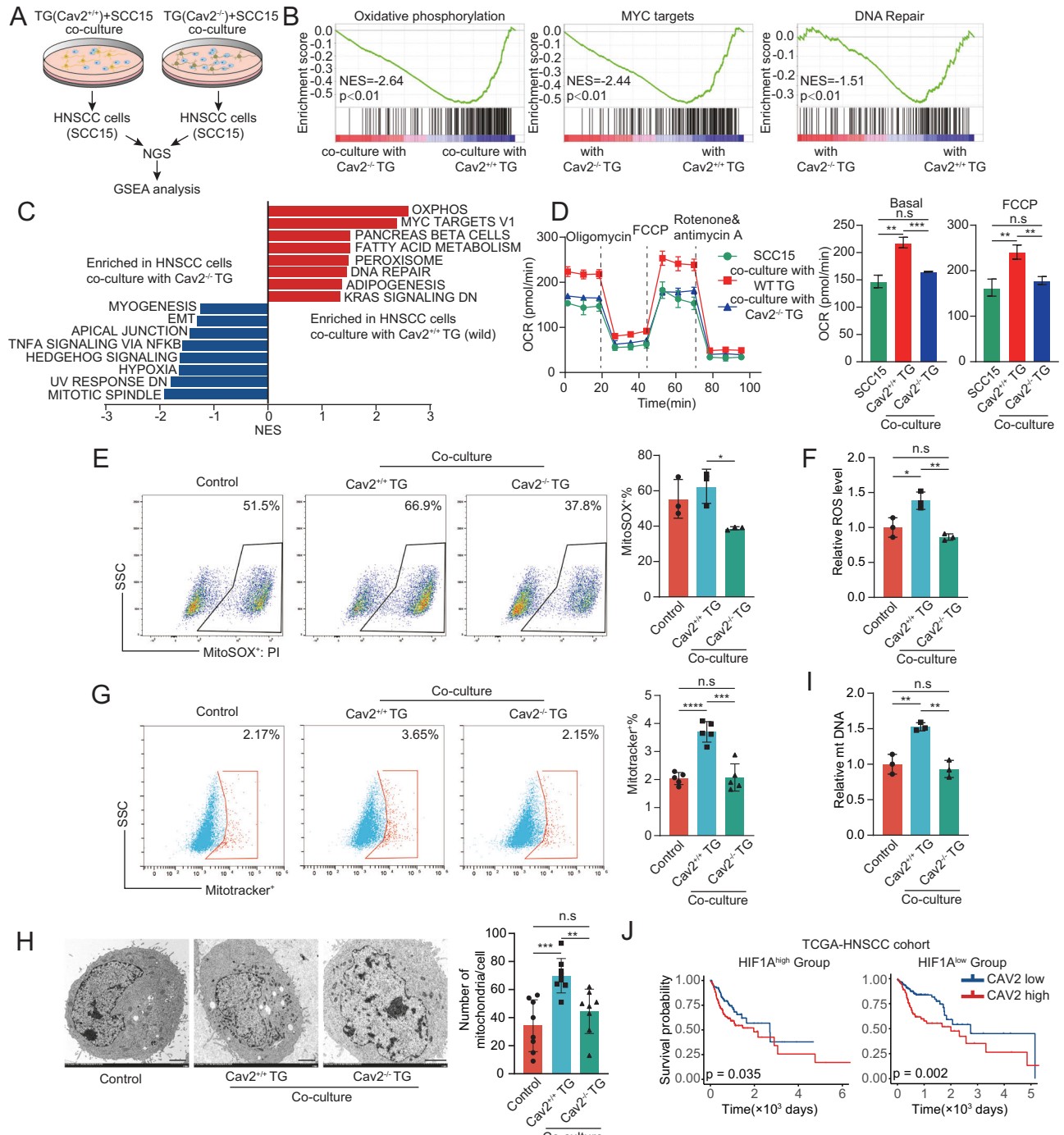

**Fig. 5 | Cav2⁺ Nerve-Driven Shift to Enhanced Mitochondrial Respiratory Phenotype in Cancer Cells. A** Schematic overview of RNA sequencing performed on SCC15 cells harvested post 72-h co-culture with Cav2⁻/⁻ or wild-type mouse TGs. (B, C) GSEA profiling reveals significantly upregulated pathways in SCC15 cells co-cultured with Cav2⁺/⁺ TGs over Cav2⁻/⁻ TGs. Two-sided permutation test. EMT: EPI-THELIAL MESENCHYMAL TRANSITION; OXPHOS: OXIDATIVE PHOSPHORYLA-TION. **D** Evaluation of OCRs in SCC15 cells post 72-h co-culture with or without Cav2⁺/⁺ TGs or Cav2⁻/⁻ TGs. FCCP, carbonyl cyanide-4-(trifluoromethoxy)phenylhy-drazone. $n = 3$ biologically independent co-cultures/group. Basal: **$p = 0.0012$, ***$p = 0.0007$, FCCP: line 1 vs line 2: $p = 0.0051$; line 2 vs line 3: $p = 0.0037$. Student's t-test (two-sided). Data are mean ± s.d. **E** MitoSOX Red staining and flow cytometry analysis of SCC15 cells after 72-h co-culture conditions. $n = 3$ biologically independent co-cultures/group. *$p = 0.0132$, Student's t-test (two-sided). Data are mean ± s.d. **F** ROS-Glo determination of ROS levels. $n = 3$ biologically independent co-cultures/group. *$p = 0.0236$, **$p = 0.0024$, Student's t-test. Data are mean ± s.d.

**G** SCC15 cells from different co-culture conditions stained with MitoTracker Red CMXRos and analyzed via flow cytometry. $n = 3$ biologically independent co-cultures/group. ***$p = 0.0003$, ****$p < 0.0001$, Student's t-test (two-sided). Data are mean ± s.d. **H** Transmission electron microscopy on SCC15 cells post co-culture conditions, with mitochondrial counts recorded. Scale: 2μm. $n = 8$ biologically independent co-cultures per group. **$p = 0.0030$, ***$p = 0.0005$, Student's t-test (two-sided). Data are mean ± s.d. **I** Mitochondrial DNA quantification in SCC15 cells using real-time qPCR and dedicated primer sets. $n = 3$ biologically independent co-cultures/group. line 1 vs line 2: $p = 0.0088$; line 2 vs line 3: $p = 0.0047$, Student's t-test (two-sided). Data are mean ± s.d. **J** Kaplan-Meier survival plots comparing overall survival (OS) in CAV2^high and CAV2^low subgroups within both HIF1A^high and HIF1A^low HNSCC TCGA datasets. Two-sided log-rank test. HIF1A^high group: $n = 250$ patients; HIF1A^low group: $n = 250$ patients. TG trigeminal ganglion, NES normalized enrichment score, OCR oxygen consumption rate, ROS reactive oxygen species.

nerves drive a transition to an enhanced mitochondrial respiratory character in cancer cells.

## Cav2-expressing nerves enhance tumor stemness through OXPHOS augmentation

We have demonstrated that Cav2[+] nerves precipitate an enhanced mitochondrial respiratory profile in HNSCC cells. While the traditional view holds that cancer cells are reliant on aerobic glycolysis, recent insights suggest a more complex landscape with CSCs exhibiting a reliance on OXPHOS[20,33]. However, reliance on oxidative phosphorylation is not a consistent feature across CSCs of all cancer types[34]. The metabolic predispositions of HNSCC CSCs are, to date, not fully understood. Our data from SCC15 cells show that CSCs, isolated based on ALDH1A1 activity, manifest an increase in both basal and maximal respiratory rates, indicating upregulated mitochondrial OXPHOS (Fig. 6A and Supplementary Fig. 9A). This is further corroborated by increased mitochondrial superoxide levels and a concomitant rise in mitochondrial membrane potentials in these cells (Fig. 6B, C), underscoring the enhanced mitochondrial respiratory function inherent to HNSCC CSCs.

Our hypothesis suggests that Cav2[+] nerves may facilitate a metabolic transition toward oxidative phosphorylation in tumor cells, thus conferring stemness characteristics. Remarkably, the gene expression profiles of HNSCC cells cultured with Cav2[+/+] nerves compared to those cultured with Cav2[-/-] nerves (shown in Fig. 5A–C) closely resemble the differential patterns noted between ALDH[+] and ALDH[-] cell populations[35], including the upregulation of OXPHOS and MYC target gene signatures that characterize ALDH1[+] breast cancer CSCs (Fig. 5A, B of ref. 35). Gene set enrichment analysis (GSEA) also revealed that stemness gene signatures are significantly enriched in SCC15 cells co-cultured with Cav2[+/+] nerves versus those with Cav2-deficient TGs (Fig. 6D). Additionally, TCGA data analysis indicates that patients exhibiting high CAV2 expression are associated with enriched Epithelial-Mesenchymal Transition (EMT) gene signatures, a process intimately linked to stemness (Supplementary Fig. 9B). Subsequent histological evaluations of human HNSCC specimens revealed a tendency for an increased presence of CD44[+] cancer cells in tumor zones in close proximity to neural niches with high CAV2 expression (Fig. 6E). In vitro assays further confirm that Cav2[+/+] nerve co-culture augments the proportion of stem cell-like cells in both SCC15 and MOC2 cell lines (Fig. 6F, G). Consequently, this cellular interaction results in a marked increase in both the size and quantity of spheroids formed by HNSCC cells in comparison to those co-cultured with Cav2[-/-] TGs (Fig. 6H). Our continued research on the influence of Cav2[+] TGs on HNSCC cell resistance to platinum-based chemotherapeutics revealed that SCC15 cells co-cultured with Cav2[+/+] TGs displayed reduced apoptosis following cisplatin treatment compared to SCC15 cells co-cultured with Cav2[-/-] TGs. Additionally, the growth capability was significantly sustained in the cells co-cultured with Cav2[+/+] TGs relative to those with Cav2[-/-] TGs (Fig. 6I).

To evaluate the in vivo effects of Cav2[+/+] nerves on tumor cell behavior, we employed an orthotopically co-injection model. Post-digestion Cav2[+/+] and Cav2[-/-] TGs were co-injected with MOC2 cells into murine tongues at a 1:1 ratio. As expected, the presence of wild-type TGs significantly bolstered tumor formation, a phenomenon abrogated in Cav2[-/-] TGs (Fig. 6J). At a sufficient inoculum density ($2\times10^3$ or $5\times10^3$ cells), the dependency on co-injected nerves for tumor formation was overshadowed. Additionally, we utilized multicolor immunohistochemical analysis to probe the expression of HNSCC stemness markers (Aldh1a1, CD44, Bmi1) in orthotopic tongue xenografts derived from wild-type and Cav2[-/-] mice, as depicted in Fig. 4. The evaluation revealed that the proportion of stemness marker-positive cells in the xenografts from wild-type mice was significantly higher compared to those from Cav2[-/-] mice. Furthermore, an interesting observation was made regarding the expression of these stemness markers in the normal epithelial tissue of wild-type mice, which was also higher than in the Cav2[-/-] mice (Fig. 6K). Further analysis using the same staining approach in conditionally knocked-out mice (Adv[Cre]Cav2[f/f] and Aldh1l1[CreERT2]Cav2[f/f]) showed a substantially reduced prevalence of marker-positive cells in these models, especially in the Aldh1l1[CreERT2]Cav2[f/f] xenografts (Supplementary Fig. 10). Collectively, these findings suggest that Cav2[+/+] nerves are facilitators of tumor cell stemness, likely through an enhancement of oxidative phosphorylation.

## Depletion of Cav2 inhibits 4-NQO-induced oral tumorigenesis

To elucidate the influence of host Cav2 on HNSCC initiation, we employed the 4-NQO-induced HNSCC mouse model, a 16-week exposure to 4-NQO culminates in the emergence of hyperplasia and dysplasia, progressing to invasive squamous cell carcinoma by weeks 20-25[23]. Pathological assessments at 23 weeks showed invasive carcinoma in 9 out of 10 wild-type mice, contrasting with its presence in just 1 out of 10 Cav2[-/-] counterparts (Fig. 7A–C). To delineate the neuronal Cav2's contribution to HNSCC initiation, we euthanized Adv[Cre]Cav2[f/f] and control Cav2[f/f] mice at the 23-week post-induction. Notably, the incidence of invasive cancer was diminished in the Adv[Cre]Cav2[f/f] group compared to the control (Fig. 7D–F). To further explore the role of glial Cav2 in HNSCC initiation, under the same conditions, we studied the tumorigenesis in Aldh1l1[CreERT2]Cav2[f/f] mice induced by 4-NQO. The results indicated that only 1/10 of these mice developed invasive cancer, highlighting the significance of glial cell-derived Cav2 in the initiation of HNSCC (Fig. 7G–I).

To comprehensively elucidate Cav2's role throughout the tumor development trajectory, we initiated a separate set of experiments. At the 30-week mark following the onset of 4-NQO treatment, the wild-type and Cav2[-/-] mice exhibited apparent tongue lesions (Fig. 7J, K). Notably, we observed a significant reduction in both the count and size of lesions in Cav2[-/-] mice (Fig. 7L, M). Through histological assessment, we identified a significant reduction in both HNSCC numbers and areas following Cav2 depletion (Fig. 7N, O). Moreover, the invasive depth of Cav2[-/-] mice was decreased significantly (Fig. 7P). To elucidate the consequences of Cav2 ablation on metastasis, cervical lymph nodes underwent detailed microscopic analysis. This revealed a decline in the number of metastatic lymph nodes and in the metastatic rate per mouse for Cav2[-/-] mice (Fig. 7Q). Upon staining for the neuronal marker β3-tubulin, Cav2[-/-] mice displayed a reduced presence of neural filaments compared to control wild mice (Fig. 7R). Additionally, the incidence of PNI was significantly higher in CAV2[+/+] mice compared to CAV2[-/-] mice (Fig. 7S). Likewise, epithelial pS6 stain scores were higher in wild-type mice relative to Cav2[-/-] counterparts (Fig. 7T), however, the disparity is not as pronounced as the difference observed in xenograft experiments between wild-type and Cav2[-/-] mice (shown in Fig. 4E). Additionally, we conducted multicolor immunohistochemical staining targeting common HNSCC stemness markers, such as Aldh1a1, CD44, and Bmi1. The evaluation revealed that the proportion of stemness marker-positive cells in 4-NQO-induced tumors from wild-type mice was significantly higher than in those from Cav2[-/-] mice (Fig. 7U). Collectively, we found that the systemic knockout of Cav2 or its conditional knockout in either neurons or glial cells significantly inhibits tumorigenesis of HNSCC.

## Neural Cav2's role in the expression of extracellular proteins in ganglia

To elucidate the mechanism by which neural Cav2 influences tumor cell phenotype, we first analyzed the subcellular localization of Cav2 in neural cells. Previous studies have reported that Cav2 can localize to various subcellular compartments, including lipid rafts, the Golgi apparatus, the endoplasmic reticulum (ER), and the nucleus[36,37]. Our immunofluorescence images revealed a prominent colocalization of Cav2 with GM130, a Golgi marker, in both neuronal and non-neuronal

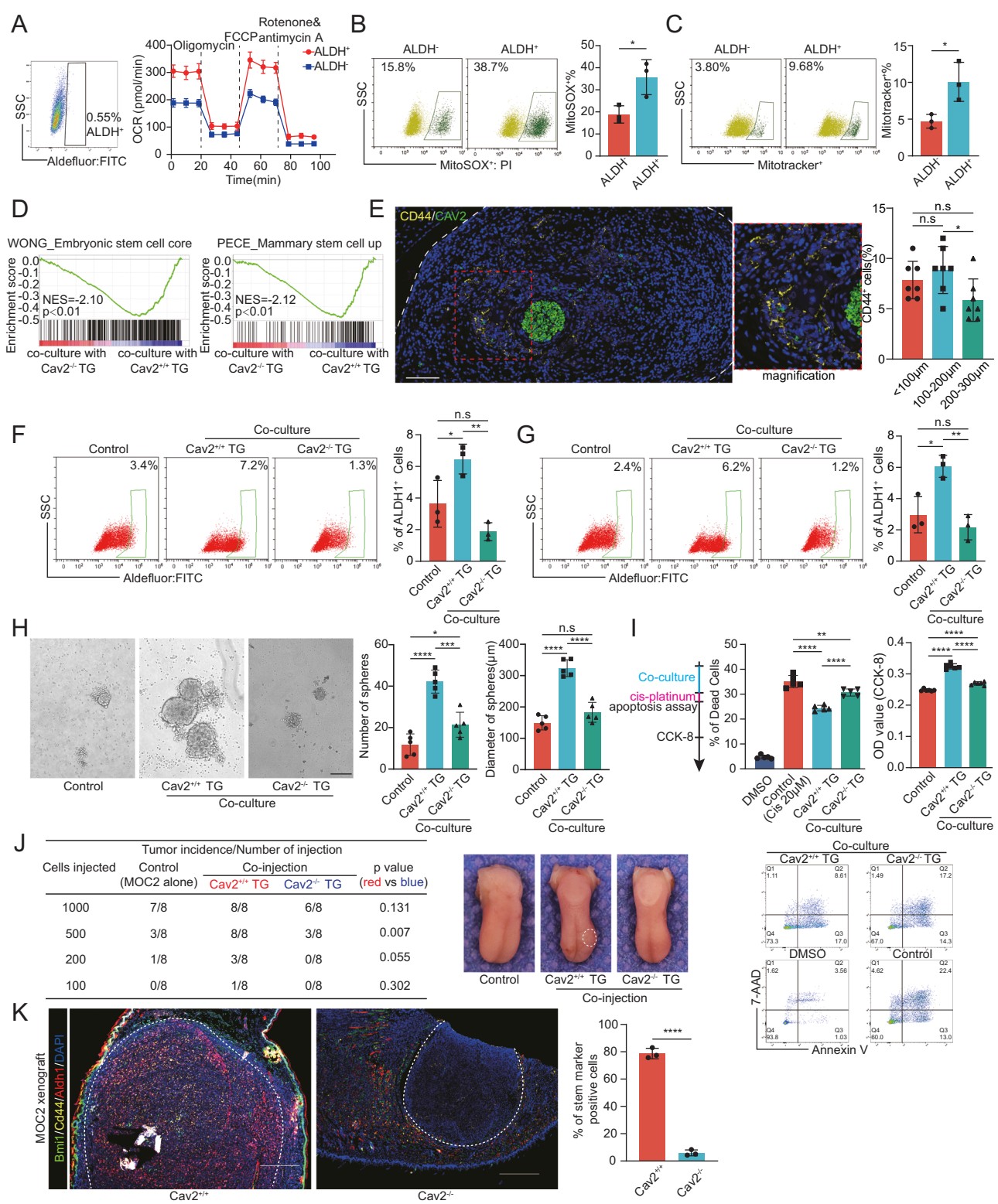

cells of the TG (Fig. 8A, B). Moreover, upon co-culture with tumor cells, we observed that Cav2 redistributed along the axons in a subset of neurons, while in most neurons Cav2 remained primarily localized to the Golgi apparatus (Fig. 8C). These findings suggest that neural Cav2 may be involved in protein modification, processing, and secretion.

Next, we co-cultured SCC15 (HNSCC cells) with trigeminal ganglia or cultured trigeminal ganglia alone for 48 h, followed by removal of tumor cells and a 6-h culture of ganglia alone in fresh medium. We then measured the protein concentration in the conditioned medium. Results showed that protein secretion from ganglia was significantly elevated following co-culture with tumor cells (Fig. 8D). Subsequently, we compared the protein concentrations in the conditioned media of wild-type (CAV2[+/+]) and Cav2-deficient (CAV2[-/-]) TG, following the same co-culture protocol. Results indicated a significantly reduced protein concentration in the medium of Cav2-deficient ganglia (Fig. 8E).

To further investigate which proteins were specifically affected by neural Cav2 in terms of processing and secretion, we performed

**Fig. 6 | Cav2⁺ Nerves Enhance Tumor Stemness Through Mitochondrial OXPHOS Augmentation. A** ALDH1A1 enzymatic activity assessed via the ALDE-FLUOR assay in SCC15 cells. ALDH1⁺ cells, isolated by flow cytometry, were subsequently evaluated for their OCRs. $n = 3$ biologically independent experiments per group. Data are mean ± s.d. **B** Flow cytometry of MitoSOX Red-stained SCC15 cells sorted as in (**A**). $n = 3$ biologically independent experiments per group. *$p = 0.0295$, Student's t-test (two-sided). Data are mean ± s.d. **C** MitoTracker Red CMXRos staining followed by flow cytometric analysis. $n = 3$ biologically independent experiments per group. *$p = 0.0302$, Student's t-test. Data are mean ± s.d. **D** GSEA reveals stemness gene set enrichment in SCC15 cells co-cultured with Cav2⁺/⁺ TGs over Cav2⁻/⁻ TGs. Two-sided permutation test. **E** Representative images and quantification of CD44⁺ cells in human HNSCC tissues ($n = 7$ niches from different patients). Scale: 100μm. *$p = 0.0271$, Student's t-test (two-sided). Data are mean ± s.d. (**F-G**) The percentage of ALDH1⁺ cells of SCC15 (**F**) and MOC2 (**G**) cells post specified co-culture conditions. $n = 3$ biologically independent co-cultures/group. **F**: *$p = 0.0487$, **$p = 0.0019$, **G**: *$p = 0.0168$, **$p = 0.0033$, Student's t-test (two-sided). Data are mean ± s.d. **H** Representative tumorspheres from SCC15 cells alone or co-cultured with Cav2⁺/⁺ or Cav2⁻/⁻ TGs, with quantification of number and diameter. Scale: 200μm. $n = 5$ biologically independent co-cultures/group. *$p = 0.0269$, ***$p = 0.0005$, ****$p < 0.0001$, Student's t-test (two-sided). Data are mean ± s.d. **I** Cells treated with cisplatin, followed by flow cytometry for apoptotic cells. Four days later, cell viability was measured (CCK-8 assay). $n = 5$ biologically independent experiments per group. **$p = 0.0086$, ****$p < 0.0001$, Student's t-test (two-sided). Data are mean ± s.d. **J** MOC2 cells were orthotopically injected into C57BL/6 mice either alone or combined with Cav2⁺/⁺ or Cav2⁻/⁻ TGs at a 1:1 volume ratio ($n = 8$ per group). Tumor formation was evaluated after two weeks, with the incidence rates being subsequently reported. χ²-test (two-sided). **K** Representative multicolor immunohistochemical staining of Aldh1a1, CD44, and Bmi1 in the tongues of MOC2-engrafted wild-type versus Cav2 knockout mice. Cells positive for any one of these markers are defined as marker-positive cells. Scale: 500μm. $n = 3$ mice per group. ****$p < 0.0001$, Student's t-test (two-sided). Data are mean ± s.d. OCR oxygen consumption rate, TG trigeminal ganglion, NES normalized enrichment score.

quantitative proteomic mass spectrometry on the conditioned media collected after co-culturing wild-type or Cav2-deficient trigeminal ganglia with tumor cells for 48 h, followed by 6 h of solo culture. Given the potential influence of these extracellular proteins on tumor cells, we focused our analysis on the proteins present in the conditioned medium. A total of 225 differentially expressed proteins were identified, with 158 proteins downregulated and 67 upregulated in the Cav2-deficient group (Fig. 8F). Interestingly, these differentially expressed proteins were primarily involved in neuronal growth and development, cellular metabolism, and protein processing and secretion, along with several proteins whose functions remain poorly characterized.

Focusing on cellular metabolism, particularly oxidative phosphorylation (OXPHOS), we found several secreted proteins that may influence mitochondrial metabolism. These included CYB5A (cytochrome b5) and CYB5R3 (cytochrome b5 reductase), both involved in electron transfer and potentially linked to OXPHOS; ATP5PB, a subunit of ATP synthase, which is a key component of the mitochondrial ATP synthase complex; ETFDH (electron transfer flavoprotein dehydrogenase), involved in mitochondrial fatty acid oxidation and OXPHOS; MICU3, a mitochondrial calcium regulatory protein that controls calcium homeostasis, potentially impacting mitochondrial OXPHOS; MIGA2, a mitochondrial intermembrane assembly regulator protein, which regulates mitochondrial dynamics and is related to OXPHOS; TRIAP1, a mitochondrial inner membrane protein involved in mitochondrial protein folding and oxidative stress response, indirectly participating in OXPHOS; and VPS13D, implicated in mitochondrial quality control and indirectly affecting OXPHOS. Additionally, the differentially expressed proteins included numerous other metabolic proteins involved in glycolysis, lipid metabolism, and amino acid metabolism, processes that also impact OXPHOS levels. Thus, neural Cav2 appears to regulate OXPHOS by orchestrating the secretion of a wide array of metabolism-related proteins rather than simply affecting a single cytokine or signaling pathway in tumor cells. In summary, neural Cav2 is primarily localized to the Golgi apparatus and influences the extracellular secretion of a variety of proteins involved in neuronal growth and development, cellular metabolism, and protein processing and secretion.

## Discussion

Here, we elucidate the critical involvement of neural Cav2 in the interaction between nerves and cancer cells in HNSCC. Our findings indicate that cancer cells trigger an upregulation of Cav2 in both neuronal cell bodies within ganglia and nerve fibers within the tumor microenvironment. This overexpression of Cav2 in nerves, in turn, fosters stemness characteristics in tumor cells, primarily by inducing a metabolic shift towards mitochondrial OXPHOS. These insights uncover a previously unrecognized pathway through which neural elements enhance the stem-like properties of cancer cells.

The PNS, encompassing varied neuronal and glial cell populations, innervates organs and tissues throughout the body, notably in the head and neck areas like the tongue and oral mucosa. Research has consistently highlighted the significance of nerve-cancer interactions and intratumoral neural infiltration in the progression of cancers. These interactions are not only pivotal in disease advancement but also significantly impact patient prognosis[2,3,38]. PNI, a phenomenon where cancer cells either surround or invade nerve tracts, stands out as the most evident form of nerve-cancer interplay. This is particularly pronounced in tumors with dense innervation, such as HNSCC, pancreatic cancer, and prostate cancer[24,39]. The expanding focus on tumor hyperinnervation and nerve-cancer interactions has catalyzed the development of cancer neuroscience, a field that extends well beyond the traditional boundaries of PNI[3]. For instance, HNSCC patients exhibiting close nerve-tumor proximity and larger nerve sizes have been found to have poorer outcomes, even when PNI is negative using current criteria[40]. Recent studies have shed light on the role of tumoral neurotrophic factors and axon guidance molecules in facilitating nerve-cancer interactions (reviewed by ref. 41). However, the molecular alterations of neuro-glial cells have not been extensively studied. Some explorations indicate that nerve-derived elements like neurotransmitters and neuropeptides could play significant roles in modulating anti-tumor immunity and influencing tumor growth.[15,42,43]

Caveolins are the major proteins of caveolae, submicroscopic bulb-shaped plasma membrane pits[10]. Our previous research showed that tumoral CAV2 facilitates the invasion and metastasis of HNSCC through the regulation of ubiquitylation and subsequent degradation of S100 proteins[44]. In clinical cases, however, CAV2 expression did not show a significant correlation with T stage, partially determined by the depth of invasion, or N stage. In contrast, CAV2 expression was markedly elevated in PNI positive cases, and CAV2 predominantly localized to nerve structures within the tumor microenvironment. These observations led to an exploration of the role of neural CAV2. Note that both Cav1 and Cav2 have been identified in differentiating PC12 cells and DRG neurons[45]. Previous studies have implicated Cav1 in a paracrine antiapoptotic loop in prostate cancer PNI[46]. However, despite sharing a common lineage, Cav1 and Cav2 display a low degree of amino acid sequence homology (38% identity)[29,47], suggesting potential distinct functional roles. The findings presented herein indicate that host Cav2 does not significantly impact apoptosis in HNSCC.

To investigate the impact of Cav2 expression in glial cells on HNSCC, we generated Aldh1l1^CreERT2^Cav2^f/f^ mice. Currently, there are no glial cell-specific Cre mouse models for the TG and DRG. Aldh1l1, a known marker of astrocytes in the central nervous system (CNS), has

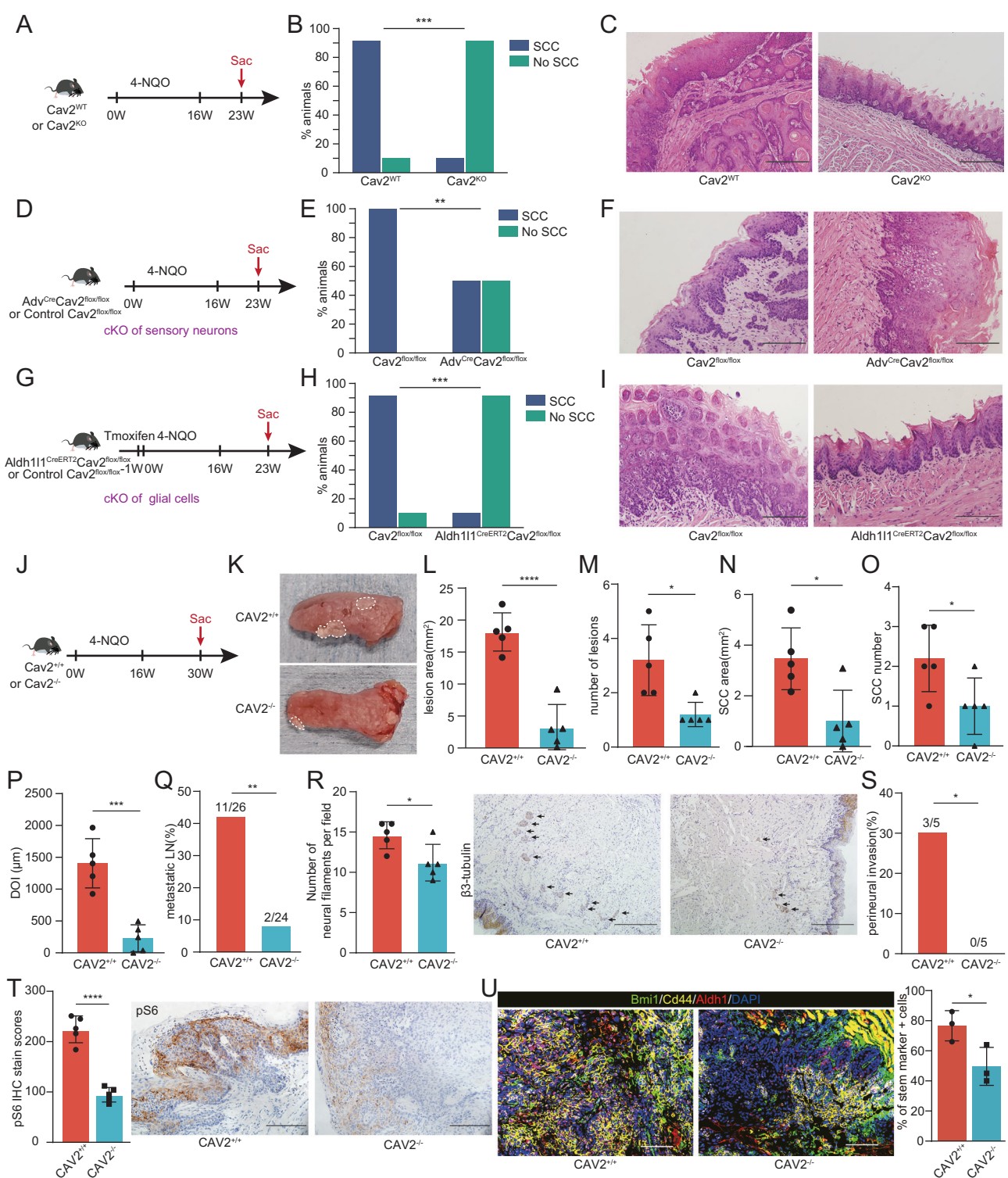

been reported to be expressed in glial cells of peripheral ganglia[32]. Therefore, we employed this model for our study. However, it is important to note that while we observed Aldh1l1 expression in Schwann cells within the tongue tissue, the specificity of this model is limited. Specifically, Aldh1l1[CreERT2]Cav2[f/f] mice also resulted in the deletion of Cav2 in astrocytes of the CNS. This represents a limitation in our study, and we look forward to the development of more specific Cre mouse models targeting glial cells in the TG and DRG in future research.

Sequencing analysis from our study suggests that Cav2[+/+] nerves may influence the metabolic processes in HNSCC. A recent study

reported that HNSCC cells exploit CGRP-positive nociceptive nerves to enhance cell growth in low-glucose environments, and targeting nociceptive nerves enhances anti-glycolysis therapies[4]. However, our data suggests that the influence of neurogenic Cav2 on HNSCC cell growth is independent of low-glucose conditions simulated by 2-DG exposure. Consequently, we propose that Cav2-positive nerves may contribute to tumorigenesis through alternative mechanisms. The role of a nerve-expressed protein from the Caveolin family in influencing tumor cell metabolism and stem-like phenotypes is not immediately intuitive. Caveolins are primarily found in lipid rafts, where they form caveolae. In addition to lipid rafts, Cav2 has also been detected in the

**Fig. 7 | Cav2 deficiency inhibits 4-NQO-induced HNSCC tumorigenesis. A** Study design: 4-NQO water administration for 16 weeks; euthanasia initiated from week 23 post-exposure. **B** Comparative SCC incidence in Cav2$^{-/-}$ and its wild counterpart. $n = 10$ mice per group. ***$p = 0.0003$, $\chi^2$-test (two-sided). **C** Representative H&E sections of tongue tumors from 4-NQO-treated mice. All mice ($n = 10$ per group) underwent histological analysis, and representative images are shown. Scale: 200 μm. **D–F** Study design and results for Adv$^{Cre}$Cav2$^{flox/flox}$ versus Cav2$^{flox/flox}$ mice with 4-NQO treatment. Scale: 100μm. $n = 10$ mice per group. **$p = 0.0098$, $\chi^2$-test (two-sided). **G–I** Study design and results for Aldh1l1$^{CreERT2}$Cav2$^{flox/flox}$ versus Cav2$^{flox/flox}$ mice post 4-NQO administration. Scale: 100μm. $n = 10$ mice per group. ***$p = 0.0003$, $\chi^2$-test (two-sided). **J** Study design: 16-week 4-NQO exposure; euthanasia from week 30. **K** Tongue lesion visuals, 30 weeks post 4-NQO. (**L-O**) Tongue lesion area, HNSCC area, lesion count and HNSCC count quantifications. $n = 5$ mice per group. **L:** ****$p < 0.0001$, **M:** *$p = 0.0118$, **N:** *$p = 0.0129$, **O:** *$p = 0.0400$, Student's t-test (two-sided). Data are mean ± s.d. **P** Quantification of HNSCC invasion depths. DOI = Depth of Invasion. $n = 5$ mice per group. ***$p = 0.0003$, Student's t-test (two-sided). Data are mean ± s.d. **Q** Number of cervical lymph nodes with metastasis in Cav2$^{-/-}$ versus control. $\chi^2$-test (two-sided). **$p = 0.0062$. **R** Neural filament counts per view, β3-tubulin-marked, with representative images. Scale: 200μm. $n = 5$ mice per group. *$p = 0.0276$, Student's t-test (two-sided). Data are mean ± s.d. **S** Comparative PNI incidence in Cav2$^{-/-}$ versus control. $n = 5$ mice per group. *$p = 0.0384$. $\chi^2$-test (two-sided). **T** Epithelial pS6 staining quantification, complemented by representative images. Intensity: 0 (negative) to 3 (strongly positive). The final score resulted from multiplying intensity by the percentage of positive staining. Scale: 100μm. $n = 5$ mice per group. ****$p < 0.0001$, Student's t-test (two-sided). Data are mean ± s.d. **U** Representative multicolor immunohistochemical staining of Aldh1a1, Cd44, and Bmi1 in the 4-NQO-induced tumors. Cells positive for any one of these markers are defined as marker-positive cells. Scale: 100 μm. $n = 3$ mice per group. *$p = 0.0445$, Student's t-test (two-sided). Data are mean ± s.d.

Golgi apparatus, endoplasmic reticulum (ER), and nucleus, while other Caveolins have been also observed in mitochondria and other subcellular locations[36,37]. Our subcellular localization imaging revealed that Cav2 is mainly expressed in the Golgi apparatus and its periphery in ganglion cells, which include both neuronal and non-neuronal cells. Following co-culture with tumor cells, Cav2 was found to localize to the axons and axonal surface in a subset of cells. A previous study has shown that CAV2 can migrate from the Golgi apparatus to the cell surface, a process that depends on CAV1[48]. These findings suggest that Cav2 might be involved in processes such as protein processing and secretion. Notably, spatial transcriptomic studies have shown that nerves in close proximity to tumors exhibit a significant upregulation of ribosome-associated proteins, indicating that tumor cells may stimulate neural cells to actively express specific proteins[40]. Our mass spectrometry analysis further revealed significant differences in protein expression between conditioned media from Cav2$^{+/+}$ and Cav2$^{-/-}$ trigeminal ganglia co-cultured with tumor cells, particularly proteins related to metabolism, neural growth and development, protein processing, and vesicle trafficking. This suggests that neural Cav2 does not merely regulate the secretion of individual cytokines to influence tumor cell pathways but rather orchestrates and coordinates the secretion of multiple proteins to mediate neuron-tumor crosstalk. Notably, we identified several mitochondrial proteins in the conditioned medium, even though these proteins are not typically classified as secreted. Consequently, further research is warranted to elucidate the mechanisms underlying their presence in the conditioned medium and to determine the extent to which they may be internalized by tumor cells.

The longstanding belief that cancer cells predominantly depend on aerobic glycolysis, known as the Warburg effect[49], has been recently reconsidered, with growing evidence supporting the role of mitochondrial OXPHOS in cancer progression[50]. Particularly, CSCs within tumors, known for their ability to repopulate tumors and drive recurrence post-adjuvant therapy, exhibit increased reliance on OXPHOS[34]. Recent gene expression analysis in MDA-MB-468 derived ALDH1$^+$ and ALDH1$^-$ cells, along with GSEA, identified strong associations of oxidative phosphorylation and c-Myc targets in ALDH1$^+$ cells[35], resembling the profiles of HNSCC cells co-cultured with Cav2$^{+/+}$ nerves in our study. Another independent study also showed breast CSCs exhibit enhanced mitochondrial respiration[18]. However, the metabolic profiles of CSCs vary across cancer types; for instance, high-OXPHOS ovarian cancers show greater chemosensitivity, contrasting the typical chemotherapy resistance of CSCs[51]. The metabolic characteristics of HNSCC CSCs remain elusive, leading us to uncover their metabolic traits. Notably, ALDH$^+$ HNSCC cells exhibit enhanced mitochondrial respiration and OXPHOS. Based on this, a link between Cav2-expressing nerves and cancer cell stemness was established. An increasing amount of evidence suggests that in the tumor microenvironment, various components such as cancer-associated fibroblasts and immune cells, participate in maintaining the stemness of tumors[52,53]. Notably, several studies have implicated the nervous system in tumor initiation and the maintenance of cancer cell stemness. For instance, the activation of autonomic nerve networks has been shown to influence stem-like properties in breast, prostate, and pancreatic cancers[14,54,55]. Furthermore, it has been observed that cutaneous nerves can promote the initiation of basal cell carcinoma, with tumors preferentially arising from stem cells in mechanosensory niches[56]. Recent findings also reveal that the neural signal norepinephrine enhances the stemness of proximal cancer cells through the cAMP-CRE axis[57]. Despite these indications that the nervous system may be involved in the stemness of cancer cells and tumor initiation, the underlying regulatory mechanisms remain largely elusive, significantly hindering the development of novel therapeutic approaches. Our study confirms that neurogenic Cav2 is a critical molecular pathway through which the tumor microenvironment influences tumor stemness, offering potential targets and strategies for cancer treatment.

Currently, although direct pharmacological targeting of Caveolins has not been widely reported, emerging nanotechnologies have shown significant potential in cancer therapy. Specifically, some nanomaterials have been developed to selectively deliver kinase inhibitors to tumor cells expressing Caveolin-1[58]. Moreover, research has indicated that certain nanomaterials can effectively cross the blood-brain barrier through Caveolin-1-mediated transcytosis[59]. These achievements suggest that further modifications of these nanomaterials, such as loading them with cytotoxic drugs, could enable precise targeting of cells expressing Caveolin-1, thereby offering new strategies for treating related diseases. However, it is important to note that CAV1 and CAV2 differ significantly in their intracellular distribution and functions. Materials or drugs that target CAV2 are rarely reported, which may be due to the limited research on CAV2 compared to CAV1, thus its significant roles have not yet garnered attention. It is hoped that strategies targeting CAV2 will be developed in the future.

## Methods

### Study population

This study utilized tumor specimens from patients treated within the Department of Head and Neck Surgical Oncology at Tianjin Medical University Cancer Institute & Hospital (TJCH cohort), Tianjin, China, from 2010 to 2015. A total of 101 patients were included in this study, comprising 73 males and 28 females. This gender distribution reflects the higher incidence rate of head and neck squamous carcinoma in the male population. All participants provided informed consent prior to their inclusion in the study. No compensation was offered to participants, as the study did not result in any adverse effects. These specimens underwent immunohistochemical analysis to assess CAV2

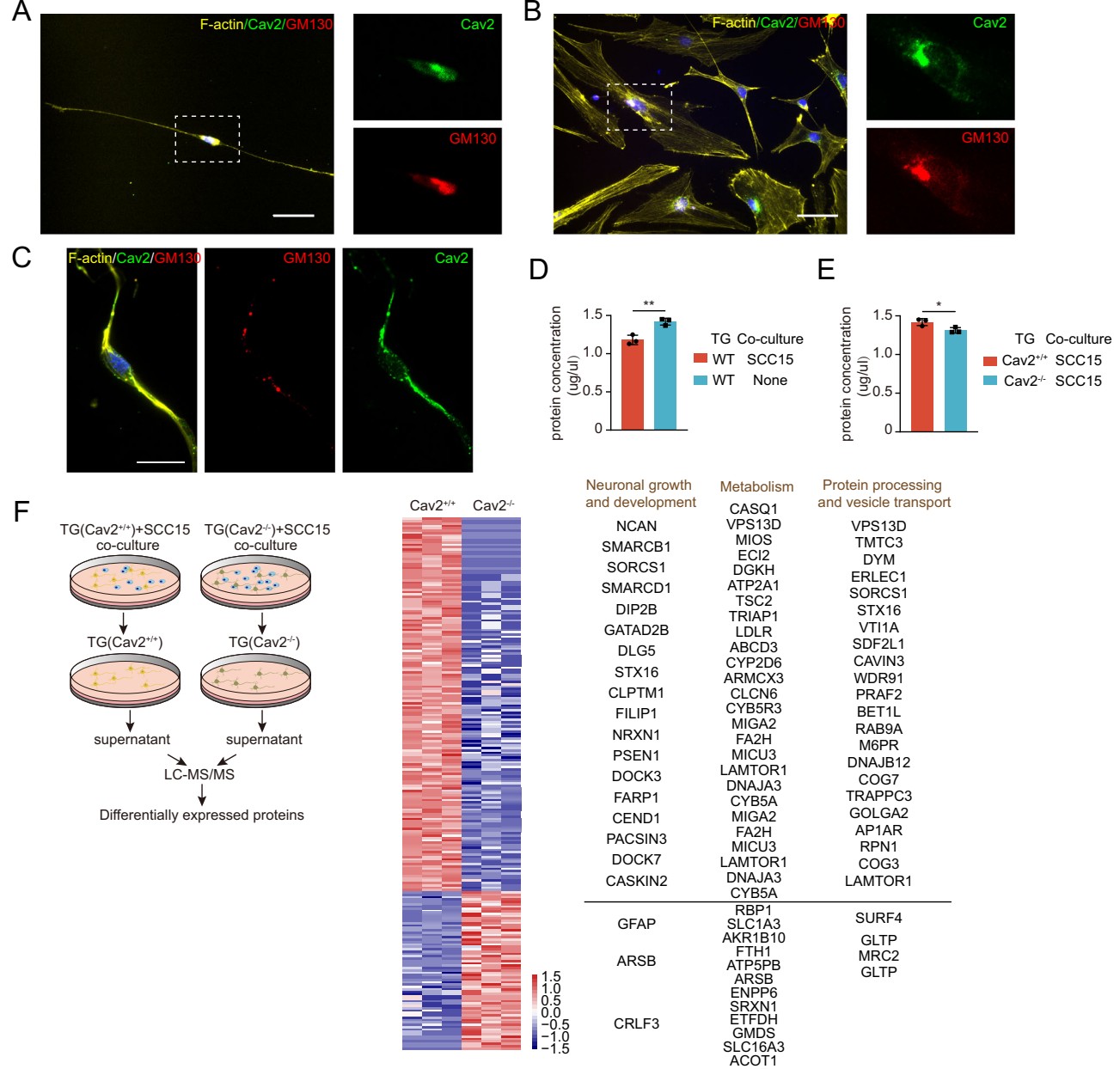

**Fig. 8 | Neural Cav2's Role in the Expression of Extracellular Proteins in Ganglia.**
**A** Cav2 is expressed in the Golgi apparatus of neurons within the trigeminal ganglia. Representative images are shown. Results were consistent across images from 10 cells. Scale: 50 μm. **B** Cav2 is also localized to the Golgi apparatus in non-neuronal cells of the trigeminal ganglia. Results were consistent across images from 10 cells. Representative images are shown. Scale: 50 μm. **C** After co-culture with SCC15 cells for 2 h, a subset of neurons exhibited Cav2 redistribution along the axons. Results were consistent across images from 5 cells. Scale: 20 μm. **D** Trigeminal ganglia from wild-type mice were cultured alone or co-cultured with SCC15 tumor cells for 48 h, followed by a 6-h period of independent culture in fresh medium. The protein concentration in the supernatant was subsequently measured. $n = 3$ biologically

independent experiments per group. $**p = 0.0057$, Student's t-test (two-sided). Data are mean ± s.d. **E** Trigeminal ganglia from Cav2+/+ or Cav2-/- mice were co-cultured with SCC15 tumor cells for 48 h, followed by a 6-h independent culture in fresh medium. The protein concentration in the supernatant was measured. $n = 3$ biologically independent experiments per group. $*p = 0.0330$, Student's t-test (two-sided). Data are mean ± s.d. **F** Mass spectrometry analysis was performed on supernatants collected from Cav2+/+ and Cav2-/- trigeminal ganglia co-cultured with SCC15 cells for 48 h, followed by a 6-h independent culture. The figure presents only those proteins with a fold change (FC) ≥ 2 or ≤1/2 and $p < 0.05$. $n = 3$ biologically independent experiments per group. Student's t-test (two-sided). WT wild-type, TG trigeminal ganglion.

localization within the HNSCC microenvironment and to investigate the correlation between neural CAV2 expression and patient survival outcomes. Furthermore, we extracted gene expression data encompassing 500 cases, along with pertinent clinical details, from TCGA for HNSCC. This data facilitated an extensive evaluation of the prognostic implications of CAV2 expression in bulk tissues and its interrelation with various clinical parameters.

## 4-NQO induced mouse model of HNSCC

The 4-NQO-induced murine model of HNSCC was generated following established protocols[23]. In summary, 4-NQO (N8141, Sigma-Aldrich) was dissolved in propylene glycol (W294025, Sigma-Aldrich), achieving a 100 μg /ml concentration, and was supplied to C57BL/6 mice via their water intake. The water was routinely replaced every two weeks over a 16-week treatment phase with 4-NQO. Following this phase,

mice reverted to a regular water diet, and health checks were conducted twice weekly until the endpoint was reached at euthanasia. Post-euthanasia, immediate dissection was performed to collect tongues and cervical lymph nodes, which were then meticulously examined for various lesion types (hyperplasias, dysplasias, and SCCs). Comprehensive lesion counts and area measurements were conducted. Histological investigations involved embedding the tissues in paraffin, slicing into 4-μm sections, and staining with hematoxylin and eosin (H&E).

## Animals

Cav2[+/-] mice (C57BL/6 background) were procured from Cyagen Biosciences (S-KO-01350, Cyagen Biosciences) and subsequently bred to yield Cav2[+/+], Cav2[+/-] and Cav2[-/-] offspring for utilization in in vivo studies, as well as for in vitro experiments necessitating TG or DRG. Separately, conditional Cav2 mice (C57BL/6 background), with loxP sequences flanking exon 1 of the Cav2 gene, were sourced from Cyagen Biosciences (S-CKO-01562, Cyagen Biosciences). These mice were crossbred with Adv-Cre (C57BL/6 background, NM-KI-215036, SHANG HAI MODEL ORGANISMS) or Aldh1l1-CreERT2 (C57BL/6 background, C001288, Cyagen Biosciences) strains to produce Adv[Cre]Cav2[flox/flox] or Aldh1l1[CreERT2]Cav2[flox/flox] genotypes. B6.129(Cg)-Gt(ROSA)26Sor[tm4(ACTB-tdTomato,-EGFP)Luo]/J mice (mTmG mice) were procured from The Jackson Laboratory (007676, The Jackson Laboratory).

All mice used in the animal experiments of this study were aged 6–10 weeks, including those administered with 4-NQO at the initiation of tumor induction (This excludes mice from which trigeminal ganglia (TG) and dorsal root ganglia (DRG) were harvested for in vitro experiments). In this study, the gender of the mice was determined by the sex of the offspring produced through mating and breeding, rather than being pre-selected. The sex of each mouse is indicated in Fig. 4, and no significant phenotypic differences were observed between males and females. All animal subjects were maintained in a thermo-regulated environment with a consistent 12:12-h light cycle, and were provided ad libitum access to nourishment and hydration. The composition of all diets, including chow (XTI01ZJ-012,XIETONG-SHENGWU), followed standard formulations provided by a reputable animal facility. Protocols conformed to the National Institutes of Health Guide for the Care and Use of Laboratory Animals. All animal experiments were conducted in accordance with the guidelines of the Animal Ethical and Welfare Committee of Tianjin Medical University Cancer Institute and Hospital, which permits a maximal tumor volume of 1000 mm³. We confirm that this limit was not exceeded in any of the experimental animals in our study.

## Retrograde tracer labeling

The procedure for retrograde tracer labeling was conducted in accordance with previously established methodologies[43]. Twelve days prior to MOC2 inoculation or seven days after MOC1 inoculation, the retrograde tracer, 1,1′-dioctadecyl-3,3,3′,3′-tetramethylindocarbocyanine perchlorate (DiI; HY-D0083, MCE), was prepared at a concentration of 170 mg/mL in DMSO, then further diluted in a 1:10 ratio with sterile saline. This solution was subsequently administered via injection into the anterior lateral region of the tongue, using approximately 7 μL per tongue, to label the trigeminal neurons innervating the tongue.

## Mouse tongue xenograft model

Consistent with methodologies delineated in prior studies[43], a respective total of $3 \times 10^5$ MOC1 cells and $5 \times 10^4$ MOC2 cells, each in a 30 μL mixture of DMEM and Matrigel (#354234, Corning) at a 1:1 ratio, were administered via injection into the anterior lateral regions of the tongues of various mouse models: Adv[Cre]Cav2[flox/flox] mice, Cav2[flox/flox] mice, Aldh1l1[CreERT2]Cav2[flox/flox] mice, Cav2[-/-] mice, and wild C57BL/6 mice. Subsequent monitoring of tongue tumors occurred twice a week, with

quantification efforts executed at 14 days post-inoculation. Upon the culmination of the study and the subsequent euthanasia of the mice, tongues were promptly excised, subjected to fixation in 10% formalin, embedded in paraffin, and sectioned into 4-μm slices. These specimens were earmarked for further immunoblotting and histological examinations. Regarding the sham groups shown in Fig. 2G and H, sham controls were injected with a tumor-cell–free buffer composed of 30 μL of DMEM and Matrigel.

## IHC and multiplex fluorescent IHC

The IHC assay was conducted in strict adherence to the methodologies outlined in earlier studies[60]. Evaluation of all expression scores was undertaken by two independent pathologists, who remained blinded throughout the process. For the implementation of multiplex fluorescent IHC, a specialized multiplex fluorescent IHC kit (10001100020, PANOVUE) was utilized in compliance with the manufacturer's prescribed protocol. The stained tissue specimens were digitally imaged using a confocal fluorescence microscope. The antibodies enlisted for the IHC assay, along with their corresponding sources, included: anti-CAV2 (NBP1-31116, Novus), anti-S100 (15146-1-AP, Proteintech), anti-panCK (#ab9377, Abcam), anti-Ki67 (#9027, Cell Signaling Technology), anti-TH (25859-1-AP, Proteintech), anti-TRPV1 (ACC-030, Alomone labs), anti-β3-tubulin (#ab52623, Abcam), anti-ALDH1A1 (15910-AP, Proteintech), and anti-CD44 (15675-1-AP, Proteintech), and anti-pS6 (#4858, Cell Signaling Technology), and anti-BMI1 (10832-1-AP, Proteintech), and anti-ALDH1L1 (#ab235197, Abcam), and anti-NDUFB8 (14794-1-AP, Proteintech), and anti-ATP5A1 (14676-1-AP, Proteintech), and anti-SDHB (10620-1-AP, Proteintech), and anti- UQCRC1 (21705-1-AP, Proteintech), and anti-CD31 (#ab281583, Abcam), anti- Cleaved Caspase-3 (#9664, Cell Signaling Technology).

## DRG and TG acquiring and primary culture

DRGs and TGs were extracted from Cav2[-/-] mice aged 2 to 4 weeks or their wild-type counterparts, following established protocols[61]. The method for acquiring DRGs and TGs involves anesthetizing mice, and then euthanizing them for dissection. To isolate TGs, incisions were made along the parietal bones post-euthanasia, and TGs were retrieved adjacent to the petrosal bone apex within the middle cranial fossa[62]. The procedure of acquiring DRGs requires precise surgical removal of organs and tissues to expose the lumbar spine, followed by careful extraction of the DRGs. The harvested ganglia were immediately rinsed in ice-cold PBS with 4% penicillin/streptomycin and conserved in neurobasal medium (#2458367, Thermo Fisher Scientific) enriched with B-27 supplement (#17504044, Thermo Fisher Scientific) and 4% penicillin/streptomycin (#15140122, Thermo Fisher Scientific).

## Isolation of TGs and co-injection assay

The dissected TG tissue was subjected to enzymatic digestion, beginning with a 10-minute exposure to 2 mg/ml papain (G8430, Solarbio) at 37 °C. This step was followed by low-speed centrifugation (300 × g) and the addition of a solution containing 0.1% collagenase II (C8150, Solarbio) and 0.25% trypsin in a 3:1 ratio. The mixture was then incubated in a water bath and centrifuged. Subsequent steps included the addition of 500 μg/ml DNase I (D8071, Solarbio), a 10-minute water bath incubation, and termination of the digestion process using DMEM/F-12 supplemented with 10% fetal bovine serum (FBS; ST30-3302p, PAN). The sample was then filtered through a 70 μm mesh. Post-filtration, cells were resuspended and plated onto poly-L-lysine-coated (100 μg/ml, P8141, Solarbio) coverslips and cultured for 24 h. The culture medium was later switched to Neurobasal-A (#2458367, Thermo Fisher Scientific) supplemented with 100 ng/ml NGF (#AF-450-01, PeproTech), B-27 (#17504044, Thermo Fisher Scientific), and glutamine.

A designated volume of MOC2 cells, illustrated in the accompanying figure, was amalgamated with digested TGs and dissolved in

30 μL of a 1:1 mixture of DMEM and Matrigel. This solution was then co-injected into the anterolateral aspect of the mouse tongue. Subsequently, the tumors were monitored daily, and the mice were euthanized after 14 days to ultimately confirm tumor formation.

## Immunofluorescence

For the immunofluorescence of dissected neurons or glial cells from TGs or DRGs, the cells were meticulously washed and blocked using 5% horse serum and 0.2% Triton X-100, followed by blocking for one h at room temperature. Following this, an incubation with the primary antibody immersed in 5% horse serum occurred at 4 °C for 16 h, succeeded by a 60-minute exposure to a FITC-conjugated secondary antibody (A21206, Invitrogen) under room temperature conditions. Fluorescence microscopy was employed for image acquisition. The antibodies utilized comprised anti-CAV2 (NBP1-31116, Novus), anti-β3-tubulin (ab52623, Abcam), and GM130 (AF8199, R&D).

## Co-culture experiments

For the transwell co-culture model, HNSCC cells were seeded in the upper chamber and ganglia (three TGs or six DRGs) were implanted in the low chamber of a 24-well transwell device with 0.4um pore size (Corning Incorporated). The same medium of 1:1 mix of DEME/F12 and neurobasal medium enriched with B-27 supplement was added to the upper and lower parts of the chamber. At 28 h after co-culturing HNSCC cells and TGs or DRGs, the membrane was removed and the cells that passed through the chamber were fixed in 4% paraformaldehyde for 30 min and then stained.

For the Matrigel bridge co-culture model, SCC15 cells were precisely resuspended in 20 μl of Matrigel (#354234, Corning) and established at a proximal distance of 2.5 mm from a subsequent 20 μl Matrigel aliquot. A TG was positioned upon the unoccupied 20 μl Matrigel locus. A continuous Matrigel conduit was then architecturally fabricated between the dual droplets, in accordance with the techniques delineated by Renz et al.[55]. The culture vessels were subjected to incubation for a twenty-minute duration at 37 °C in an atmosphere of 5% $CO_2$, preceding the supplementation with DMEM medium. On the tenth day, the quantification of the neural invasion index was executed by ascertaining the quotient derived from the distance navigated by the invasive HNSCC cellular entities along the TG neurites (α) relative to the aggregate interstice betwixt the neoplastic Matrigel locus and the TG Matrigel locus (β), as described by Gil et al.[63].

For other co-culture tests, HNSCC cells were placed in the top chamber, and ganglia (TG or DRG) at the bottom of the well. After 72 h, further analyses (such as mitochondrial assessment) were conducted on the ganglia or cancer cells.

## OCR analysis

The OCR levels were analyzed using an XFe24 Extracellular Flux Analyzer (Agilent Technologies), following the manufacturer's guidelines. Initially, $1.5 \times 10^4$ cells were allocated per well in 24-well micro cell culture plates (Agilent Technologies) and cultured in DMEM/F12 enriched with 10% FBS. They were then incubated at 37 °C overnight in a 5% $CO_2$ incubator. Subsequently, the original growth medium was substituted with pH 7.4 DMEM, devoid of phenol red and bicarbonate, and the cells underwent another incubation period at 37 °C in a $CO_2$-free incubator to stabilize atmospheric $CO_2$ levels. The XFe24 analyzer was employed to measure OCRs initially under baseline conditions and then following the administration of various metabolic drugs such as 1.5 μM oligomycin, 1.0 μM FCCP (carbonyl cyanide-p-trifluoromethoxyphenylhydrazone), and a combination of 0.5 μM Rotenone/Antimycin A, all of which are components of the Cell Mito Stress Test kit (#103015-100, Agilent Technologies).

## Flow cytometry

The ALDEFLUOR kit (# 01700, Stem Cell Technologies) was employed, following the manufacturer's guidelines, to gauge ALDH1A1 activity within cancer cells and sort cell groups exhibiting elevated ALDH1A1 activity. For mitochondrial assessment, cells were stained with 10 nM MitoTracker Red CMXRos (M7512, Invitrogen) under conditions of room temperature for 20 min or 5 μM Mitosox (M36008, Invitrogen) under conditions of 37 °C and 5% $CO_2$ for 30 min. For the apoptosis assay, HNSCC cells, whether co-cultured with wild-type TG or Cav2[-/-] TG, underwent treatment with the specified concentration of cisplatin, followed by dissociation using 0.25% trypsin and subsequent collection via centrifugation. The cells were then subjected to a 30-minute staining procedure at 4 °C utilizing the apoptosis kit from BioLegend, in compliance with the manufacturer's prescribed protocol. The BD LSRFortessa X-20 Cell Analyzer (BD Biosciences) and the CytoFLEX LX Cell Analyzer (Beckman Coulter) were utilized for all assay analyses in this study.

## Tumorsphere formation assays

The tumorsphere formation assay was conducted using $5 \times 10^2$ HNSCC cells cultured in 24-well ultra-low attachment plates, each fitted with a transwell chamber. TGs were specifically cultured in the upper compartment of these chambers. The culture medium employed was serum-free DMEM/F12 (#11320033, Thermo Fisher Scientific), enriched with 1% B27 supplement (#17504044, Thermo Fisher Scientific), 1% N2 supplement (#17502048, Thermo Fisher Scientific), and a mix of penicillin-streptomycin (100 μg/ml; #15140122, Thermo Fisher Scientific). Further supplementation included human recombinant epidermal growth factor (EGF; 20 ng/ml; #236-EG-01M, R&D Systems) and human recombinant basic fibroblast growth factor (bFGF; 10 ng/ml; #233-FB-025, R&D Systems). Culturing was performed in a 5% $CO_2$ incubator maintained at 37 °C, following established protocols[64]. Following a 12-day co-culture period, quantitative analyses were conducted to determine both the number and diameters of the resultant spheres.

## RNA sequencing and GSEA analysis

After 72 h of co-culture with wild TG or Cav2[-/-] TG, total RNA of SCC15 cells was extracted using TRIzol reagent, and RNA quality was assayed by Nanodrop and Bioanalyzer. Three biological replicates with good quality RNA were used to make cDNA libraries. The sequencing was performed on an Illumina NextSeq platform by BGI Genomics. GSEA was performed using GSEA 4.1.0 (http://www.broadinstitute.org/gsea/). To ensure that all measured transcripts contributed to pathway scoring, we did not perform a preliminary differential gene expression (DEG) filtering step. A nominal p-value < 0.05 and a false discovery rate (FDR) q-value < 0.25 were used as cutoffs for significance. MSigDB collections of h.all.v7.4.symbols[Hallmarks] and c2.cp.kegg.v7.4.symbols[Curated] were analyzed.

## Cell lines

SCC15 human squamous carcinoma cells (ATCC, CRL-1623), PC-12 rat pheochromocytoma cells (National Collection of Authenticated Cell Cultures, TCR-9), and murine oral carcinoma cell lines MOC1 (BFN6021632) and MOC2 (BFN6021637) (BLUEFBIO) were used. SCC15 cells were cultured in a 1:1 mixture of DMEM and Ham's F-12 containing 1.2 g/L sodium bicarbonate, 2.5 mM L-glutamine, 15 mM HEPES, and 0.5 mM sodium pyruvate, supplemented with 400 ng/mL hydrocortisone and 10% fetal bovine serum (FBS), according to ATCC. PC-12 cells were maintained in RPMI-1640 (Gibco, 11875093) + 10% FBS. MOC1 and MOC2 cells were cultured in high-glucose DMEM + 10% FBS + 1% penicillin–streptomycin. All cells were grown at 37 °C in a humidified incubator with 5% $CO_2$ and were routinely tested and confirmed mycoplasma-free.

## Apoptosis analysis

Post-treatment with predetermined chemotherapeutic agents, cell dissociation was effected using 0.25% trypsin, specifically omitting EDTA, followed by centrifugation. According to BD Biosciences' outlined procedure, these cells were then subjected to apoptosis assessment employing the Apoptosis Kit (#559763). The staining protocol initiated with a dual wash in ice-cold PBS, followed by cell resuspension in 1X Binding Buffer at a concentration of $1 \times 10^6$ cells/ml. A quantified cell aliquot was transferred to a culture tube, stained with PE Annexin V and 7-AAD, and incubated at room temperature shielded from light. Following this incubation, the introduction of extra Binding Buffer prepared the sample for imminent flow cytometry analysis.

## Western blotting

Cells were harvested and lysed in lysis buffer for 30 min at 4 °C. Total protein concentration was determined using a BCA protein assay kit (Thermo Fisher Scientific). Equal amounts of protein were separated by SDS-PAGE, transferred to PVDF membranes, and blocked. Membranes were then incubated with the primary antibodies listed in the Reporting Summary, followed by horseradish peroxidase-conjugated secondary antibodies (Cell Signaling Technology). Protein bands were detected using enhanced chemiluminescence (ECL) reagents (Merck Millipore). In the supplementary information, uncropped and unprocessed scans of the blots are provided.

## Quantification of mitochondrial DNA

The quantification of mitochondrial DNA within the cells was conducted employing the Human Mitochondrial DNA Monitoring Primer Set from Takara (#7246, Takara), strictly adhering to the manufacturer's prescribed guidelines. This process was preceded by the extraction of genomic DNA from HNSCC cells that had been subjected to co-culture with TG, either of the wild type or Cav2$^{-/-}$ variant.

## ROS-Glo H$_2$O$_2$ assays

The ROS-Glo H$_2$O$_2$ assay was conducted according to the manufacturer's instructions (#G8820, Promega). HNSCC cells, either cultured alone or co-cultured with Cav2$^{+/+}$ TGs or Cav2$^{-/-}$ TGs, were exposed to the H$_2$O$_2$ substrate solution over a span of 6 h. Post-incubation, the media were combined with the ROS-Glo detection solution that contains D-Cysteine, along with a signal enhancer solution. This mixture was then allowed to stand for 20 min at room temperature. Luminescence intensities were subsequently quantified.

## Mass spectrometry

For mass spectrometry (MS) analysis, three biological replicates were performed for each experimental group. Liquid chromatography-mass spectrometry (LC-MS/MS) was performed using a Vanquish liquid chromatograph (Thermo Fisher Scientific, USA) coupled with an Orbitrap Fusion™ Lumos™ Tribrid™ Mass Spectrometer (Thermo Fisher Scientific, USA).

Proteins were extracted using RIPA Lysis and Extraction Buffer. Briefly, samples were sonicated on ice with 200 μL of buffer for 10 min, followed by centrifugation at $12,000 \times g$ for 10 min at 4 °C. The supernatant was transferred to a pre-cooled 1.5 mL EP tube. For trypsin digestion, DTT was added to a final concentration of 10 mmol/L and incubated at 56 °C for 1 h. IAM solution was then added to 55 mmol/L, and the reaction proceeded in the dark for 40 min. Six volumes of pre-cooled acetone (−20 °C) were added, and the sample was frozen overnight at −20 °C. After centrifugation at $10,000 \times g$ for 10 min at 4 °C, the pellet was dried and resuspended in 100 μL 50 mM NH$_4$HCO$_3$. Trypsin (1 μg) was added, and the samples were incubated overnight at 37 °C. After digestion, peptides were desalted, concentrated, and dissolved in 0.1% formic acid. The supernatant was transferred to a sample tube for mass spectrometry analysis.

For LC-MS/MS, a 100 μm i.d. × 180 mm Reprosil-Pur 120 C18-AQ 3 μm column was used with a flow rate of 600 nL/min. Mobile phase A was 0.1% formic acid, and mobile phase B was 0.1% formic acid with 80% acetonitrile. The gradient was: 0 min (4% B), 2 min (8% B), 45 min (28% B), 55 min (40% B), 56 min (95% B), and 66 min (95% B). MS parameters included a resolution of 120,000, AGC target of 400,000, maximum IT of 50 ms, and scan range of 400 to 1200 m/z. For MSn scanning, resolution was 30,000, AGC target was Standard, maximum IT was 54 ms, and scan range was 200 to 2000 m/z, with an NCE of 30.

Peptide identification was performed using Spectronaut 14.2 (Biognosys) with the UniProt-Homo sapiens database. The search parameters included fixed carbamidomethylation on cysteine **C** and variable oxidation on methionine **M** and acetylation at the N-terminus. Trypsin was used as the enzyme with a maximum of 2 missed cleavages. The mass accuracy for precursor and fragment ions was set to 20 ppm, with MS1 accuracy also at 20 ppm. Peptide and protein identification was validated with a 1% false discovery rate (FDR), and only proteins identified with at least two unique peptides were considered. Statistical analysis was conducted to assess protein expression differences between groups. The mass spectrometry proteomics data have been deposited to the ProteomeXchange Consortium via the PRIDE partner repository with the dataset identifier PXD060212.

## Isolation of lipid rafts

Lipid rafts (LR) were isolated as described previously[65]. PC12 cells were collected in ice-cold PBS, resuspended in 0.5 M Na$_2$CO$_3$ (pH 11.0), and homogenized by sonication. The homogenate was adjusted to 45% sucrose and layered over a sucrose gradient (5% and 35% sucrose in 250 mM Na$_2$CO$_3$). After centrifugation at 39,000 rpm for 14 h (SW41 rotor, Beckman), lipid rafts appeared as a band at the 5–35% sucrose interface. Fractions were collected, and SDS-PAGE analysis confirmed lipid rafts in fractions 3 and 4.

## Statistics & reproducibility

The study design followed standard protocols for experimental reproducibility. All experiments were performed with at least three independent biological replicates, where each data point represents an individual sample. Reproducibility was ensured by conducting experiments with independent biological replicates, and similar results were obtained across all repetitions. No statistical method was used to predetermine sample size. Sample sizes were chosen based on prior experience, the feasibility of performing the experiments, and the number of transgenic mice available for breeding. No data were excluded from the analyses. The experiments were not randomized, and the researchers were blinded during the measurement of the tumor size and body weight of mice.

Statistical analyses were conducted using Prism 9.5.1 and R (version 4.0.3) for statistical computing. Statistical methods, including whether the tests were one-sided or two-sided and the corresponding p-values, are detailed in the figure legends for each panel.

## Ethics statement

This study was conducted in accordance with all relevant ethical regulations and was approved by the Institutional Review Board (IRB) of Tianjin Medical University Cancer Institute and Hospital. Informed consent was obtained from all human participants. All animal experiments were conducted in compliance with the guidelines set forth by the Animal Ethical and Welfare Committee of Tianjin Medical University Cancer Institute and Hospital.

We are committed to inclusivity in both our research team and participant selection. All data presented in this study are publicly available in accordance with Nature Communications policies. We strive to maintain transparency and ensure that no participants were excluded based on gender, race, or other characteristics.

## Reporting summary

Further information on research design is available in the Nature Portfolio Reporting Summary linked to this article.

## Data availability

The RNA sequencing (RNA-seq) data generated in this study have been deposited in the GEO database under accession number GSE288675. The mass spectrometry data generated in this study have been deposited in the PRIDE repository under accession number PXD060212. These repositories provide open access to the dataset needed to interpret, verify, and extend our findings. The remaining data are available within the Article, Supplementary Information or Source Data file. Source data are provided with this paper.

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

## Acknowledgements

This study was supported by the Natural Science Foundation of China (No. 82073006(J. L.), 82002757(R.L.)), the Natural Science Foundation of Tianjin City (25JCZDJC00710(J.L.)), and the Tianjin Key Medical Discipline (Specialty) Construction Project (TJYXZDXK-009A). We thank the TCGA project for its valuable contributions to HNSCC research.

## Author contributions

Z.Z., J.L., H.G., and R.L. contributed to the study design. Z.Z., C.W., Z.S., R.L., X.S., Y.W. (Yafei Wang), and Y.S. contributed to implementing the experiment and data collection. Z.Z., Y.W., and Z.Z., R.L.contributed to the data analysis. Z.Z. and R.L. contributed to the writing of the draft version of the manuscript. All authors have read, revised, and approved the final version of the manuscript being submitted for peer review.

## Competing interests

The authors declare no competing interests.
