## [Transparent Peer Review file · Nature Communications]

CAV2-expressing Nerves induce Metabolic Switch toward Mitochondrial Oxidative Phosphorylation to Promote Cancer Stemness

Corresponding Author: Dr Ze Zhang

Version 0:

Reviewer comments:

Reviewer #1

(Remarks to the Author)

Liu et al. were able to show the importance of CAV2+ nerves stimulate mitochondrial oxidative phosphorylation (OXPHOS) and thereby promoting cancer stemness in HNSCC. The finding is very interesting and provides the link between the microenvironment of neural cells and tumorigenesis, which is largely unknown in HNSCC. Most experimental designs are rigorous, and the phenotypic results are very convincing. However, the molecular mechanisms underlying these events are still not clear. How CAV2+ nerves stimulate OXPHOS and how OXPHOS promotes cancer stemness are not investigated. Furthermore, additional validations are required to fully support their conclusion. And at last, some other minor issues also reduce the overall quality of this manuscript.

I have several concerns listed below.

1. The major concern of this study is lacking mechanistic potential. How CAV2+ nerves stimulate OXPHOS and how OXPHOS promotes cancer stemness have not been investigated.
2. Since CAV2 expression is observed in vascular sites, the authors may consider demonstrating whether co-culturing endothelial cells with SCC15/MOC2 induces OXPHOS or not. I understand that the authors did not pursue endothelial CAV2 due to the insignificant association between lymphovascular invasion and CAV2 expression, according to TCGA data. However, experimental evidence regarding this question would be a great addition to the paper.
3. Aldh111 is ubiquitously expressed among tissues, not specifically in glial cells. Whether the glial Cav2 plays a significant role in promoting tumor progression needs further validation in their mouse models.
4. It would be more convincing if the HNSCC cancer stem maker's expression were examined in mice tumor tissues in Fig. 4 and Fig. 7, such as ALDH, CD44, SOX2, or BMI1.
5. The quality of Fig. 1A could be better. Visualizing the box plot, as covered by the dots, is very hard.
6. Fig. 3B is missing labels.
7. The statistical analysis should be performed in Fig. 6J.
8. Fig. 7K does not have any label.
9. Fig. 7L-7S are not labeled very clearly.

Reviewer #2

(Remarks to the Author)

CAV2-expressing nerves induce metabolic switch toward mitochondrial oxidative phosphorylation to promote cancer stemness

- What are the noteworthy results?
- Will the work be of significance to the field and related fields? How does it compare to the established literature? If the work is not original, please provide relevant references.
- Does the work support the conclusions and claims, or is additional evidence needed?
- Are there any flaws in the data analysis, interpretation and conclusions? Do these prohibit publication or require revision?
- Is the methodology sound? Does the work meet the expected standards in your field?
- Is there enough detail provided in the methods for the work to be reproduced?

In this manuscript, Liu et al characterize an interesting association between caveolin-e (CAV-2) expression in neurons and glia and the induction of mitochondrial oxidative phosphorylation in tumor cells. The authors use a combination of in vitro co-culture [tumor cells/ganglia (trigeminal or DRG)] as well as in vivo studies with transgenic animals in which Cav2 is either knocked out of neurons, glia or the whole mouse and assess the impact on disease. The authors home in on ribosomal protein S6 (a marker of mTORC1 activation) yet why they even looked at this protein is not clear. The jump is made without any rationalization which makes it difficult to follow.

While this type of neuronal/glia influence on tumor cells is novel, the mechanism is not delineated. Moreover, a study in 2012 (FASEB J PMID: 22859397) demonstrated that caveolae can mediate the transfer of caveolin into mitochondria and that this is a stress response that alters mitochondrial function. It seems that something similar might be at the root of the findings in this study with the initiation of transfer of caveolin-2 occurring from tumor-infiltrating nerves (or their associated glial cells). If so, neuronal transfer of CAV-2 into tumor cells and the subsequent trafficking to mitochondria would be very novel. However, the mechanism is not defined. A simple way to assess this mechanism would be to fluorescently tag-CAV2 in neurons and co-culture with tumor cells. Is the fluorescently tagged CAV2 now found in tumor cell mitochondria? Can this be blocked by inhibiting endocytosis?

The authors utilized Adv-cre::Cav2-f/f to deplete CAV2 from sensory neurons which is a good strategy. However, they do not validate that KO of CAV2 does not alter the number of Advilin positive nerves. They find a decrease in tumor volume in Adv-cre::Cav2-f/f mice but if the number of Advilin positive nerves is decreased in these animals, there would be decreased tumor innervation and subsequently decreased tumor growth. This would have nothing to do with the absence of CAV2 in the nerves but rather an absence of nerves themselves. This was not shown and needs to be demonstrated. If there is no impact of loss of CAV2 on the nerve population, then the decreased tumor growth is mediated by their absence of CAV2 but this has not been shown with the current data.

Similarly, the Aldh111-CreERT2 mouse is crossed with the Cav2-f/f to KO CAV2 expression in glial cells. However, they do not show that glial cells in the periphery have been eliminated. These simple controls must be included.

The authors mention the paper by Zhang et al in which CGRP-expressing nerves alter tumor cell metabolism but argue it is via a different mechanism than demonstrated here. If CAV2 expression is within sensory nerves, it would be unlikely that they are not also CALCA (CGRP gene) expressing. This would be simple to determine by doing CAV2/CGRP double staining alongside, beta-III tubulin/CAV2 and beta-III tubulin/CGRP staining.

DRG and TGM are isolated from juvenile mice (age 2-4 weeks). Why so young? HNSCC is an adult disease. Why not do experiments with ganglia isolated from adult mice? What is the rationale?

Retrograde labeling of nerves in the tongue is performed BEFORE tumors are present. This will label resident nerves, not necessarily nerves that sprout and newly infiltrate the tumor once the tumor grows. These newly sprouted nerves are the ones of interest and they are not being tagged.

How was neural CAV2 expression defined? Was there co-staining for beta-III tubulin and CAV2? (page 7, line 153).

There is a consistent utilization of trigeminal and DRG ganglia but it is unclear why both are used. The disease being studied is HNSCC so the logical ganglia is the TGM. Why use DRG at all? This is not explained.

The sex of mice used in these experiments is not mentioned. If both sexes were utilized, were any sex-differences identified?

There needs to be a reference for the statement that HNSCC HPV negative patients have a heightened predisposition to PNI (page 6, line 130).

Figures:

Figure 1F is too small is very difficult to see anything. This is true of several figures: Fig 6H, Fig 1G, Fig 1J, Fig 4O, Fig 4N, Supplemental figure 1F

Fig 1J: why isn't the normal mucosa innervated? It would be helpful to add beta-III tubulin (pan neuronal marker) to determine if all tumor-infiltrating nerves are CAV2 positive. An H&E panel would also be helpful.

In figure 1I, there is a patient population labeled "TJCH cohort" but what this cohort is was not defined.

Fig 1E and F: please add arrows to point out caveolae.

Fig 2A: would be easy to do densitometry to look at the differences; this would allow for statistical analysis as well.

Fig 2D: are these glial cells that migrated off the ganglia? They look like fibroblasts. Staining with a glial marker would be helpful. If these are glia that migrated off ganglia, how many days did this take? The time could influence expression of CAV2.

Fig 3A: what is the difference between TGM and DRG that leads to the difference in migration?

Fig 3B: I'm not sure what I'm supposed to be seeing here. Arrows would be helpful.

Fig 3C: this needs to be brighter to make the point.

Fig 4E: looks like a lot of non-specific staining. Are these fibroblasts? Tumor cells? Double IHC or IF would help define these CAV2 expressing cells.

Fig 6A: what % of cells are ALDH1+ cells?

Fig 6E: it is very difficult to interpret much from this figure. Too small, too dim. Cav2 and CD44 colors too similar.

Supplemental Figure 2: needs to be brighter to make the point.
Supplemental Figure 4E: needs to be brighter. When printed, the panels look all the same.

Reviewer #3

(Remarks to the Author)

The authors showed an interesting correlation between CAV2 expression and HSCC prognosis. The increased expression of CAV2 expression in HSCC patients with PNI together with the predominant expression of CAV2 in neural structures suggest the involvement of CAV2 in nerve-cancer crosstalk. The ability of cancer cells to induce CAV2 expression was demonstrated in vitro, by co-culturing TG or TG cells with cancer cells, and in vivo, by performing xenograft and a 4-NQO-induced murine HNSCC model.

Interestingly, depletion of CAV2 expression in sensory neurons reduced tumor growth and changed the metabolic profile of the cancer cells. Based on these observations, the authors conclude that “CAV2-expressing Nerves induce Metabolic Switch toward Mitochondrial Oxidative Phosphorylation to Promote Cancer Stemness”.

This is a potentially interesting paper. Unfortunately, the manuscript is very descriptive and mechanistic insights are lacking. I feel that there is something of an opportunity missed to deeply explore the role of CAV2 in tumorigenesis, that will allow important follow up studies relating to therapeutic approaches.

The study brings new observations but the actual novelty on the underlying mechanism seems still limited.

Major concerns:

(1) The authors did not elucidate the mechanisms linking Cav-2 expression by nerves to the reduced tumor growth, metabolic rewiring of the cancer cells, and cancer stemness. Although these are interesting observations, it is not possible to understand how a protein present in caveolae can affect all these processes. More specifically:

- Which cellular processes are affected by Cav-2 loss in nerves and how they could be relevant for the described mechanism?

-Which is the cancer-signal that induce Cav-2 expression in nerves?

-Why/How Cav-2 overexpression should promote cancer cells migrations?

-Why/How Cav-2 overexpression should affect neurogenesis?

-Why/How Cav-2 expression in nerves induce the metabolic rewiring?

Understanding these mechanisms could have important clinical implications.

It is more than surprising to find no discussion about the known functions of Cav-2 and the potential role of Cav-2 in the described mechanism.

Based on these observations the authors cannot state “This interplay between cancer cells, neurons, and glial cells unveils a mechanism through which tumor-associated nerves contribute to cancer stemness via metabolic reprogramming, presenting a novel avenue for anticancer therapy” (Lanes 59-60). In the absence of experimental data supporting the mechanisms underlying the reported observations, the conclusion should be softened.

Also, the “novel avenue for anticancer therapy” is very speculative. Is it feasible to pharmacologically target Cav-2? This should also be discussed.

(2) The authors showed reduced tumor growth in Cav2^{-/-} animals. Surprisingly, deletion of Cav2 only in sensory neurons had a modest impact on tumor growth whereas the deletion of Cav2 in glial cells strongly suppressed tumor growth (Lanes 250-254). Although the authors themselves concluded that Cav2 expression in glial cells play a major role in tumor growth (Lane 264), in the next paragraphs the authors continued to study the impact of nerve-Cav-2 on tumor growth. Considering the predominant role of CAV2 ablation in glia cells, it is difficult to understand the reasons why the authors decided to focus on the role of Cav-2 in nerves instead of the role of Cav-2 in glial cells on cancer cell metabolism.

Furthermore, why do authors analyze the influence of neural Cav-2 on the metabolism of cancer cells? Why CAV2 should affect the metabolism of the cancer cell? Which is the rationale? Although the observations are interesting the mechanism is lacking.

Then, in the last paragraph, the authors used the 4-NQO-induced model of HSSC and they evaluated the impact of Cav2 depletion, either globally and specifically in sensory neurons or glial cells. Also in this case, upon deletion of Cav2 globally or in glial cells only 1/10 mice developed cancer. The impact of Cav2 deletion in sensory neurons is modest. The authors concluded that these results highlight the importance of Cav-2 in glial cells on tumor initiation. Furthermore, the authors concluded the manuscript by evaluating the impact of the global deletion on tumor growth and metastasis.

As mentioned above, why the authors did not further explore the role of Cav2 in glial cells on tumor growth.

In my opinion, the logical description of data in the entire manuscript is confused

Additional questions:

(3) “Depletion of Neural Cav2 Impedes Tumor Cell Infiltration Toward Nerves and Tumor-Associated Neurogenesis”. In this

paragraph, the authors provided only in vitro evidence. Is it also true in vivo? The paper would benefit from addressing PNI in vivo, in both the xenograft and in the 4-NQO-induced murine HNSCC model.

(4) “Cav2+ Nerve-Driven Shift to Enhanced Mitochondrial Respiratory Phenotype in Cancer Cells”

This conclusion derives from in vitro data. Does Cav2-expression in TG cells and glial cells affect the metabolism of the cancer cells in vivo?

In addition to the evaluation of oxygen consumption rates, did the authors also measure glycolysis and lactate production/secretion in vitro?

(5) “Depletion of Cav2 Inhibits 4-NQO-Induced Oral Tumorigenesis”

In this paragraph and related picture, more experiments are required. For example:

Which is the impact on animal survival? Kaplan-Maier curves should be reported for the three models: global Cav-2 KO, Sensory neurons Cav-2 KO, glial cells Cav-2 KO.

Concerning the Sensory neurons Cav-2 KO - glial cells Cav-2 KO, the authors did not show tumor volume or metastasis. Is there an impact on PNI or tumor innervation in the three mouse models (global Cav-2 KO, Sensory neurons Cav-2 KO, glial cells Cav-2 KO) upon 4-NQO-Induced Oral Tumorigenesis?

Minor questions:

(6) The quality of all immunofluorescence analyses is poor. Images with higher resolution are required. For example:

-Fig. 1F: the colocalization between CAV2 and TRPV1 is difficult to observe.

-Fig. 2H: the colocalization between CAV2 and TdTomato is not clear, higher resolution is necessary.

-Fig 2J: The increased expression of Cav2 in nociceptive neurons in the 4-NQO-induced murine HNSCC model is not evident. Higher resolution and higher magnification images are required.

-The resolution of transmission electron microscopy images (TEM) pictures is low (Fig.2E and 2F).

The manuscript would also benefit from image analysis of the co-localization efficiency.

(7) Figure 1H shows the correlation between overall survival and CAV2 expression, based on TCGA cohort sequencing.

Which is the difference between this analysis and the one the same authors reported in the previous paper (<https://doi.org/10.1038/s41420-022-01176-1>)?

(8) The authors showed that CAV2 expression and the number of caveolae increases upon co-culture of TG cells with cancer cells. Which is the underlying mechanism?

(9) There is no information about statistical analyses in the figure legends: type of analysis, N, P values are absent. What is represented in the graph? Media +/- SD or SEM? This is true for Fig.2 and others. many figure legends are poorly described

(10) “Ki-67 and cleaved caspase-3 markers to probe tumor proliferation and apoptosis yielded no notable disparities between the two mouse cohorts”.

Which is the reason for these observations?

What about necrosis?

(11) “platinum-based chemotherapeutics revealed that SCC15 cells co-cultured with Cav2+/+ TGs displayed reduced apoptosis following cisplatin treatment compared to SCC15 cells co-cultured with Cav2-/- TGs”.

What about in vivo?

Version 1:

Reviewer comments:

Reviewer #1

(Remarks to the Author)

The author has presented additional evidence to support their intriguing findings. Although the mechanisms are not yet fully understood, the novelty of this study would be a significant contribution to the field.

Reviewer #2

(Remarks to the Author)

The authors did an excellent job in this revised manuscript. They addressed all of my questions, many of which required additional experiments. I believe that the modifications have significantly improved the manuscript and strengthened their findings and conclusions.

Reviewer #3

(Remarks to the Author)

The authors answer almost all the concerns. I appreciated the efforts made to investigate the mechanism through which

Cav2 regulates OXPHOS in cancer cells. We also understand that fully elucidating this mechanism may not be feasible within the limited timeframe of the revision. The authors have provided new insights that could contribute to understanding this process, including the identification of several mitochondrial proteins in the conditioned medium. However, it is not immediately intuitive how these secreted proteins might modulate OXPHOS in cancer cells. The identified proteins are not usually secreted by the cell. Are there any published studies describing something similar? The manuscript may benefit from further discussion on this point.

Version 3:

Reviewer comments:

Reviewer #5

(Remarks to the Author)

This manuscript presents exciting findings demonstrating that HNSCC cells can induce CAV2 expression in trigeminal ganglia and associated neural fibers, which in turn promotes mitochondrial oxidative phosphorylation (OxPhos) in cancer stem-like cells (CSCs), reinforcing their stemness and contributing to tumor aggressiveness. The data highlight a unique cancer - nerve metabolic interaction, potentially advancing our understanding of how the tumor microenvironment supports CSC function and resistance mechanisms.

The authors support their conclusions with a range of in vitro, in vivo, and transcriptomic analyses, including TCGA data showing CAV2 as a strong prognostic marker in HNSCC. Importantly, they demonstrate that CAV2-positive nerves contribute to a metabolic shift toward OxPhos specifically in CSC mitochondria, a mechanistic insight of potential therapeutic significance.

While the study is generally well-conducted and the conclusions are compelling, several points require clarification or further elaboration.

Minor Comments:

Line 164 / Figure 2A: The text references a comparison between exposed and non-exposed trigeminal ganglia (TGs), but this comparison is not shown in the figure. Including this in the figure panel would enhance clarity and support the interpretation.

Figures 2G and 2H: The “sham” condition needs clarification. Does it refer to animals that had no tumor inoculation, or a different control setup? Please define “sham” explicitly in both the figure caption and methods.

Figure 5: GSEA Analysis (GEO data): The authors report DNA repair as an enriched pathway, but analysis of the referenced GEO dataset primarily reveals OxPhos and MYC target enrichment, no DNA Repair. However, it is unclear whether a differential gene expression (DEG) step was performed prior to GSEA. If so, this should be described in the Methods section. Otherwise, revise the pathway enrichment claims accordingly.

Line 411: Please add a sentence indicating how the observed gene expression profile aligns more closely with ALDH+ or ALDH- cells, as reported in Reference 35, which is on breast cancer. This would help contextualize the findings for HNSCC within known CSC phenotypes.

Line 413: The manuscript describes GSEA of SCC15 cells, but what were the results for MOC2 cells? These should be included for completeness.

Figures 6H & 6I: Are there equivalent results available for MOC2 cells in these panels?

Figure 8F: The term “differentially expressed genes (DEGs)” is incorrect here, as the data refer to proteins. Please revise to “differentially expressed proteins” to avoid confusion.

Version 4:

Reviewer comments:

Reviewer #5

(Remarks to the Author)

I would like to thank the authors for their careful and detailed responses to the comments raised during this round of review. I appreciate the effort that has gone into addressing the concerns and clarifying points raised, as well as the thoughtful explanations provided regarding the feasibility of additional experiments.

Thank you for your thoughtful explanation regarding the feasibility of generating MOC2 data. I agree that inclusion of GSEA and corresponding analyses in MOC2 cells would improve the completeness of the study. At the same time, I understand the logistical and ethical challenges associated with additional Cav2-/- breeding.

Given this context, I do not consider the absence of full MOC2 replication to be a barrier to publication. The manuscript already presents a substantial body of data that supports the conclusions, and the addition of MOC2 experiments would, in my view, primarily serve to further strengthen completeness rather than alter the central findings.

If feasible, the inclusion of any partial or surrogate MOC2 analyses would still be a valuable addition, but I leave this to the editors' and authors' discretion. Overall, I find the manuscript suitable for publication if now the data availability requirements are fulfilled.

REVIEWER COMMENTS

Reviewer #1 (Remarks to the Author):

Liu et al. were able to show the importance of CAV2+ nerves stimulate mitochondrial oxidative phosphorylation (OXPHOS) and thereby promoting cancer stemness in HNSCC. The finding is very interesting and provides the link between the microenvironment of neural cells and tumorigenesis, which is largely unknown in HNSCC. Most experimental designs are rigorous, and the phenotypic results are very convincing. However, the molecular mechanisms underlying these events are still not clear. How CAV2+ nerves stimulate OXPHOS and how OXPHOS promotes cancer stemness are not investigated. Furthermore, additional validations are required to fully support their conclusion. And at last, some other minor issues also reduce the overall quality of this manuscript.

I have several concerns listed below.

1. The major concern of this study is lacking mechanistic potential. How CAV2+ nerves stimulate OXPHOS and how OXPHOS promotes cancer stemness have not been investigated.

Response: Thank you for your valuable comments. You have raised two points

regarding the mechanistic gaps in our study: first, how CAV2+ nerves stimulate oxidative phosphorylation (OXPHOS); and second, how OXPHOS promotes cancer stemness.

Regarding the first point, we have further explored the specific localization of Cav2 in neurons and non-neuronal cells within the trigeminal ganglion. High-magnification confocal imaging revealed that Cav2 is primarily localized to the Golgi apparatus (see Fig. 8 in the revised manuscript). Additionally, after co-culture with tumor cells, we observed that Cav2 began to be expressed along the axons of a small proportion of neurons, while in most cells, Cav2 remained in the Golgi apparatus. Based on these observations, we speculate that Cav2 may be involved in biological processes such as protein modification, processing, and secretion in neural cells.

Next, we co-cultured trigeminal ganglia from Cav2^{+/+} or Cav2^{-/-} mice with SCC15 (HNSCC cells) for 48 hours, then removed the tumor cells and replaced the culture medium. After 6 hours, we collected the supernatant from the Cav2^{+/+} or Cav2^{-/-} trigeminal ganglia and measured the protein concentration. The results showed that the protein content in the supernatant of Cav2^{+/+} trigeminal ganglia was significantly higher than that of Cav2^{-/-} ganglia, further confirming that neural Cav2 may be involved in protein processing and secretion.

We then used the same experimental approach (co-culturing trigeminal ganglia

from Cav2^{+/+} or Cav2^{-/-} mice with SCC15 cells for 48 hours, removing the tumor cells, and replacing the medium, followed by supernatant collection after 6 hours) and performed quantitative mass spectrometry analysis to examine differences in protein expression in the supernatant. The differentially expressed proteins were mainly related to neural growth and development, metabolism, and protein processing and vesicle transport. Focusing on cellular metabolism, particularly oxidative phosphorylation, we found several proteins in the supernatant that may affect mitochondrial metabolism, such as CYB5A (cytochrome b5) and CYB5R3 (cytochrome b5 reductase), which are involved in electron transport and may be associated with OXPHOS. ATP5PB, as a subunit of ATP synthase, is directly involved in OXPHOS and is a key component of the mitochondrial ATP synthase complex. ETFDH (electron-transferring flavoprotein dehydrogenase) is involved in mitochondrial fatty acid oxidation and is linked to OXPHOS. MICU3 (mitochondrial calcium uptake protein 3) regulates calcium homeostasis, which may influence mitochondrial OXPHOS function. MIGA2 (mitochondrial intermembrane space assembly protein) regulates mitochondrial dynamics and is related to OXPHOS, while TRIAP1 (TP53-regulated inhibitor of apoptosis 1), an inner mitochondrial membrane protein, is involved in protein folding and oxidative stress response, indirectly contributing to OXPHOS. VPS13D, which regulates mitochondrial quality control, also affects mitochondrial health and dynamics, thus indirectly impacting OXPHOS.

In addition, the differentially expressed proteins include numerous other

metabolic-related proteins involved in glucose, lipid, and amino acid metabolism, processes that can also affect cellular OXPHOS levels. Therefore, we propose that neural Cav2 orchestrates OXPHOS by regulating the extracellular secretion of a large number of metabolism-related proteins, rather than simply by influencing the secretion of a single cytokine or a specific signaling pathway in tumor cells (please see the marked revised manuscript, lines 504-555 and Figure 8).

Of course, the more specific mechanisms require further study, such as which of these differentially expressed proteins can be taken up by tumor cells, how much is taken up, and through which pathways they influence OXPHOS. These are still unclear, and we hope you understand that these questions cannot be fully addressed within the scope of a single study.

Regarding the second point, indeed, elucidating how OXPHOS promotes tumor stemness is a quite complex issue, primarily because a shift in metabolic phenotype entails alterations in numerous metabolic products, which interact and continuously transform within metabolic processes. Many high-profile studies focusing on tumor metabolism and stemness have indicated that OXPHOS enhances tumor stemness (at a phenotypic and phenomenological level) but have not delineated the mechanisms in detail (such as Nature 2014; Cell Metab 2017)[1, 2].

There are some findings in the literature providing clues in this area. For instance,

it has been reported that α -ketoglutarate (α -KG) sustains the pluripotency of embryonic stem cells and that direct manipulation of intracellular α -KG/succinate levels can regulate various chromatin modifications, including H3K27me3 marks and Tet-dependent DNA demethylation, which are associated with the regulation of pluripotency-related gene expression (Nature 2015)[3].

Further studies have shown that intestinal stem cells, especially Lgr5⁺ crypt base columnar cells (CBCs) in the small intestine, exhibit high levels of OXPHOS activity, crucial for meeting their high proliferative demands. Mitochondrial function relates not only to energy production but also to the regulation of the redox balance and the generation of reactive oxygen species (ROS). ROS, acting as signaling molecules, are involved in regulating stem cell self-renewal and differentiation (Nature 2017)[4].

Additionally, mitochondrial dysfunction has been closely linked to the aging of muscle stem cells (MuSCs). Aged MuSCs exhibit decreased mitochondrial membrane potential and reduced cellular ATP levels, indicating impaired mitochondrial function. Treatment with nicotinamide riboside (NR), a precursor of NAD⁺, has been shown to improve mitochondrial function. NAD⁺ supplementation not only delayed the onset of cellular aging in MuSCs but also enhanced their regenerative capacity, and its benefits extend to neural and melanocyte stem cells (Science 2016)[5].

However, as our study primarily focused on the impact of neural-expressed CAV2

on tumor metabolic changes and stemness, we regret that we currently lack the capability to delineate in detail the mechanisms by which OXPPOS promotes tumor stemness through our own experiments. We appreciate your understanding and will continue to monitor advancements in this field, hoping to further elucidate these mechanisms in future research.

2. Since CAV2 expression is observed in vascular sites, the authors may consider demonstrating whether co-culturing endothelial cells with SCC15/MOC2 induces OXPPOS or not. I understand that the authors did not pursue endothelial CAV2 due to the insignificant association between lymphovascular invasion and CAV2 expression, according to TCGA data. However, experimental evidence regarding this question would be a great addition to the paper.

Response: Thank you for your professional and helpful suggestion. Based on your comment, we initially attempted to isolate endothelial cells from the tongues of Cav2^{+/+} and Cav2^{-/-} mice using Fluorescence-Activated Cell Sorting (FACS) with CD31 and other markers, followed by co-culture with tumor cells to assess the impact of endothelial CAV2 on tumor cell OXPPOS and related phenotypes. However, after several attempts, we found it challenging to obtain a sufficient number of viable endothelial cells in good condition for co-culture with tumor cells using this method.

Therefore, we adopted an alternative approach by infecting HUVECs (Human

Umbilical Vein Endothelial Cells) with shCAV2 lentivirus and control lentivirus. After selection with puromycin, we co-cultured these HUVECs with tumor cells and measured mitochondrial superoxide levels in the tumor cells. Our results showed no significant difference in mitochondrial superoxide levels between tumor cells co-cultured with shCAV2 HUVECs and those co-cultured with control HUVECs (Please see the marked revised manuscript, lines 370-376).

3. Aldh111 is ubiquitously expressed among tissues, not specifically in glial cells. Whether the glial Cav2 plays a significant role in promoting tumor progression needs further validation in their mouse models.

Response: Thank you for raising this highly pertinent and insightful question. This issue has indeed been a longstanding challenge in our research. Glial cells in peripheral ganglia, such as the dorsal root ganglia (DRG) and trigeminal ganglia (TG), are distinct from those in the central nervous system (CNS). Cre-driver mouse lines for targeting various types of glial cells in the CNS are well-established and readily available; however, suitable Cre-driver mouse lines for specifically targeting glial cells such as satellite glial cells and Schwann cells in DRG and TG are still lacking.

Based on the literature, Aldh111 is expressed in glial cells in peripheral ganglia[6], which prompted us to utilize Aldh111 for our studies. In response to the comments from you and other reviewers, we first conducted multicolor immunohistochemical staining

of human HNSCC tissue and mouse orthotopic tongue tumor models. We observed co-expression of Aldh111 with S100, a marker predominantly expressed in Schwann cells, suggesting Aldh111 expression in glial cells of the tongue. Furthermore, in Aldh111^{CreERT2}Cav2^{f/f} mice, immunohistochemical staining showed a significant reduction of Cav2 expression in neural tissues, while endothelial cells still expressed Cav2. Thus, we believe that using the Aldh111^{CreERT2}Cav2^{f/f} model is a feasible approach under current circumstances (Please see the marked revised manuscript, lines 307-314 and Supplementary Figure 5).

However, we also acknowledge the limitations of this model. The lack of more specific Cre-driver lines hinders our ability to distinctly target satellite glial cells, Schwann cells, and fibroblasts within the ganglia. Additionally, Aldh111 is widely used in CNS research as a marker for astrocytes, and using the Aldh111^{CreERT2}Cav2^{f/f} model results in Cav2 deletion not only in glial cells of the trigeminal ganglia and their nerve fibers but also in astrocytes in the CNS. This could potentially confound the interpretation of our in vivo findings, as the role of Cav2 in CNS astrocytes with respect to HNSCC remains unclear. We have addressed these limitations in the Discussion section of the revised manuscript (please see the marked revised manuscript, lines 598-608).

Despite these acknowledged limitations, we believe that our study makes a valuable contribution to the field. Previous research in cancer neuroscience has

predominantly treated peripheral ganglia as a homogeneous entity without differentiating between neurons and glial cells, or has focused solely on neurons while neglecting glial cells [7-12]. Our work represents an initial effort to explore and refine these distinctions within the constraints of current techniques, which we hope will contribute to advancing research methodologies and tools in cancer neuroscience.

In addition, we aim to further validate the effect of glial Cav2 on tumor phenotypes through in vitro experiments. The primary challenge we face is how to isolate glial cells from the ganglia. The first method we considered was using molecular markers for FACS to separate neurons from glial cells. For various immune cells, there are well-established and easy-to-implement FACS sorting strategies. However, in cancer neuroscience—a rapidly emerging field—the situation is quite different. The ganglia contain a diverse population of neuronal cells, Schwann cells, satellite cells, as well as a small number of fibroblasts and even fewer immune cells. Upon reviewing the literature, we found that studies using FACS to isolate neurons and glial cells typically rely on transgenic mice that express a fluorescent protein in neurons, rather than using specific markers. This suggests that suitable molecular markers for sorting neurons and glial cells from the TG or DRG are not readily available. Additionally, it is worth noting that the follow-up experiments in these transgenic mouse studies typically focus on sequencing the sorted neurons, rather than culturing them further. Given that neurons are notoriously fragile, the feasibility of culturing and conducting functional experiments on FACS-sorted neurons remains uncertain.

Our second approach involved breeding mTmG mice with Adv-Cre mice, where neurons express tdTomato and other cells express EGFP. Following FACS sorting, cells would be infected with either a control lentivirus or a shCav2 lentivirus. However, after the dual procedures of FACS sorting and lentiviral infection, the cell condition was poor, making subsequent functional experiments challenging.

Therefore, our third strategy was to first breed mTmG mice with Adv-Cre mice to generate mTmG^{+/+};Adv-Cre mice, and then cross these with Cav2^{+/-} mice to obtain mTmG^{+/+};Adv-Cre;Cav2^{+/-} mice. Since both the mTmG and Cav2 loci are located on chromosome 6, we could only obtain heterozygous Cav2 mice. FACS sorting of these mice yielded a population of Adv-negative cells, which were primarily non-neuronal cells (mainly glial cells). Co-culturing these glial cells with tumor cells revealed a moderate reduction in the mitochondrial superoxide levels of the tumor cells (due to the involvement of multiple genes, obtaining a large number of offspring was challenging, limiting us to only a small number of functional experiments). These experiments provided clear evidence of the impact of Cav2 in non-neuronal cells of the trigeminal ganglion on tumor cells (Please see the marked revised manuscript, lines 355-370 and Supplementary Figure 6E-F).

Based on these findings, we have made significant efforts to validate the effect of Cav2 in non-neuronal cells (mainly glial cells) from the ganglia on tumor cell function

in vitro. We would greatly appreciate any further suggestions you may have, and once again, thank you for your valuable feedback.

4. It would be more convincing if the HNSCC cancer stem maker's expression were examined in mice tumor tissues in Fig. 4 and Fig. 7, such as ALDH, CD44, SOX2, or BMI1.

Response: Thank you for your valuable and professional suggestion. Assessing the expression of cancer stem cell markers in in vivo tumor tissues indeed enhances the reliability of our findings. Following your suggestion, we performed multicolor immunohistochemical staining on the tumor tissues presented in Figure 4, focusing on stemness markers such as Aldh1, CD44, and Bmi1. As anticipated, the expression of these stemness markers was significantly higher in the orthotopic tumors of WT mice compared to those in Cav2^{-/-} mice. Interestingly, we also observed elevated expression of stemness markers in the normal epithelial tissues of WT mice, while the epithelial tissues of Cav2^{-/-} mice showed minimal expression (please see the marked revised manuscript, lines 451-462 and Figure 6K and Supplementary Figure 10).

Furthermore, we conducted similar analyses on orthotopic tumors from Adv^{Cre}Cav2^{f/f} and Aldh111^{CreERT2}Cav2^{f/f} conditional knockout mice, where stemness marker expression was also significantly reduced compared to control mice, though the differences between Adv^{Cre}Cav2^{f/f} and control mice were relatively smaller (please see

the marked revised manuscript, lines 451-462 and Figure 6K and Supplementary Figure 10).

Additionally, we examined the 4-NQO-induced tumors presented in Figure 7 using the same multicolor immunohistochemical approach. Consistently, stemness marker expression was markedly higher in 4-NQO-induced tumors from WT mice compared to those from Cav2^{-/-} mice (please see the marked revised manuscript, lines 496-500 and Figure 7U).

5. The quality of Fig. 1A could be better. Visualizing the box plot, as covered by the dots, is very hard.

Response: Thank you for your feedback. In the revised manuscript, we have changed the solid dots to hollow ones in Figure 1A to enhance the visibility of the box plot. However, due to the large sample size in some groups (such as “cN0 group”), there is still slight obscuration of the box plot. Nonetheless, the position of the box plot is generally clear.

Figure 1(response letter) The upper set of images is Figure 1A from the original manuscript, and the lower set of images is Figure 1A from the revised manuscript.

6. Fig. 3B is missing labels.

Response: We apologize for this oversight. Labels have been added to Figure 3B in the revised manuscript. Thank you for bringing this to our attention.

7. The statistical analysis should be performed in Fig. 6J.

Response: Thank you for your suggestion. We have conducted chi-squared tests for tumor formation results across different conditions (varying cell numbers) in the revised manuscript. The calculated p-values have been added to the figure.

8. Fig. 7K does not have any label.

Response: We apologize for this oversight. Labels have been added to Figure 7K in the revised manuscript.

9. Fig. 7L-7S are not labeled very clearly.

Response: Thank you for pointing out this issue. In the revised manuscript, we have replaced the simplistic labels (WT and KO) with more precise designations, $CAV2^{+/+}$ and $CAV2^{-/-}$, to enhance clarity.

In summary, we have made every effort to address all of your concerns and hope that our revisions meet with your approval. We sincerely appreciate your valuable feedback and welcome any further comments you may have.

Reviewer #2 (Remarks to the Author):

1. In this manuscript, Liu et al characterize an interesting association between caveolin-1 (CAV-2) expression in neurons and glia and the induction of mitochondrial oxidative phosphorylation in tumor cells. The authors use a combination of in vitro co-culture [tumor cells/ganglia (trigeminal or DRG)] as well as in vivo studies with transgenic animals in which Cav2 is either knocked out of neurons, glia or the whole mouse and assess the impact on disease. The authors home in on ribosomal protein S6 (a marker of mTOC1 activation) yet why they even looked at this protein is not clear. The jump

is made without any rationalization which makes it difficult to follow.

Response: Thank you very much for your positive assessment of our work. We apologize for any confusion caused regarding our focus on the phosphorylation of ribosomal protein S6 (pS6), a marker of mTORC1 activation. In Figure 4, we initially investigated the differences in tumor growth between orthotopic HNSCC tumors implanted in the tongues of WT and various transgenic mice. Upon observing significant differences in tumor growth rates between WT and Cav2^{-/-} mice, we proceeded to evaluate several fundamental phenotypic changes, including proliferation (Ki-67), apoptosis (Caspase-3), angiogenesis (CD31), nerve growth (β 3-tubulin), metabolic changes (pS6), and necrosis (HMGB1).

Given that metabolic alterations in tumors are often associated with varying degrees of mTORC1 activation, we used pS6 as a general marker to assess metabolic changes in the tumors. Although pS6 is limited in its ability to fully capture the metabolic state of tumor cells by merely reflecting mTORC1 activity, we employed it here as an initial phenotypic assessment of orthotopic tumors. Subsequently, in Figure 5, through co-culture experiments, sequencing, and GSEA analysis, we discovered that Cav2 expressed in nerves may influence oxidative phosphorylation (OXPHOS) in tumor cells, a more specific metabolic phenotype.

We acknowledge that, after observing this phenomenon, we primarily conducted

in vitro experiments to confirm our findings, without evaluating OXPHOS-related phenotypes in in vivo models. This omission may have contributed to the perceived lack of logical flow in the initial submission. To address this concern, we have now assessed the expression of OXPHOS-related markers (SDHB, UQCRC, ATP5A1, NDUFB8) in the tumors presented in Figure 4. Our results indicate that OXPHOS marker expression was overall higher in orthotopic tumors from WT mice compared to Cav2^{-/-} mice, with similar trends observed between control mice and Aldh111^{CreERT2}Cav2^{f/f} or Adv^{Cre}Cav2^{f/f} mice. However, the difference between control mice and AdvCreCav2^{f/f} mice was relatively smaller (please see the marked revised manuscript, lines 379-386 and Supplementary Figure 7A).

We hope that the revised manuscript provides a more coherent narrative. If you have any further suggestions or concerns, we would be grateful for your guidance.

2. While this type of neuronal/glia influence on tumor cells is novel, the mechanism is not delineated. Moreover, a study in 2012 (FASEB J PMID: 22859397) demonstrated that caveolae can mediate the transfer of caveolin into mitochondria and that this is a stress response that alters mitochondrial function. It seems that something similar might be at the root of the findings in this study with the initiation of transfer of caveolin-2 occurring from tumor-infiltrating nerves (or their associated glial cells). If so, neuronal transfer of CAV-2 into tumor cells and the subsequent trafficking to mitochondria would be very novel. However, the mechanism is not defined. A simple way to assess this

mechanism would be to fluorescently tag-CAV2 in neurons and co-culture with tumor cells. Is the fluorescently tagged CAV2 now found in tumor cell mitochondria? Can this be blocked by inhibiting endocytosis?

Response: Thank you very much for pointing out the gaps in our understanding of the mechanism underlying the neuronal/glial influence on tumor cells. We acknowledge that our original manuscript did not sufficiently elaborate on how neural CAV2 influences tumor cell behavior, and we are grateful for your constructive suggestion, including the relevant literature (PMID: 22859397). After reviewing the study, we hypothesize that you might have intended to reference PMID: 22859372, as PMID: 22859397 pertains to a clinical study in colorectal cancer. We appreciate the detailed experimental approach you suggested, which has greatly informed our subsequent research.

Based on your recommendation, we constructed an EGFP-Cav2 plasmid and corresponding lentivirus, which we then used to infect cells derived from the dissociated trigeminal ganglia. These cells were subsequently co-cultured with tumor cells. Unfortunately, we did not observe any green fluorescence in the tumor cells, indicating that neural Cav2 was not transferred to tumor cell mitochondria.

To further investigate the underlying mechanisms, we performed high-resolution confocal imaging to explore the localization of Cav2 within neurons and non-neuronal

cells of the trigeminal ganglia. Our results indicate that Cav2 is primarily localized to the Golgi apparatus (see revised manuscript, Figure 8). Additionally, upon co-culture with tumor cells, a subset of neurons exhibited Cav2 expression along the entire axon, although this observation was limited to a small proportion of cells, with the majority still showing Golgi-localized Cav2. Based on these observations, we hypothesize that Cav2 might be involved in protein modification, processing, and secretion within neural cells.

To further substantiate our hypothesis, we co-cultured trigeminal ganglia from Cav2^{+/+} or Cav2^{-/-} mice with SCC15 (HNSCC) cells for 48 hours, subsequently removed the tumor cells, replaced the medium, and after 6 hours, collected the conditioned medium from the trigeminal ganglia. We found that protein concentrations in the conditioned medium from Cav2^{+/+} ganglia were significantly higher than those from Cav2^{-/-} ganglia, suggesting that neural Cav2 may indeed play a role in protein processing and secretion.

To characterize these secreted proteins, we used a similar co-culture setup and performed quantitative proteomic analysis on the conditioned medium. We focused on extracellular proteins, as they are more likely to influence other cells, such as tumor cells. Our analysis identified differential expression of proteins primarily associated with neural growth and development, metabolism, and protein processing and vesicle trafficking. Specifically, in relation to cellular metabolism, and particularly oxidative

phosphorylation (OXPHOS), we identified numerous proteins potentially involved in mitochondrial metabolism, including CYB5A (cytochrome b5) and CYB5R3 (cytochrome b5 reductase), which participate in electron transport and may influence OXPHOS; ATP5PB, a subunit of ATP synthase that directly contributes to OXPHOS; ETFDH, involved in mitochondrial fatty acid oxidation and OXPHOS; MICU3, a mitochondrial calcium regulator that influences OXPHOS function; MIGA2, a regulator of mitochondrial dynamics; TRIAP1, involved in mitochondrial protein folding and oxidative stress response; and VPS13D, implicated in mitochondrial quality control and dynamics. Additionally, we identified numerous proteins related to carbohydrate, lipid, and amino acid metabolism, all of which can impact OXPHOS levels (please see the marked revised manuscript, lines 504-555 and Figure 8).

These findings suggest that neural CAV2 orchestrates tumor cell OXPHOS by modulating the extracellular secretion of a multitude of metabolism-related proteins rather than merely regulating the secretion of a single factor or pathway. However, we acknowledge that the precise mechanisms require further investigation, such as identifying which of these secreted proteins are internalized by tumor cells, quantifying their uptake, and elucidating their specific roles in modulating OXPHOS. While these questions extend beyond the scope of the present study, we hope you can appreciate our efforts to provide preliminary insights and refine our experimental approach accordingly. We look forward to any further guidance or suggestions you may have.

3. The authors utilized Adv-cre::Cav2-f/f to deplete CAV2 from sensory neurons which is a good strategy. However, they do not validate that KO of CAV2 does not alter the number of Advilin positive nerves. They find a decrease in tumor volume in Adv-cre::Cav2-f/f mice but if the number of Advilin positive nerves is decreased in these animals, there would be decreased tumor innervation and subsequently decreased tumor growth. This would have nothing to do with the absence of CAV2 in the nerves but rather an absence of nerves themselves. This was not shown and needs to be demonstrated. If there is no impact of loss of CAV2 on the nerve population, then the decreased tumor growth is mediated by their absence of CAV2 but this has not been shown with the current data.

Response: Thank you very much for your insightful comment, which pointed out an important aspect that was not adequately addressed in our original manuscript. Following your suggestion, we quantified the number of nerve fibers (including smaller nerve fibers) in the tumor-bearing regions of both Adv^{Cre}Cav2^{f/f} and control Cav2^{f/f} mice. Our analysis revealed no significant difference in the number of nerve fibers between the two groups (please see Supplementary Figure 4E).

However, as shown in Figure 4E, when evaluating nerve distribution using an alternative method (i.e., the area of β 3-tubulin-positive regions), we found that the positive area in Adv^{Cre}Cav2^{f/f} mice was slightly smaller compared to control Cav2^{f/f} mice, although the difference was relatively minor. As indicated in Figure 3, our data

demonstrate that neural Cav2 has a significant impact on tumor-nerve interactions, with the knockout of neural Cav2 markedly inhibiting tumor-associated nerve regeneration. Therefore, in addressing the question of whether decreased tumor growth is mediated by the absence of Cav2 in nerves or by a reduction in the number of nerves themselves, it is difficult to completely eliminate confounding factors.

We appreciate your valuable suggestion and welcome any further recommendations you may have to refine our approach. Once again, we are grateful for your constructive feedback.

4. Similarly, the Aldh111-CreERT2 mouse is crossed with the Cav2-f/f to KO CAV2 expression in glial cells. However, they do not show that glial cells in the periphery have been eliminated. These simple controls must be included.

Response: Thank you very much for raising this important concern. The Advillin-Cre model has been widely used for gene knockout in trigeminal ganglion (TG) and dorsal root ganglion (DRG) neurons. However, glial cells in the DRG and TG differ from those in the central nervous system (CNS). Currently, there are well-established Cre driver lines available for targeting various types of glial cells in the CNS, but suitable Cre lines for specifically targeting glial cells in the peripheral ganglia—primarily satellite glial cells and Schwann cells—are lacking. Based on previous reports that Aldh111 is expressed in the glial cells of peripheral ganglia[6], we employed the Aldh111-CreERT2

system.

In response to your and other reviewers' suggestions, we performed multicolor immunohistochemical staining on human HNSCC tissues and tissue sections from the mouse tongue orthotopic tumor model. Our analysis revealed co-expression of Aldh111 and S100 (a marker predominantly expressed in Schwann cells), indicating that Aldh111 is expressed in the glial cells of the tongue. Additionally, immunohistochemical staining in Aldh111^{CreERT2}Cav2^{f/f} mice showed a significant reduction of Cav2 expression in nerves, while endothelial cells still expressed Cav2, suggesting that our model is relatively effective for targeting glial cells under the current experimental conditions (Please see the marked revised manuscript, lines 307-315 and Supplementary Figure 5).

However, we must also acknowledge the limitations of this approach. First, there is a lack of specific Cre lines to distinguish between satellite cells, Schwann cells, and fibroblasts in peripheral ganglia. Moreover, Aldh111 is widely used in CNS studies as a marker for astrocytes, meaning that the Aldh111^{CreERT2}Cav2^{f/f} model not only knocks out CAV2 in glial cells of the trigeminal ganglia and their nerve fibers but also affects astrocytes in the CNS, which may confound the interpretation of in vivo results (as the impact of astrocytic CAV2 on HNSCC remains unclear). We have discussed these limitations thoroughly in the revised manuscript (please see the marked revised manuscript, lines 598-608).

We appreciate your insightful comments and welcome any further suggestions you may have to improve our study. Thank you once again for your constructive feedback.

5. The authors mention the paper by Zhang et al in which CGRP-expressing nerves alter tumor cell metabolism but argue it is via a different mechanism than demonstrated here. If CAV2 expression is within sensory nerves, it would be unlikely that they are not also CALCA (CGRP gene) expressing. This would be simple to determine by doing CAV2/CGRP double staining alongside, beta-III tubulin/CAV2 and beta-III tubulin/CGRP staining.

Response: Thank you very much for your insightful suggestion. You raised an important point regarding the potential co-expression of CALCA (CGRP) and CAV2 in sensory nerves, which requires further clarification. Based on single-nucleus sequencing of the trigeminal ganglion, we found that the expression patterns of CALCA (CGRP) and CAV2 in neurons are indeed distinct, with some neurons expressing CALCA but not CAV2, and vice versa. This suggests that not all CGRP-expressing neurons also express CAV2.

Following your recommendation, we performed multicolor immunohistochemical staining with S100 to label nerve fibers, and CALCA and CAV2 to mark the respective proteins. Our results demonstrated that CALCA and CAV2 are not consistently co-expressed in nerves, as shown in the revised Figure 4. Specifically, in case 1, we

observed nerve fibers expressing CAV2 but lacking CALCA, while in case 2, we observed nerve fibers expressing CALCA without CAV2. We have included these results in the revised manuscript (please see the marked revised manuscript, lines 398-403 and Supplementary Figure 8C-D).

Furthermore, even though some neurons may express both CALCA and CAV2, it is important to emphasize that our primary findings were derived from specifically knocking out the CAV2 gene, rather than employing genetic ablation techniques such as DTA to eliminate these neurons. Therefore, our data indicate that while both neuronal CGRP and CAV2 can influence tumor cell metabolism, they likely do so through distinct mechanisms. We appreciate your suggestion, as it has helped to better delineate these differences in our study.

Thank you once again for your constructive feedback, and we welcome any further suggestions you may have.

6.DRG and TGM are isolated from juvenile mice (age 2-4 weeks). Why so young? HNSCC is an adult disease. Why not do experiments with ganglia isolated from adult mice? What is the rationale?

Response: Thank you for raising this insightful question, which reflects a common concern in the field. Indeed, this issue puzzled us as well when we first began exploring

tumor-neural interactions. Upon reviewing the literature, we found that most studies in this area employed very young animals, such as juvenile rats or mice (2-4 weeks old) [8-10, 13-17], and some even used fetal specimens [11]. There are also several studies that did not specify the age of the animals used [12, 18-20] (please refer to the table below for details).

Initially, we were unsure why such young animals were chosen for ganglia isolation. It was only after conducting our own experiments that we realized the substantial advantages of using juvenile neural tissues. Specifically, the regenerative and adaptive capacities of ganglia from juvenile mice are significantly higher, making them more amenable to manipulation and in vitro culture. In contrast, ganglia from older mice exhibit slow growth ex vivo, posing challenges to maintaining viable experimental conditions. Furthermore, ganglia from younger animals display greater plasticity, allowing them to adapt and respond more effectively to experimental manipulations.

Additionally, younger animals are generally in better overall health, which helps provide stable and consistent experimental conditions, thereby reducing variability due to health issues that may arise in older animals. Given these factors, we opted to use ganglia from juvenile mice to ensure reliable and reproducible outcomes. We hope this explanation clarifies our rationale, and we welcome any further questions or suggestions you may have.

Origin of ganglia	Publication
DRG from BALB/c nu/nu mice 2 to 4 weeks old	Nature 2020[9]
Primary sensory neurons were isolated from TG dissected from 6- to 8-week-old mice as previously described	
Rat Neonatal Dorsal Root Ganglion (Lonza;R-DRG-505)	Cell 2020[10]
Embryos (E14.5) from timed pregnant C57BL/6 mice were utilized for DRG isolation.	Cancer cell 2018[11]
Primary DRG cells were collected from postnatal day 1 to 7 mice.	Gastroenterology2019[13]
DRG were dissected from postnatal day-30 Sprague–Dawley rats.	Nat Commun 2015[14]
DRG neurons isolated from 4- to 7-week-old C57BL/6 mice were cultured	Gastroenterology2020[8]
DRG from a 9-week rat	Clin Cancer Res2022[15]
DRG from newborn male Balb/c mice were isolated	J Natl Cancer Inst 2010[16]
Dorsal root ganglia (DRG) from 4- to 8-week-old C57BL/6J mice were harvested	Cancer Res 2017[17]

7. Retrograde labeling of nerves in the tongue is performed BEFORE tumors are present. This will label resident nerves, not necessarily nerves that sprout and newly infiltrate the tumor once the tumor grows. These newly sprouted nerves are the ones of interest and they are not being tagged.

Response: Thank you for your insightful question, which addresses a critical aspect of our study that had been previously overlooked. Indeed, as you correctly pointed out, we performed retrograde labeling of nerves prior to tumor implantation, which resulted in labeling resident nerves rather than those newly sprouting and infiltrating the tumor after its growth. This limitation meant that the newly sprouted nerves, which are of particular interest in the context of tumor progression, were not tagged.

Given the rapid growth rate of MOC2 xenografts, performing retrograde labeling at an appropriate time point was challenging, as effective labeling requires sufficient time, and the mice often reached their endpoint tolerance before this could be achieved. In response to your suggestion, we have redesigned our experiments using the slower-growing MOC1 cell line. We implanted these cells and subsequently, after 7 days, injected DiI to perform retrograde labeling. We then assessed the expression of Cav2 in neurons, and these results are now presented in the revised manuscript (Please see the marked revised manuscript, lines 197-205, and Figure 2I).

We hope this revised experimental approach addresses your concern, and we welcome any further feedback or suggestions you may have to improve the robustness of our study. Thank you once again for your valuable feedback.

8.How was neural CAV2 expression defined? Was there co-staining for beta-III tubulin

and CAV2? (page 7, line 153).

Response: Thank you for your insightful question regarding the definition of neural CAV2 expression. As stated in the legend of Figure 1, neural CAV2 expression levels were determined by immunohistochemical staining scores. These scores integrated both staining intensity and the extent of staining within neural structures, with intensity ranging from 0 (negative) to 3 (strongly positive). The final score was calculated by multiplying the staining intensity score by the proportion of the stained neural area, resulting in a comprehensive score ranging from 0 to 300. The neural context was delineated using S100 staining, and within this context, CAV2 intensity and the corresponding area percentages were assessed. Based on these CAV2 expression scores, patients were categorized into high-score and low-score groups.

Both S100 and β 3-tubulin are commonly used as markers for neural tissue, with β 3-tubulin primarily labeling neurons and S100 predominantly marking glial cells, especially Schwann cells. In our study, we utilized both markers to ensure accurate identification of neural structures. Specifically, for Figure 1J, we attempted to stain neural structures using both S100 and β 3-tubulin in HNSCC patient sections. We found that S100 provided superior specificity and sensitivity, allowing for clearer delineation of neural tissues at the tissue level, which facilitated a more reliable assessment of neural CAV2 expression.

We hope this response clarifies our methodology, and we welcome any further suggestions you may have. Thank you once again for your valuable feedback.

9. There is a consistent utilization of trigeminal and DRG ganglia but it is unclear why both are used. The disease being studied is HNSCC so the logical ganglia is the TGM. Why use DRG at all? This is not explained.

Response: Thank you for your insightful and pertinent question. The use of dorsal root ganglia (DRG) alongside trigeminal ganglia (TG) in our study was motivated by several considerations. Firstly, the availability of TG from each transgenic mouse is quite limited compared to DRG, which are more abundant. Given the large number of transgenic mice used in our studies, TG samples are even more restricted compared to those from wild-type mice. Secondly, although there are some differential gene expressions between TG and DRG, their overall expression patterns are remarkably similar[21-23], making DRG a suitable substitute for TG. This similarity may also explain why DRG has been utilized in other studies investigating neural interactions in HNSCC [9, 14, 15]. Thirdly, while our study predominantly focuses on TG, DRG were included in some experiments to maximize resource efficiency and prevent wastage of ganglia, serving as a supplementary group. For instance, as illustrated in Figure 3A and Supplementary Figure 4, the results from DRG not only corroborate our main findings from TG but also suggest that neural-expressed CAV2 might play a role in tumor progression beyond HNSCC, potentially extending to other cancers such as pancreatic

cancer. Although this hypothesis requires further verification with pancreatic cancer cell and in vivo models, our findings preliminarily suggest that CAV2 expressed in DRG may also influence tumor progression.

We hope this explanation clarifies our rationale for including DRG in our study, and we welcome any further questions or suggestions you may have. Thank you once again for your valuable feedback.

10. The sex of mice used in these experiments is not mentioned. If both sexes were utilized, were any sex-differences identified?

Response: Thank you for your insightful question regarding the sex of the mice used in our experiments. In this study, the mice were primarily transgenic animals bred within our institution, and the sex of the mice used depended on the genders of the offspring produced from breeding. We did not observe any phenotypic differences between male and female mice in our experiments. In the revised manuscript, we have explicitly indicated the sex of the mice used in Figure 4C, 4I, and 4O. We appreciate your attention to this detail and hope that this additional information will enhance the clarity of our experimental methods. Thank you once again for your valuable feedback.

11. There needs to be a reference for the statement that HNSCC HPV negative patients have a heightened predisposition to PNI (page 6, line 130).

Response: Thank you for pointing out this issue. In the revised manuscript, we have cited a previously published article of ours[24], wherein Figure 2 demonstrates that HPV-negative HNSCC patients are more susceptible to PNI.

Figures:

12. Figure 1F is too small is very difficult to see anything. This is true of several figures: Fig 6H, Fig 1G, Fig 1J, Fig 4O, Fig 4N, Supplemental figure 1F

Response: Thank you for highlighting an important issue regarding the size of the images in our manuscript. Indeed, the dimensions of the images are constrained by the layout limits, which has inadvertently led to difficulties in viewing the details when enlarged. While the images captured using our confocal microscope are initially clear, we acknowledge that their resolution significantly decreases during the conversion process to the PDF format for review purposes, for which we sincerely apologize.

To mitigate this issue, we have included all original fluorescence and immunohistochemistry images in the supplementary materials of the revised manuscript for detailed examination. Additionally, should the manuscript be accepted, we intend to collaborate with the publisher's staff to explore ways to preserve the original resolution of the images in the final publication as effectively as possible. We appreciate your attention to this matter and are committed to improving the presentation of our data. Thank you once again for your valuable feedback.

13.Fig 1J: why isn't the normal mucosa innervated? It would be helpful to add beta-III tubulin (pan neuronal marker) to determine if all tumor-infiltrating nerves are CAV2 positive. An H&E panel would also be helpful.

Response: Thank you for your insightful comment. In our analysis, we observed that the number of nerves in the normal mucosa was very limited, primarily because we only considered the distribution of nerves within the epithelium and did not include the nerves present in the tongue muscle. Compared to the diseased tissue, the normal epithelium is relatively "thin", and we indeed found very few Trpv1-positive areas in the normal epithelium.

Following your suggestion, we supplemented our analysis with an additional set of multicolor immunohistochemical staining using β 3-tubulin, Trpv1, and Cav2. The results were consistent with our previous findings (please see the marked revised manuscript, lines 210-214, and Supplementary Figure 3A). Not all tumor-infiltrating nerves were positive for Cav2. As shown in the figure below, white arrows indicate Cav2-positive nerves, whereas orange arrows point to Cav2-negative nerves. Moreover, we found a high degree of overlap between the β 3-tubulin and Trpv1 staining regions, suggesting that the majority of the nerves in the tongue are sensory nerves.

Additionally, while we do have corresponding H&E images, they were not

included in the original manuscript because we found that H&E staining is effective in identifying larger nerves but is less reliable for detecting smaller nerve fibers. We hope this additional analysis addresses your concerns, and we appreciate your valuable feedback. Thank you once again for your constructive suggestions.

Figure 2(response letter) The upper image shows β 3-tubulin staining, while the lower image shows TRPV1 staining. Scale: 200 μ m.

Figure 3(response letter) The white arrow indicates Cav2⁺ neurons, while the orange arrows point to Cav2⁻ neurons. (red: Cav2; yellow: β3-tubulin; green: TRPV1)

14.In figure 1I, there is a patient population labeled “TJCH cohort” but what this cohort is was not defined.

Response: Thank you for pointing out the omission regarding the definition of the "TJCH cohort." We apologize for any confusion this may have caused. In the revised manuscript, we have clarified in both the figure legend and the Methods section that "TJCH cohort" refers to the Tianjin Medical University Cancer Institute & Hospital cohort. We appreciate your attention to this detail, and we hope that this clarification will enhance the clarity of our manuscript. Thank you once again for your valuable feedback.

15.Fig 1E and F: please add arrows to point out caveolae.

Response: Thank you for your comments. I believe you are referring to Figures 2E and 2F. In the original manuscript, we had already included arrows to indicate the caveolae structures. To enhance clarity, we have increased the size of these arrows and adjusted their color in the revised manuscript. Given the abundance of caveolae in some images, we opted to use a representative arrow in each panel to point out the caveolae, avoiding overcrowding and ensuring a clear visual presentation.

We appreciate your attention to this detail and hope that these adjustments meet your expectations. Thank you once again for your valuable feedback.

16.Fig 2A: would be easy to do densitometry to look at the differences; this would allow for statistical analysis as well.

Response: Thank you for your insightful suggestion. In the original manuscript, the Western blot images provided had relatively high contrast, which was not fully compliant with the publisher's requirements. In the revised manuscript, we repeated this experiment and, following your recommendation, included the densitometry results along with molecular weight markers and uncropped bands in the supplementary materials. We appreciate your attention to this detail, and we hope that the added quantitative analysis will enhance the rigor and clarity of our findings.

17.Fig 2D: are these glial cells that migrated off the ganglia? They look like fibroblasts.

Staining with a glial marker would be helpful. If these are glia that migrated off ganglia, how many days did this take? The time could influence expression of CAV2.

Response: Thank you for your question. The cells in Figure 2D are not cells that migrated from the ganglia. In this experiment, we enzymatically digested the ganglia into dispersed cells, which were then co-cultured with tumor cells or cultured independently, followed by immunofluorescence staining for Cav2. In the revised manuscript, we have replaced the term "glial cells" with "non-neuronal cells" because we observed that the non-neuronal cells within the ganglia universally expressed CAV2, regardless of whether they were Schwann cells, satellite cells, or fibroblasts. This observation is consistent with the single-nucleus sequencing results of trigeminal ganglia (please see the marked revised manuscript, lines 173-179, and Supplementary Figure 2). Additionally, there are currently no markers with high specificity for glial cells in the trigeminal ganglia, which is why we did not perform staining with a glial marker.

We appreciate your attention to this distinction and have updated the manuscript accordingly to ensure clarity and accuracy. Thank you once again for your valuable feedback.

18.Fig 3A: what is the difference between TGM and DRG that leads to the difference in migration?

Response: Thank you for your question. Although the expression profiles of the trigeminal ganglia (TG) and dorsal root ganglia (DRG) are quite similar[21-23] , there are significant differences in tissue volume between them, with TG being larger than DRG. As a result, direct comparisons between TG and DRG are not feasible due to these inherent differences in tissue volume. However, comparisons within the same type of ganglia remain valid and informative. For instance, in the CAV2^{+/+} TG group and the CAV2^{-/-} TG group, the tissue volume of TG is consistent, providing a basis for comparability. Similarly, in the CAV2^{+/+} DRG group and the CAV2^{-/-} DRG group, the tissue volume of DRG is consistent, ensuring that the comparisons made between these groups are meaningful. We appreciate your attention to this detail, and we hope this explanation clarifies our approach. Thank you once again for your valuable feedback.

19.Fig 3B: I'm not sure what I'm supposed to be seeing here. Arrows would be helpful.

Response: Thank you for your suggestion. As shown in revised Figure 3B, arrows have now been added to indicate the direction of tumor invasion towards the nerves and the length of the outgrowth neurites.

20.Fig 3C: this needs to be brighter to make the point.

Response: Thank you for your suggestion. As shown in revised Figure 3C and

subsequent images, we have brightened all pictures according to a uniform standard.

Figure 4(response letter) The upper set of images is Figure 3C from the original manuscript, and the lower set of images is Figure 3C from the revised manuscript (all images have been uniformly brightened). Scale: 100 μ m.

21.Fig 4E: looks like a lot of non-specific staining. Are these fibroblasts? Tumor cells?

Double IHC or IF would help define these CAV2 expressing cells.

Response: We sincerely apologize for the reduced image resolution caused by the file conversion process. In the original images, the cell morphology is more distinguishable, allowing for better differentiation between tumor cells, adjacent nerves, and stromal

components. To address this, we have shown a high-resolution immunohistochemical image of pS6 for this specific case below. Additionally, H&E staining of the corresponding mouse tissue reveals a clear demarcation between the MOC2 xenograft and the surrounding host tissue.

Following your suggestion, we also performed multiplex immunohistochemical staining for PanCK and pS6 on sections from this mouse. However, PanCK did not distinctly label the xenograft as it did in the HNSCC patient sections (e.g., Figure 1). This discrepancy may be attributed to several factors: First, in the context of immunohistochemical analysis of HNSCC xenografts co-existing with normal epithelial tissue, the presence of normal epithelial cells, which also express cytokeratins, can lead to increased background staining. This elevated background may obscure the specific signal from tumor cells, making it more challenging to discern and accurately assess tumor-specific staining. Second, structural differences between normal and tumor cells may result in variations in cytokeratin distribution and abundance, leading to differences in staining intensity. Such differences can impact the clarity of observation and make tumor-specific cytokeratin expression less apparent. Lastly, when both normal epithelial and tumor cells strongly express pan-cytokeratin, competitive binding for the antibody may occur, reducing the effective labeling of target proteins in tumor cells. This competitive binding can hinder the ability to clearly distinguish tumor cells from the surrounding normal epithelial background during staining.

We appreciate your insightful suggestions, and we hope that these additional clarifications, along with the improved images, provide a more comprehensive understanding of our analysis. Thank you once again for your valuable feedback.

Figure 5(response letter) The large-scale image of Figure 4E that you have pointed out as problematic. IHC representation for pS6 in tumors from wild-type MOC2-engrafted mouse. Scale: 100 μ m

Figure 6(response letter) H&E staining image of a section from the same mouse. Scale: 100 μ m.

Figure 7(response letter) Multiplex immunohistochemical staining results of a section from the same mouse, where red indicates PanCK and yellow indicates pS6. Scale: 200 μ m.

22.Fig 6A: what % of cells are ALDH1+ cells?

Response: As shown in the revised Figure 6, approximately 0.55% of the cells are categorized as ALDH1⁺ cells.

23.Fig 6E: it is very difficult to interpret much from this figure. Too small, too dim. Cav2 and CD44 colors too similar.

Response: Thank you very much for pointing out this issue, as it may significantly affect the quality of the manuscript. To avoid overcrowding in the images, we have

replaced the staining panel and now include only CAV2 and CD44 to improve the clarity of the figures. We found that CD44 is more specific for marking tumor stem cells compared to ALDH1A1, which also shows strong positivity in vascular structures. We appreciate your attention to this detail and hope that these adjustments enhance the interpretability and quality of the figure. Thank you once again for your valuable feedback.

24. Supplemental Figure 2: needs to be brighter to make the point.

Response: Thank you for your suggestion. In the revised manuscript, we have brightened all the images according to a uniform standard.

25. Supplemental Figure 4E: needs to be brighter. When printed, the panels look all the same.

Response: Thank you for your suggestion. As shown in revised Supplementary Figure 4E and subsequent images, we have brightened all pictures according to a uniform standard.

Figure 8(response letter) The upper set of images is Supplementary Figure 4E from the original manuscript, and the lower set of images is Supplementary Figure 4E from the revised manuscript (all images have been uniformly brightened).

In summary, we have made every effort to address all of your concerns and hope that our revisions meet with your approval. We sincerely appreciate your valuable feedback and welcome any further comments you may have.

Reviewer #3 (Remarks to the Author):

The authors showed an interesting correlation between CAV2 expression and HSCC prognosis. The increased expression of CAV2 expression in HSCC patients with PNI together with the predominant expression of CAV2 in neural structures suggest the involvement of CAV2 in nerve-cancer crosstalk. The ability of cancer cells to induce CAV2 expression was demonstrated in vitro, by co-culturing TG or TG cells with

cancer cells, and in vivo, by performing xenograft and a 4-NQO-induced murine HNSCC model.

Interestingly, depletion of CAV2 expression in sensory neurons reduced tumor growth and changed the metabolic profile of the cancer cells. Based on these observations, the authors conclude that “CAV2-expressing Nerves induce Metabolic Switch toward Mitochondrial Oxidative Phosphorylation to Promote Cancer Stemness”.

This is a potentially interesting paper. Unfortunately, the manuscript is very descriptive and mechanistic insights are lacking. I feel that there is something of an opportunity missed to deeply explore the role of CAV2 in tumorigenesis, that will allow important follow up studies relating to therapeutic approaches.

The study brings new observations but the actual novelty on the underlying mechanism seems still limited.

Response: Thank you very much for your positive feedback. We acknowledge that the lack of in-depth mechanistic studies, as pointed out by you and other reviewers, is indeed a limitation of our research. In the revised manuscript, we have made every effort to further explore the underlying mechanisms. Please find the details below. Thank you once again for your valuable feedback.

Major concerns:

1.(1) The authors did not elucidate the mechanisms linking Cav-2 expression by nerves to the reduced tumor growth, metabolic rewiring of the cancer cells, and cancer stemness. Although these are interesting observations, it is not possible to understand how a protein present in caveolae can affect all these processes. More specifically:

2.- Which cellular processes are affected by Cav-2 loss in nerves and how they could be relevant for the described mechanism?

3.-Which is the cancer-signal that induce Cav-2 expression in nerves?

4.-Why/How Cav-2 overexpression should promote cancer cells migrations?

5.-Why/How Cav-2 overexpression should affect neurogenesis?

6.-Why/How Cav-2 expression in nerves induce the metabolic rewiring?

Understanding these mechanisms could have important clinical implications.

Response: Thank you very much for your question. We have conducted a series of investigations to explore the mechanisms by which neural CAV2 influences tumor cell metabolism and other phenotypes. As you mentioned, understanding how a protein present in caveolae can exert such significant effects on various phenotypes is indeed

challenging. Previous studies on tumor-nerve interactions have often focused on secreted proteins such as neuropeptides, neurotrophic factors, and axon guidance molecules, which can activate downstream signaling pathways in tumor or nerve cells via autocrine or paracrine mechanisms—an approach that is more straightforward to investigate.

In our study, we have begun to elucidate two major aspects of the mechanisms involved: (1) how neural CAV2 influences tumor cell metabolism, and (2) how neural CAV2 promotes tumor-associated neurogenesis.

1. Mechanisms of Neural CAV2 in Tumor Cell Metabolism:

To explore the mechanism by which neural CAV2 affects tumor cell metabolism, we followed the suggestion from Reviewer #2 regarding the potential role of CAV2 in mediating mitochondrial transfer to tumor cells. Unfortunately, our initial investigations did not yield positive results in this regard.

We then further investigated the specific localization of Cav2 in both neuronal and non-neuronal cells within the trigeminal ganglia. High-resolution confocal imaging revealed that Cav2 primarily localizes to the Golgi apparatus in both neuronal and non-neuronal cells (see Figure 8 in the revised manuscript). Additionally, we observed that upon co-culture with tumor cells, Cav2 expression extended along the axons in a subset

of neurons, although this was observed in a small fraction of cells, with most still exhibiting Golgi-localized Cav2. Based on these observations, we hypothesized that Cav2 might be involved in protein modification, processing, and secretion.

We proceeded by co-culturing trigeminal ganglia from Cav2^{+/+} or Cav2^{-/-} mice with SCC15 cells (HNSCC cells) for 48 hours, followed by removal of the tumor cells and changing the culture medium. After an additional 6 hours, we collected the conditioned medium from Cav2^{+/+} and Cav2^{-/-} trigeminal ganglia and measured protein concentration levels. The conditioned medium from Cav2^{+/+} ganglia showed significantly higher protein concentrations compared to Cav2^{-/-} ganglia, suggesting that neural Cav2 may indeed participate in protein processing and secretion.

Subsequently, we conducted quantitative proteomic analysis of the conditioned medium from Cav2^{+/+} and Cav2^{-/-} trigeminal ganglia, focusing on extracellular proteins, as these are likely to influence other cells, such as tumor cells. We identified differentially expressed proteins mainly related to neural growth and development, metabolism, and protein processing and vesicle trafficking. Focusing on cellular metabolism, particularly oxidative phosphorylation (OXPHOS), we found numerous proteins in the conditioned medium that could potentially affect mitochondrial metabolism, including:

CYB5A (Cytochrome b5) and CYB5R3 (Cytochrome b5 reductase), which are involved in electron transport and may influence OXPHOS.

ATP5PB, a subunit of ATP synthase, directly involved in OXPHOS as a key component of the mitochondrial ATP synthase complex.

ETFDH (Electron transfer flavoprotein dehydrogenase), which is involved in mitochondrial fatty acid oxidation and OXPHOS.

MICU3 (Mitochondrial calcium uptake protein 3), which regulates calcium homeostasis and may affect mitochondrial OXPHOS function.

MIGA2 (Mitochondrial intermembrane space bridging protein 2), which regulates mitochondrial dynamics and is related to OXPHOS.

TRIAP1 (TP53-regulated inhibitor of apoptosis 1), an inner mitochondrial membrane protein involved in protein folding and oxidative stress response, indirectly participating in OXPHOS.

VPS13D, which is involved in mitochondrial quality control and influences mitochondrial health and dynamics, indirectly affecting OXPHOS.

Additionally, differentially expressed proteins included those related to carbohydrate, lipid, and amino acid metabolism, which can further impact OXPHOS. Therefore, neural CAV2 appears to orchestrate tumor cell OXPHOS by regulating the extracellular secretion of numerous metabolism-related proteins, rather than simply affecting the secretion of a single factor or pathway (please see the marked revised manuscript, lines 504-555 and Figure 8). However, further research is needed to elucidate the precise mechanisms, such as which of these proteins are internalized by tumor cells, their uptake levels, and how they specifically influence OXPHOS. These

are complex questions that may not be fully addressed in a single study, and we appreciate your understanding.

2.Mechanisms of Neural CAV2 in Tumor-Associated Neurogenesis:

The study by Ambre Spencer provided us with a potential mechanism for how neural CAV2 may promote tumor-associated neurogenesis[25]. Based on this, we found that under NGF (a neurotrophic factor highly expressed in HNSCC tissues) stimulation, the proportion of PC12 cells exhibiting neurite outgrowth was significantly higher in cells overexpressing CAV2 compared to controls, whereas overexpression of CAV1 led to a significant reduction in neurite outgrowth.

Ambre Spencer's study suggested that Caveolin-1 inhibits NGF receptor (TrkA and p75NTR) internalization and downstream pathway activation by sequestering these receptors in lipid rafts. In contrast, our experimental results indicate that CAV2 slightly promotes NGF receptor internalization and subsequent activation of downstream signaling pathways (Please see the marked revised manuscript, lines 238-249, and Figure 3D-F). This may represent a potential mechanism by which neural CAV2 promotes tumor-associated neurite outgrowth. Additionally, our aforementioned proteomic analysis of the conditioned medium from Cav2^{+/+} and Cav2^{-/-} trigeminal ganglia co-cultured with tumor cells revealed many differentially expressed proteins involved in neural growth and development, which undoubtedly influence neurogenesis

and neurite outgrowth.

In summary, we have made every effort to elucidate the mechanisms underlying our observations, and we hope these findings address most of your questions. However, regarding the question of "Which is the cancer signal that induces Cav-2 expression in nerves?" we do not currently have a clear answer. We have reviewed the literature and found no prior reports identifying specific signaling pathways that regulate Cav2 expression. We hope to investigate this further in future studies and appreciate your understanding. If you have any further suggestions, we would greatly value your input.

Thank you once again for your constructive feedback.

7.It is more than surprising to find no discussion about the known functions of Cav-2 and the potential role of Cav-2 in the described mechanism.

Response: Thank you very much for pointing out this important omission. As mentioned earlier, we have conducted additional investigations to explore the mechanisms by which neural CAV2 influences tumor cell metabolism and other phenotypes in the revised manuscript. Additionally, following your suggestion and based on the known roles of CAV2 as reported in the literature, we have expanded the discussion section of the revised manuscript to include these insights (please see the marked revised manuscript, lines 616-637). We believe this addition provides a more

comprehensive understanding of the potential role of CAV2 in the described mechanisms. We appreciate your attention to this matter, and we hope these revisions meet your expectations. Thank you once again for your valuable feedback.

8. Based on these observations the authors cannot state “This interplay between cancer cells, neurons, and glial cells unveils a mechanism through which tumor-associated nerves contribute to cancer stemness via metabolic reprogramming, presenting a novel avenue for anticancer therapy” (Lines 59-60). In the absence of experimental data supporting the mechanisms underlying the reported observations, the conclusion should be softened.

Response: Thank you very much for your insightful suggestion. Indeed, without a thorough elucidation of the underlying mechanisms, it is prudent to moderate our conclusions. Therefore, even though we have identified some underlying mechanisms in the revised manuscript, we have revised the statement to read: "This interplay observed between cancer cells, neurons, and glial cells suggests a potential mechanism through which tumor-associated nerves might influence cancer stemness via metabolic reprogramming. This highlights a possible new direction for anticancer therapy that warrants further investigation." (Please see marked revised manuscript, lines 61-64). If you have any additional suggestions, we would greatly appreciate your guidance.

9. Also, the “novel avenue for anticancer therapy” is very speculative. Is it feasible to

pharmacologically target Cav-2? This should also be discussed.

Response: Thank you very much for your constructive suggestion. Your question is indeed pertinent, as the ability to pharmacologically target CAV2 is crucial for the rapid translation of laboratory developments into clinical applications. We have explored this issue in the discussion section of our revised manuscript (please see the marked revised manuscript, lines 670-682). If you have any further suggestions, we would greatly appreciate your expert guidance.

10.(2) The authors showed reduced tumor growth in Cav2^{-/-} animals. Surprisingly, deletion of Cav2 only in sensory neurons had a modest impact on tumor growth whereas the deletion of Cav2 in glial cells strongly suppressed tumor growth (Lanes 250-254). Although the authors themselves concluded that Cav2 expression in glial cells play a major role in tumor growth (Lane 264), in the next paragraphs the authors continued to study the impact of nerve-Cav-2 on tumor growth. Considering the predominant role of CAV2 ablation in glia cells, it is difficult to understand the reasons why the authors decided to focus on the role of Cav-2 in nerves instead of the role of Cav-2 in glial cells on cancer cell metabolism.

Response: Thank you very much for raising this insightful question. You highlighted an important aspect of our findings regarding the differential impact of conditional deletion of Cav2 in glial cells and neurons on tumor progression, as shown in Figure 4.

Indeed, as observed, deletion of Cav2 in glial cells had a more pronounced effect on tumor progression. However, in the following sections (Figure 5), we continued to investigate the role of nerve-CAV2 in tumor growth.

In Figures 5 and 6, we utilized trigeminal ganglia from Cav2^{-/-} mice as the main study model (with Cav2^{+/+} trigeminal ganglia as the control). The trigeminal ganglia contain both neuronal and non-neuronal cells, including Schwann cells, satellite glial cells, a small number of fibroblasts, and an even smaller number of immune cells.. Therefore, our investigation in these sections focused on Cav2 in neural cells, including both neuronal and glial cell Cav2, rather than exclusively on neuronal Cav2.

In cancer microenvironment research, it is typical to specify the phenotypic changes caused by the deletion of specific genes in particular cell subtypes, such as fibroblasts, endothelial cells, and various immune cells. However, cancer neuroscience is an emerging field, and the complexity of the ganglia presents unique challenges. In recent studies investigating tumor-nerve interactions published in prominent journals, dorsal root ganglia and trigeminal ganglia are commonly used as the minimal unit in vitro[7-12], comprising various types of neuronal and glial cells, along with a small number of fibroblasts and immune cells[23]. The absence of well-established separation techniques for these cell subpopulations has limited further differentiation of their specific roles. In vivo studies face similar challenges—although tools such as Aldh111-Cre and GFAP-Cre have been developed to target glial cells in the central nervous

system, there are currently no well-validated Cre lines available for specifically targeting satellite cells or Schwann cells in trigeminal or dorsal root ganglia. Aldh111 has been reported to be expressed in glial cells of the ganglia[6], which is why we employed the Aldh111-Cre mouse model as a preliminary tool to investigate the role of glial Cav2. However, this approach has limitations, primarily due to insufficient specificity, as Aldh111 is also expressed in astrocytes in the central nervous system, and the effects of CAV2 in these astrocytes on HNSCC are not well understood. These limitations are acknowledged and discussed in the revised manuscript (Please see the marked revised manuscript, lines 598-608). Despite these constraints, we believe that the use of Aldh111^{CreERT2}Cav2^{ff} mice in our in vivo experiments represents a meaningful exploration of the functional roles of ganglia cell subpopulations, contributing to an underexplored area of research.

To address your concerns more comprehensively, we sought to further explore the distinct roles of neuronal and glial Cav2 in tumor biology. Specifically, we aimed to separate neurons and glial cells from the ganglia for co-culture with tumor cells to better delineate the effects of neuronal versus glial Cav2. The first approach we considered was using FACS to sort neurons and glial cells based on molecular markers. However, our review of the literature revealed that studies using FACS to isolate neurons typically rely on transgenic mice expressing fluorescent proteins rather than specific markers[21, 26-28], indicating a lack of reliable molecular markers for sorting neurons and glial cells from TG or DRG. Additionally, these studies often use isolated neurons for

sequencing rather than for subsequent culture, as neurons are highly fragile and may not withstand FACS sorting for functional assays.

Our second approach involved crossing mTmG mice with Adv-Cre mice, where tdTomato is expressed in Adv⁺ neurons and EGFP is expressed in other cells, allowing for sorting of these populations and subsequent infection with control or shCAV2 lentivirus. However, we observed that the dual stresses of FACS sorting and lentiviral infection significantly compromised cell viability, making further functional assays unfeasible.

As a third alternative, we crossed mTmG mice with Adv-Cre mice to obtain mTmG^{+/+};Adv-Cre mice, and further crossed them with Cav2^{+/-} mice to generate mTmG^{+/+};Adv-Cre;Cav2^{+/-} mice. Due to the close proximity of the mTmG and Cav2 loci on chromosome 6, only heterozygous mice could be obtained. We sorted Adv-negative cells, which primarily consisted of non-neuronal (mainly glial) cells, and co-cultured them with tumor cells. The tumor cells co-cultured with these glial cells exhibited reduced mitochondrial superoxide level. However, due to the complexity of the genetic background, the availability of offspring was very limited, and we could only perform a limited number of functional assays. These experiments nevertheless demonstrated that CAV2 in non-neuronal cells from the TG can affect tumor cell behavior (Please see the marked revised manuscript, lines 355-370 and Supplementary Figure 6E-F).

Given the challenges in sorting neurons with intact functionality, we adopted an alternative method. Trigeminal ganglia from $CAV2^{+/+}$ and $CAV2^{-/-}$ mice were cultured ex vivo and treated with cytosine arabinoside (AraC) to suppress the proliferation of non-neuronal cells[29]. The remaining neurons were then co-cultured with tumor cells, and mitochondrial superoxide level was assessed. Similar to our observations with glial cells, tumor cells co-cultured with neurons from $CAV2^{-/-}$ mice exhibited a smaller reduction in mitochondrial membrane potential compared to those co-cultured with neurons from $CAV2^{+/+}$ mice; however, the difference did not reach statistical significance. (Please see the marked revised manuscript, lines 355-370 and Supplementary Figure 6E-F).

We appreciate your attention to these details and hope that our efforts to elucidate the specific contributions of neuronal and glial $CAV2$ help address your concerns. Should you have any further suggestions or questions, we would be grateful for your guidance.

Thank you once again for your valuable feedback.

11. Furthermore, why do authors analyze the influence of neural Cav-2 on the metabolism of cancer cells? Why $CAV2$ should affect the metabolism of the cancer cell? Which is the rationale? Although the observations are interesting the mechanism is

lacking.

Response: Thank you very much for your professional comment highlighting the lack of mechanistic rationale in our analysis of neural CAV2's influence on tumor cell metabolism. In the revised manuscript, we have made every effort to further explore the mechanisms by which neural CAV2 affects tumor cells. Please refer to our response to Question 1 for detailed information. The reason we analyzed the influence of neural Cav-2 on the metabolism of cancer cells is that, as shown in Figure 5A, GSEA analysis of our sequencing results suggested that genes regulated by neural Cav-2 in tumor cells were enriched in the oxidative phosphorylation gene set.

We sincerely appreciate your insightful feedback, and we hope that these additional clarifications help address your concerns. Thank you once again for your valuable input.

12. Then, in the last paragraph, the authors used the 4-NQO-induced model of HSSC and they evaluated the impact of Cav2 depletion, either globally and specifically in sensory neurons or glial cells. Also in this case, upon deletion of Cav2 globally or in glial cells only 1/10 mice developed cancer. The impact of Cav2 deletion in sensory neurons is modest. The authors concluded that these results highlight the importance of Cav-2 in glial cells on tumor initiation. Furthermore, the authors concluded the manuscript by evaluating the impact of the global deletion on tumor growth and

metastasis.

Response: Thank you for your comment. We believe that Questions 12 and 13 address the same underlying issue. Therefore, please refer to our response to Question 13 for a comprehensive answer.

13. As mentioned above, why the authors did not further explore the role of Cav2 in glial cells on tumor growth.

Response: Thank you for raising this important question. In our response to Question 10, we provided a detailed explanation of our efforts to explore the impact of CAV2 in glial cells on tumor growth. We believe that the response to Question 10 partially addresses your concerns regarding this question.

Additionally, we acknowledge that including 30-week data from the 4-NQO-induced tumor model in $Aldh111^{CreERT2}Cav2^{f/f}$ and control mice would further strengthen our findings. However, breeding a sufficient number of $Aldh111^{CreERT2}Cav2^{f/f}$ mice requires several months, and conducting the 4-NQO-induced tumor model and subsequent experiments would take at least an additional eight months. Given that the editor requested revisions within three months, we have already requested an extension, but there remains insufficient time to complete this additional experiment.

We hope you understand the time constraints we are facing. If you feel strongly about the importance of this experiment, we are willing to discuss the possibility of requesting a further extension from the editor to complete this work. We appreciate your understanding and consideration.

Thank you once again for your valuable feedback.

14. In my opinion, the logical description of data in the entire manuscript is confused

Response: We sincerely apologize for any confusion caused by the logical organization of the data descriptions in the original manuscript. As mentioned previously, we have made several revisions to the manuscript to improve its clarity and logical flow. We have reorganized sections to ensure a more coherent presentation of the data and have refined the narrative to better link experimental findings with their interpretations.

If you have any further comments or suggestions, please do not hesitate to share them. We highly value your input and are committed to enhancing the quality of our work. Thank you once again for your valuable feedback.

Additional questions:

15.(3) “Depletion of Neural Cav2 Impedes Tumor Cell Infiltration Toward Nerves and

Tumor-Associated Neurogenesis”. In this paragraph, the authors provided only in vitro evidence. Is it also true in vivo? The paper would benefit from addressing PNI in vivo, in both the xenograft and in the 4-NQO-induced murine HNSCC model.

Response: Thank you for your insightful suggestion. As per your recommendation, we have incorporated additional data on perineural invasion (PNI) in both the xenograft model and the 4-NQO-induced murine HNSCC model in the revised manuscript (Please see the marked revised manuscript, lines 258-261, 296-297, 323-326, 491-492; Figure 4D, 4G, 4P; and Figure 7S). However, we observed significant variability in tumor volumes between the experimental groups, which could confound the analysis of PNI incidence (larger tumor volumes were associated with an increased likelihood of PNI). To mitigate this issue, we believe that the proportion of the β 3-tubulin-stained area, as presented in the original manuscript, may provide a more consistent and reliable measure of the influence of neural CAV2 on tumor-nerve interactions in vivo. Furthermore, analysis of TCGA data supports our in vivo findings, demonstrating that neural CAV2 expression is not significantly associated with T or N stage but does correlate with PNI occurrence, thereby reinforcing the relevance of neural CAV2 in tumor-nerve dynamics.

We appreciate your insightful feedback and are committed to addressing all your concerns to improve the quality of our work. Thank you once again for your valuable input.

16.(4) “Cav2+ Nerve-Driven Shift to Enhanced Mitochondrial Respiratory Phenotype in Cancer Cells”

This conclusion derives from in vitro data. Does Cav2-expression in TG cells and glial cells affect the metabolism of the cancer cells in vivo?

In addition to the evaluation of oxygen consumption rates, did the authors also measure glycolysis and lactate production/secretion in vitro?

Response: Thank you for your valuable comments. Indeed, incorporating additional in vivo data could further enhance the robustness of our conclusions. In the original manuscript (Figure 4), we used pS6 as a general marker of metabolic changes to evaluate the impact of systemic Cav2 knockout, neuron-specific Cav2 knockout, and glial cell-specific Cav2 knockout on HNSCC orthotopic tumors. We observed significant alterations in pS6 expression. However, after identifying that neural CAV2 primarily influences the oxidative phosphorylation (OXPHOS) phenotype of tumor cells (figure 5), we did not further assess OXPHOS-specific markers in vivo, which was a limitation of the original submission.

In the revised manuscript, we have addressed this issue by evaluating the expression of several OXPHOS-related markers (SDHB, UQCRC, ATP5A1, NDUFB8) in the tumors from the mice shown in Figure 4. We found that the expression levels of these OXPHOS markers were higher in orthotopic tumors from wild-type mice

compared to those from Cav2^{-/-} mice. Similarly, control mice showed higher expression compared to Aldh111^{CreERT2}Cav2^{ff} and Adv^{Cre}Cav2^{ff} mice (please see the marked revised manuscript, lines 379-386 and Supplementary Figure 7A).

Additionally, in the revised manuscript, we evaluated lactate secretion changes after co-culturing trigeminal ganglia with tumor cells. The results indicate that tumor cells co-cultured with Cav2^{-/-} ganglia exhibited increased lactate secretion compared to those co-cultured with Cav2^{+/+} ganglia (please see the marked revised manuscript, lines 358-378).

We appreciate your insightful feedback, and we hope that these additional analyses help to address your concerns and further strengthen the manuscript. Thank you once again for your valuable input.

17.(5) “Depletion of Cav2 Inhibits 4-NQO-Induced Oral Tumorigenesis”

In this paragraph and related picture, more experiments are required. For example:

Which is the impact on animal survival? Kaplan-Maier curves should be reported for the three models: global Cav-2 KO, Sensory neurons Cav-2 KO, glial cells Cav-2 KO.

Concerning the Sensory neurons Cav-2 KO - glial cells Cav-2 KO, the authors did not showed tumor volume or metastasis.

Is there an impact on PNI or tumor innervation in the three mouse models (global Cav-2 KO, Sensory neurons Cav-2 KO, glial cells Cav-2 KO) upon 4-NQO-Induced Oral

Tumorigenesis?

Response: Thank you for your valuable comments. In our study, we found that neural CAV2 primarily promotes cancer stemness by facilitating the metabolic shift of tumor cells towards oxidative phosphorylation. Therefore, our focus in this section was on tumor initiation, as it is more closely related to cancer stemness compared to tumor progression.

We agree that providing additional data, such as the impact of neuron-specific or glial cell-specific Cav2 knockout on the progression of 4-NQO-induced HNSCC (30-week endpoint data), would strengthen our findings. However, breeding a sufficient number of conditional knockout mice requires several months, and performing the 4-NQO-induced tumor model and subsequent experiments would require at least an additional eight months. Given that the editor requested revisions within three months, we have already requested an extension, but there remains insufficient time to complete this experiment. Providing survival data for the mice would require an even longer timeline of at least one year.

We sincerely hope you understand the time constraints we are facing. If you believe these experiments are crucial, we are willing to discuss the possibility of requesting a further extension from the editor to complete this work. We greatly value your input and are committed to enhancing the quality of our work.

Thank you once again for your understanding and valuable feedback.

Minor questions:

18.(6) The quality of all immunofluorescence analyses is poor. Images with higher resolution are required. For example:

-Fig. 1F: the colocalization between CAV2 and TRPV1 is difficult to observe.

-Fig. 2H: the colocalization between CAV2 and TdTomato is not clear, higher resolution is necessary.

-Fig 2J: The increased expression of Cav2 in nociceptive neurons in the 4-NQO-induced murine HNSCC model is not evident. Higher resolution and higher magnification images are required.

-The resolution of transmission electron microscopy images (TEM) pictures is low (Fig.2E and 2F).

Response: Thank you for highlighting a significant shortcoming in our manuscript. In fact, the confocal microscope we utilized is advanced, and the images captured were initially clear. However, the resolution of the images significantly decreased during the conversion to the PDF format for review, resulting in unclear visuals, for which we sincerely apologize. To address this issue, we have included all original fluorescence and immunohistochemistry images in the supplementary materials of the revised version for review. Additionally, should the manuscript be accepted, we will collaborate

with the publisher's staff to explore ways to maintain the original resolution of the images in the final publication as much as possible. Thank you once again for your understanding and valuable feedback.

19. The manuscript would also benefit from image analysis of the co-localization efficiency.

Response: Thank you very much for your suggestion. Your recommendation to analyze co-localization efficiency would indeed enhance the quality of our manuscript. We have attempted to conduct co-localization efficiency analysis using ImageJ software for our multicolor immunohistochemistry images. However, we have encountered a limitation—ImageJ seems to be primarily capable of analyzing co-localization efficiency for red and green fluorescence channels, whereas our images include additional colors such as yellow and cyan. If you have any alternative methods or specific suggestions for conducting such an analysis, we would greatly appreciate your guidance. Thank you once again for your valuable feedback and support.

20.(7) Figure 1H shows the correlation between overall survival and CAV2 expression, based on TCGA cohort sequencing. Which is the difference between this analysis and the one the same authors reported in the previous paper (<https://doi.org/10.1038/s41420-022-01176-1>)?

Response: In Figure 1H, we analyzed the relationship between CAV2 mRNA levels, obtained from bulk tumor sequencing of the TCGA cohort, and patient outcomes. In contrast, the analysis presented in our previous paper (<https://doi.org/10.1038/s41420-022-01176-1>) assessed the intensity of CAV2 staining in HNSCC patient tissue samples from our institution using immunohistochemistry, and examined its correlation with prognosis. The primary differences between these analyses lie in the populations studied and the level of CAV2 measured: mRNA levels in the TCGA analysis versus protein levels in our previous study. Importantly, both studies evaluated CAV2 levels in bulk tumor areas without detailed assessment of CAV2 expression in specific cell types.

We hope this clarifies the distinctions between the two analyses, and we appreciate your attention to this matter. Thank you once again for your valuable feedback.

21.(8) The authors showed that CAV2 expression and the number of caveolae increases upon co-culture of TG cells with cancer cells. Which is the underlying mechanism?

Response: Thank you for your insightful query, which has significantly deepened our understanding of caveolae dynamics. Initially, we assumed that the observed increase in caveolae was primarily due to the upregulation of CAV2 expression following co-culture with cancer cells. However, after considering your question, we revisited the literature and discovered that caveolae formation is mainly dictated by the expression levels of CAV1 and CAV3, whereas the influence of CAV2 on caveolae formation is

relatively minimal[30].

Therefore, it is likely that the observed increase in caveolae in our study is attributable to the upregulation of CAV1 expression following co-culture, given that CAV3 is predominantly expressed in muscle tissues. We have added this clarification to the revised manuscript to better explain the underlying mechanisms (please see the marked revised manuscript, lines 185-189).

Thank you once again for your valuable feedback, which has helped refine our understanding and improve the quality of our manuscript.

22.(9) There is no information about statistical analyses in the figure legends: type of analysis, N, P values are absent. What is represented in the graph? Media +/- SD or SEM? This is true for Fig.2 and others. many figure legends are poorly described

Response: Thank you very much for pointing out the deficiencies and areas of imprecision in our manuscript. We have carefully revised the figure legends in the revised manuscript to include all necessary information, such as the type of statistical analysis used, sample size (N), and P values, as well as clarifying whether the data are represented as mean +/- SD or SEM. Your feedback is greatly appreciated and has been instrumental in enhancing the clarity and detail of our presentation.

Thank you once again for your valuable input.

23.(10) “Ki-67 and cleaved caspase-3 markers to probe tumor proliferation and apoptosis yielded no notable disparities between the two mouse cohorts”.

Which is the reason for these observations?

What about necrosis?

Response: Thank you for your insightful questions. In our co-culture experiments, we observed differences in the proliferation rates of HNSCC cells co-cultured with CAV2^{+/+} versus CAV2^{-/-} ganglia, with a higher proportion of EdU-positive proliferating cells in the tumor cells co-cultured with CAV2^{+/+} ganglia. However, as you pointed out, in the in vivo experiments, we did not observe significant differences in tumor proliferation or apoptosis as indicated by Ki-67 and cleaved caspase-3 markers. We have objectively reported these observations, though elucidating the reasons behind negative results can be challenging, and we sincerely apologize for not being able to provide a definitive explanation.

Following your suggestion, we evaluated necrosis using Hmgb1 as a marker to assess differences between the two mouse cohorts. The results indicated that there was no significant difference in necrosis between the groups (Please see the marked revised manuscript, lines 266-267, and Supplementary Figure 4C).

Thank you again for your valuable feedback, which has helped us further refine our study.

24.(11) “platinum-based chemotherapeutics revealed that SCC15 cells co-cultured with Cav2^{+/+} TGs displayed reduced apoptosis following cisplatin treatment compared to SCC15 cells co-cultured with Cav2^{-/-} TGs”.

What about in vivo?

Response: Indeed, in vivo models are essential for assessing resistance to platinum-based chemotherapy. Following your suggestion, we attempted to include in vivo results in our study. However, as illustrated in Figure 4, there is a substantial difference in the growth rates of tumors implanted in WT mice compared to CAV2^{-/-} mice. This variation makes it challenging to consistently administer platinum-based treatments and subsequently assess apoptosis and resistance when tumor sizes vary significantly. If you could provide any specific and feasible experimental design recommendations, we would greatly appreciate it and are prepared to perform supplementary experiments to enhance our understanding.

Additionally, in response to Reviewer 1's comments regarding the assessment of the impact of host/neural CAV2 on HNSCC stemness in vivo, we have examined the expression of HNSCC cancer stem cell markers in mouse tumor tissues, as shown in Figures 4. The results indicate that the expression of stemness markers in oral

orthotopic tumors from Cav2^{+/+} mice was significantly higher compared to those from Cav2^{-/-} mice. Furthermore, in Adv^{Cre}Cav2^{f/f} and Aldh111^{CreERT2}Cav2^{f/f} conditional knockout mice, the expression of stemness markers in oral orthotopic tumors was also significantly lower compared to control mice, with a relatively smaller difference observed between Adv^{Cre}Cav2^{f/f} and control mice. In addition, we performed multicolor immunohistochemical staining on 4-NQO-induced tumors presented in Figure 7, and found that the expression of stemness markers in 4-NQO-induced tumors from Cav2^{+/+} mice was also significantly higher compared to Cav2^{-/-} mice. The results have been incorporated into the revised manuscript for a more comprehensive evaluation (please see the marked revised manuscript, lines 451-462 and Figure 6K and Supplementary Figure 10).

Thank you once again for your valuable feedback, which has been instrumental in refining our study.

In summary, we have made every effort to address all of your concerns and hope that our revisions meet with your approval. We sincerely appreciate your valuable feedback and welcome any further comments you may have.

References (response letter):

1. Viale A, Pettazoni P, Lyssiotis CA et al. Oncogene ablation-resistant pancreatic cancer cells depend on mitochondrial function. *Nature* 2014; 514: 628-632.
2. Lee KM, Giltzane JM, Balko JM et al. MYC and MCL1 Cooperatively Promote Chemotherapy-Resistant Breast Cancer Stem Cells via Regulation of Mitochondrial Oxidative Phosphorylation. *Cell*

Metab 2017; 26: 633-647 e637.

3. Carey BW, Finley LW, Cross JR et al. Intracellular alpha-ketoglutarate maintains the pluripotency of embryonic stem cells. *Nature* 2015; 518: 413-416.
4. Rodriguez-Colman MJ, Schewe M, Meerlo M et al. Interplay between metabolic identities in the intestinal crypt supports stem cell function. *Nature* 2017; 543: 424-427.
5. Zhang H, Ryu D, Wu Y et al. NAD(+) repletion improves mitochondrial and stem cell function and enhances life span in mice. *Science* 2016; 352: 1436-1443.
6. Rabah Y, Rubino B, Moukarzel E, Agulhon C. Characterization of transgenic mouse lines for selectively targeting satellite glial cells and macrophages in dorsal root ganglia. *PLoS One* 2020; 15: e0229475.
7. Padmanaban V, Keller I, Seltzer ES et al. Neuronal substance P drives metastasis through an extracellular RNA-TLR7 axis. *Nature* 2024; 633: 207-215.
8. Hirth M, Gandla J, Hoper C et al. CXCL10 and CCL21 Promote Migration of Pancreatic Cancer Cells Toward Sensory Neurons and Neural Remodeling in Tumors in Mice, Associated With Pain in Patients. *Gastroenterology* 2020; 159: 665-681 e613.
9. Amit M, Takahashi H, Dragomir MP et al. Loss of p53 drives neuron reprogramming in head and neck cancer. *Nature* 2020; 578: 449-454.
10. Banh RS, Biancur DE, Yamamoto K et al. Neurons Release Serine to Support mRNA Translation in Pancreatic Cancer. *Cell* 2020; 183: 1202-1218 e1225.
11. Renz BW, Takahashi R, Tanaka T et al. beta2 Adrenergic-Neurotrophin Feedforward Loop Promotes Pancreatic Cancer. *Cancer Cell* 2018; 33: 75-90 e77.
12. Zhang Y, Lin C, Liu Z et al. Cancer cells co-opt nociceptive nerves to thrive in nutrient-poor environments and upon nutrient-starvation therapies. *Cell Metab* 2022; 34: 1999-2017 e1910.
13. Jurcak NR, Rucki AA, Muth S et al. Axon Guidance Molecules Promote Perineural Invasion and Metastasis of Orthotopic Pancreatic Tumors in Mice. *Gastroenterology* 2019; 157: 838-850 e836.
14. Scanlon CS, Banerjee R, Inglehart RC et al. Galanin modulates the neural niche to favour perineural invasion in head and neck cancer. *Nat Commun* 2015; 6: 6885.
15. Schmitd LB, Perez-Pacheco C, Bellile EL et al. Spatial and Transcriptomic Analysis of Perineural Invasion in Oral Cancer. *Clin Cancer Res* 2022; 28: 3557-3572.
16. Gil Z, Cavel O, Kelly K et al. Paracrine regulation of pancreatic cancer cell invasion by peripheral nerves. *J Natl Cancer Inst* 2010; 102: 107-118.
17. Sinha S, Fu YY, Grimont A et al. PanIN Neuroendocrine Cells Promote Tumorigenesis via Neuronal Cross-talk. *Cancer Res* 2017; 77: 1868-1879.
18. Balood M, Ahmadi M, Eichwald T et al. Nociceptor neurons affect cancer immunosurveillance. *Nature* 2022; 611: 405-412.
19. McIlvried LA, Atherton MA, Horan NL et al. Sensory Neurotransmitter Calcitonin Gene-Related Peptide Modulates Tumor Growth and Lymphocyte Infiltration in Oral Squamous Cell Carcinoma. *Adv Biol (Weinh)* 2022; 6: e2200019.
20. Gohrig A, Detjen KM, Hilfenhaus G et al. Axon guidance factor SLIT2 inhibits neural invasion and metastasis in pancreatic cancer. *Cancer Res* 2014; 74: 1529-1540.
21. Lopes DM, Denk F, McMahon SB. The Molecular Fingerprint of Dorsal Root and Trigeminal Ganglion Neurons. *Front Mol Neurosci* 2017; 10: 304.
22. Megat S, Ray PR, Tavares-Ferreira D et al. Differences between Dorsal Root and Trigeminal Ganglion Nociceptors in Mice Revealed by Translational Profiling. *J Neurosci* 2019; 39: 6829-6847.

23. Yang L, Xu M, Bhuiyan SA et al. Human and mouse trigeminal ganglia cell atlas implicates multiple cell types in migraine. *Neuron* 2022; 110: 1806-1821 e1808.
24. Zhang Z, Liu R, Jin R et al. Integrating Clinical and Genetic Analysis of Perineural Invasion in Head and Neck Squamous Cell Carcinoma. *Front Oncol* 2019; 9: 434.
25. Spencer A, Yu L, Guili V et al. Nerve Growth Factor Signaling from Membrane Microdomains to the Nucleus: Differential Regulation by Caveolins. *Int J Mol Sci* 2017; 18.
26. Chiu IM, Barrett LB, Williams EK et al. Transcriptional profiling at whole population and single cell levels reveals somatosensory neuron molecular diversity. *Elife* 2014; 3.
27. Zhang C, Hu MW, Wang XW et al. scRNA-sequencing reveals subtype-specific transcriptomic perturbations in DRG neurons of Pirt(EGFPf) mice in neuropathic pain condition. *Elife* 2022; 11.
28. Wang K, Wang S, Chen Y et al. Single-cell transcriptomic analysis of somatosensory neurons uncovers temporal development of neuropathic pain. *Cell Res* 2021; 31: 904-918.
29. Beaudoin GM, 3rd, Lee SH, Singh D et al. Culturing pyramidal neurons from the early postnatal mouse hippocampus and cortex. *Nat Protoc* 2012; 7: 1741-1754.
30. Parton RG, Simons K. The multiple faces of caveolae. *Nat Rev Mol Cell Biol* 2007; 8: 185-194.

REVIEWER COMMENTS

Reviewer #1 (Remarks to the Author):

The author has presented additional evidence to support their intriguing findings. Although the mechanisms are not yet fully understood, the novelty of this study would be a significant contribution to the field.

Response: We are sincerely honored by your positive review and deeply appreciate your recognition of our work. We acknowledge that we have not yet fully elucidated the role of neural CAV2 in the development and progression of HNSCC. In our future studies, we plan to delve deeper into these mechanisms and explore additional molecular pathways through which neural factors may influence HNSCC pathogenesis. We greatly value your insights and believe they will guide and strengthen our ongoing research in this field.

Reviewer #2 (Remarks to the Author):

The authors did an excellent job in this revised manuscript. They addressed all of my questions, many of which required additional experiments. I believe that the modifications have significantly improved the manuscript and strengthened their findings and conclusions.

Response: We are deeply honored by your positive review and greatly appreciate the time and effort you invested in evaluating our manuscript. Your professional expertise and detailed comments have been invaluable in guiding and refining our work. We sincerely thank you for the insightful feedback, which significantly enhanced the quality of our research and strengthened our conclusions.

Reviewer #3 (Remarks to the Author):

The authors answer almost all the concerns. I appreciated the efforts made to investigate the mechanism through which Cav2 regulates OXPHOS in cancer cells. We also understand that fully elucidating this mechanism may not be feasible within the limited timeframe of the revision. The authors have provided new insights that could contribute to understanding this process, including the identification of several mitochondrial proteins in the conditioned medium. However, it is not immediately intuitive how these secreted proteins might modulate OXPHOS in cancer cells. The identified proteins are not usually secreted by the cell. Are there any published studies describing something similar? The manuscript may benefit from further discussion on this point.

Response: We are sincerely grateful for your positive feedback on our work. We also deeply appreciate the time and expertise you devoted during the peer-review process, as your thoughtful comments have greatly improved our manuscript. After a thorough

literature search, we found no reports closely resembling our observations. In light of your valuable suggestions, we have expanded the discussion section to further address these points (please see the marked revised manuscript, lines 643-650).

REVIEWER COMMENTS

Reviewer #5 (Remarks to the Author):

This manuscript presents exciting findings demonstrating that HNSCC cells can induce CAV2 expression in trigeminal ganglia and associated neural fibers, which in turn promotes mitochondrial oxidative phosphorylation (OxPhos) in cancer stem-like cells (CSCs), reinforcing their stemness and contributing to tumor aggressiveness. The data highlight a unique cancer - nerve metabolic interaction, potentially advancing our understanding of how the tumor microenvironment supports CSC function and resistance mechanisms.

The authors support their conclusions with a range of in vitro, in vivo, and transcriptomic analyses, including TCGA data showing CAV2 as a strong prognostic marker in HNSCC. Importantly, they demonstrate that CAV2-positive nerves contribute to a metabolic shift toward OxPhos specifically in CSC mitochondria, a mechanistic insight of potential therapeutic significance.

While the study is generally well-conducted and the conclusions are compelling, several points require clarification or further elaboration.

Minor Comments:

Line 164 / Figure 2A: The text references a comparison between exposed and non-exposed trigeminal ganglia (TGs), but this comparison is not shown in the figure.

Including this in the figure panel would enhance clarity and support the interpretation.

Response: Thank you for carefully reviewing our manuscript and highlighting this unclear phrasing. The figure indeed presents both TGs exposed to tumor cells for various durations (day 1–day 5) and TGs not exposed to tumor cells (day 0). To ensure that readers readily recognize day 0 as the condition in which TGs were cultured alone, we have revised the figure legend for greater clarity:

(A) Immunoblot analysis of Cav2 in TGs that were either cultured alone (day 0) or co-cultured with SCC15 or MOC2 cells for 5 days. This experiment was independently repeated three times.

We trust that this amendment will clarify the comparison for you and other readers.

Please let us know if you have any further suggestions.

Figures 2G and 2H: The “sham” condition needs clarification. Does it refer to animals that had no tumor inoculation, or a different control setup? Please define “sham” explicitly in both the figure caption and methods.

Response: Thank you for your insightful comment. We apologize for the lack of clarity in the original manuscript. As per your suggestion, we have now explicitly defined the

sham condition in both the legend for Figures 2G and 2H and in the Methods section as follows:

“Sham controls were injected with tumor-cell-free buffer (30 μ L of DMEM/Matrigel at a 1:1 ratio).”

We believe this revision enhances the rigor and readability of our presentation. Please let us know if you have any further suggestions.

Figure 5: GSEA Analysis (GEO data): The authors report DNA repair as an enriched pathway, but analysis of the referenced GEO dataset primarily reveals OxPhos and MYC target enrichment, no DNA Repair. However, it is unclear whether a differential gene expression (DEG) step was performed prior to GSEA. If so, this should be described in the Methods section. Otherwise, revise the pathway enrichment claims accordingly.

Response: Thank you for this insightful comment. To clarify, we did not perform a separate differential gene expression (DEG) step prior to GSEA, so that all genes would be included in the analysis. Upon re-examination of the WT dataset, the top enriched Hallmark gene sets were, in descending order, HALLMARK_OXIDATIVE_PHOSPHORYLATION, HALLMARK_MYC_TARGETS_V1, HALLMARK_PANCREAS_BETA_CELLS,

HALLMARK_FATTY_ACID_METABOLISM, and HALLMARK_PEROXISOME. HALLMARK_DNA_REPAIR ranked sixth and was therefore not among the most prominent pathways. You are correct that emphasizing the sixth-ranked term over the top two is not sufficiently rigorous.

In the revised manuscript, we have:

Results section: updated the Results section to reflect that OxPhos and MYC target gene sets are the most significantly enriched pathways. (Please see line 320)

Methods section: added a detailed description of the GSEA workflow, stating that no prior DEG step was applied and specifying the ranking metric used. (Please see lines 905-907)

Supplementary Data: provided a supplementary Excel file containing the full GSEA output—including gene set names, sizes, enrichment scores (ES, NES), nominal p-values, FDR q-values, FWER p-values, rank at maximum, and leading edge details—for readers' comprehensive review.

We hope these revisions address your concern and render the pathway-enrichment analysis fully transparent and reproducible. Please let us know if any additional clarification would be helpful.

Line 411: Please add a sentence indicating how the observed gene expression profile

aligns more closely with ALDH⁺ or ALDH⁻ cells, as reported in Reference 35, which is on breast cancer. This would help contextualize the findings for HNSCC within known CSC phenotypes.

Response: Indeed, as you rightly pointed out, our original wording lacked clarity and could potentially confuse readers. We have revised the description as follows:

“Remarkably, the gene expression profiles of HNSCC cells cultured with Cav2^{+/+} nerves compared to those cultured with Cav2^{-/-} nerves (shown in Figure 5A–C) closely resemble the differential patterns noted between ALDH⁺ and ALDH⁻ cell populations, including the upregulation of OXPHOS and MYC target gene signatures that characterize ALDH1⁺ breast cancer CSCs (Figure 5A–B of ref.35).”

This explicitly highlights that the CSC-associated gene expression profile reported in reference 35 is characterized by the upregulation of OXPHOS and MYC target gene signatures. We welcome any further suggestions you may have to improve this description.

Line 413: The manuscript describes GSEA of SCC15 cells, but what were the results for MOC2 cells? These should be included for completeness.

Response: Thank you for raising this point. I will address it together with the next

comment.

Figures 6H & 6I: Are there equivalent results available for MOC2 cells in these panels?

Response: Thank you for highlighting this important point. We agree that including the corresponding analyses in MOC2 cells (GSEA and the equivalents of Figs. 6H and 6I) would enhance the completeness and rigor of the study. Conceptually, these experiments are replications of assays already performed and are technically straightforward.

However, the manuscript received provisional acceptance on January 20, 2025 (**not final acceptance**). Subsequently, the journal requested deposition of all sequencing datasets in an accessible public repository; because our initial submission did not fully meet this requirement, the manuscript re-entered external review. During the intervening months, in adherence to animal-welfare considerations, we did not expand breeding of Cav2^{-/-} transgenic mice. The additional experiments you propose would require a substantial number of Cav2^{-/-} animals, as each mouse yields only two trigeminal ganglia, and thus would necessitate an extended breeding period.

In light of these constraints, we respectfully ask whether the requirement for a full set of MOC2 confirmatory data could be waived, or whether partial supplementation would be acceptable. If you deem these experiments essential, we will consult the editor to

request an extended revision timeline and will prioritize completing the studies as expeditiously as possible. We appreciate your understanding.

Figure 8F: The term “differentially expressed genes (DEGs)” is incorrect here, as the data refer to proteins. Please revise to “differentially expressed proteins” to avoid confusion.

Response: Thank you for your professional observation regarding the terminology error. In the revised manuscript, we have corrected “differentially expressed genes” to “differentially expressed proteins” as suggested to avoid any confusion.

REVIEWER COMMENTS

Reviewer #5 (Remarks to the Author):

I would like to thank the authors for their careful and detailed responses to the comments raised during this round of review. I appreciate the effort that has gone into addressing the concerns and clarifying points raised, as well as the thoughtful explanations provided regarding the feasibility of additional experiments.

Thank you for your thoughtful explanation regarding the feasibility of generating MOC2 data. I agree that inclusion of GSEA and corresponding analyses in MOC2 cells would improve the completeness of the study. At the same time, I understand the logistical and ethical challenges associated with additional *Cav2*^{-/-} breeding.

Given this context, I do not consider the absence of full MOC2 replication to be a barrier to publication. The manuscript already presents a substantial body of data that supports the conclusions, and the addition of MOC2 experiments would, in my view, primarily serve to further strengthen completeness rather than alter the central findings.

If feasible, the inclusion of any partial or surrogate MOC2 analyses would still be a valuable addition, but I leave this to the editors' and authors' discretion. Overall, I find the manuscript suitable for publication if now the data availability requirements are fulfilled.

Response: We are sincerely grateful for your thoughtful and supportive review. We deeply appreciate your kind recognition of the efforts we made to address the previous concerns and your understanding regarding the logistical and ethical challenges associated with additional Cav2^{-/-} MOC2 experiments.

We fully acknowledge the importance of validating our findings across multiple cell lines to strengthen the robustness and generalizability of the study. Although limitations prevented us from conducting additional MOC2 analyses during this revision cycle, we are committed to further exploring these findings in future studies using complementary cell models, whenever feasible.

Once again, we thank you for your understanding and encouragement, which are truly meaningful to us and will continue to guide our future research efforts.